# LIFEBENCH: Evaluating Length Instruction Following in Large Language Models

**Wei Zhang**[1]*, **Zhenhong Zhou**[1]*, **Kun Wang**[2]*, **Junfeng Fang**[3], **Rongwu Xu**[4], **Yuanhe Zhang**[1], **Rui Wang**[1], **Ge Zhang**[5], **Xavier Li**[2], **Li Sun**[6], **Lingjuan Lyu**[7], **Yang Liu**[2], **Sen Su**[1]†

[1]Beijing University of Posts and Telecommunications, [2]Nanyang Technological University,
[3]National University of Singapore, [4]Tsinghua University, [5]M-A-P,
[6]North China Electric Power University, [7]Sony AI

## Abstract

While large language models (LLMs) can solve PhD-level reasoning problems over long context inputs, they still struggle with a seemingly simpler task: *following explicit length instructions—e.g., "write a 10,000-word novel"*. Additionally, models often generate far too short outputs, terminate prematurely, or even refuse the request. Existing benchmarks focus primarily on evaluating the quality, but often overlook whether the generations meet length constraints. To this end, we introduce **Length Instruction Following Evaluation Benchmark** (LIFEBENCH) to comprehensively evaluate LLMs' ability to follow length instructions across diverse tasks and a wide range of specified lengths. LIFEBENCH consists of 10,800 instances across 4 task categories in both English and Chinese, covering length constraints ranging from 16 to 8192 words. We evaluate 26 widely-used LLMs and find that most models reasonably follow short-length instructions but deteriorate sharply beyond a certain threshold. Surprisingly, almost all models fail to reach the vendor-claimed maximum output lengths in practice, as further confirmed by our evaluations extending up to 32K words. Even long-context LLMs, despite their extended input-output windows, counterintuitively fail to improve length-instructions following. Notably, reasoning LLMs outperform even specialized long-text generation models, achieving state-of-the-art length following. Overall, LIFEBENCH uncovers fundamental limitations in current LLMs' length instructions following ability, offering critical insights for future progress.

🐙 **Data & Code:** github.com/LIFEBench/LIFEBench

🤗 **Data & Dataset Card:** huggingface.co/datasets/LIFEBench/LIFEBench

🌸 **Homepage:** ydyjya.github.io/LIFEBench

## 1 Introduction

Large language models (LLMs) [40, 13] demonstrate remarkable capabilities in sophisticated tasks such as long-context understanding, planning, and complex reasoning, among others [46, 100, 40, 48]. Unexpectedly, LLMs also often fail in a seemingly trivial and explicit task [120, 58, 50]: *precisely following length instructions*. Concretely, LLMs tend to terminate generation prematurely, especially when long lengths are instructed [8, 85, 83]. This non-intuitive observation highlights an underexplored but important problem: off-the-shelf language models exhibit shortcomings both in following explicit length instructions and generating long-text content [106, 112]. Understanding and quantifying this limitation is critical, as accurate length control and long-text generation underpin numerous real-world LLM applications and practical productions [29, 70, 98].

---

*indicates equal contribution. † indicates corresponding author.

39th Conference on Neural Information Processing Systems (NeurIPS 2025) Track on Datasets and Benchmarks.

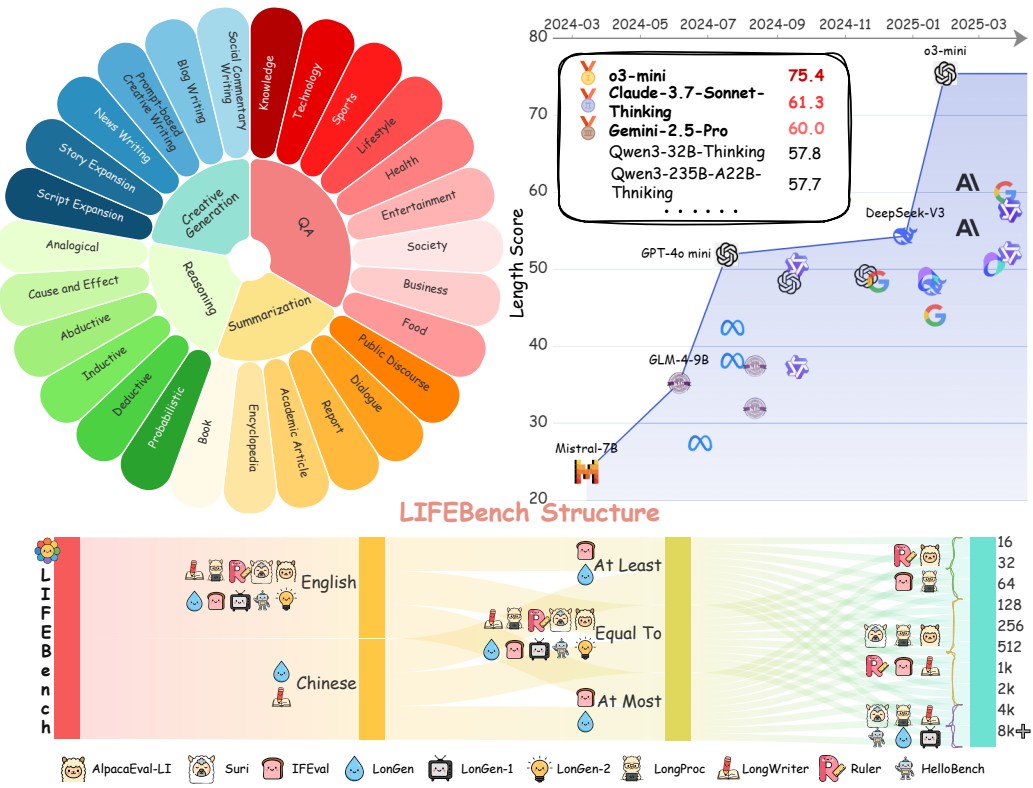

Figure 1: Overview of LIFEBENCH. **Top Left.** LIFEBENCH data types. **Top Right.** Length Instruction Following Leaderboard based on LIFEBENCH. **Bottom.** LIFEBENCH evaluates length generalization capabilities across bilingual content (Chinese/English), incorporating three distinct control methodologies and ten granular length constraints.

To systematically evaluate LLMs' length instruction following and explore why they fail to follow longer length instructions, we formulate our investigation around three core research questions (RQs):

- **RQ1:** To what extent do current LLMs accurately follow explicit length instructions, and what factors may lead to length instructions following failures?

- **RQ2:** How capable is the current LLM at outputting long generations with specific length instructions, and how reliably can they meet their claimed maximum output length?

- **RQ3:** How profoundly do input characteristics impact LLMs' length instructions following?

To address these questions comprehensively, we introduce the **Length Instruction Following Evaluation Benchmark** (LIFEBENCH). LIFEBENCH is the first full-range length instruction evaluation benchmark comprising 10,800 instances across four representative natural language generation (NLG) tasks-Question Answering [30], Summarization [28], Reasoning [69], and Creative Generation [109]—covering a wide spectrum of length constraints ranging from 16 to 8,192 bilingual (English & Chinese) words (Figure 1). Distinct from existing benchmarks that primarily assess generation quality [86, 83, 8, 85], LIFEBENCH focuses on the evaluation of explicit length instruction following capabilities, using two metrics, *Length Deviation* and *Length Score*, to quantify model performance. Compared to simple word count matching, our metrics offer a more analytic and robust evaluation, *Length Deviation* captures deviation direction and magnitude, while *Length Score* ensures robust aggregation, underscoring the superiority of our benchmark.

We conduct extensive experiments on 26 prevalent LLMs, revealing multiple intriguing insights. For *RQ1*, results indicate current LLMs typically follow short length instructions but struggle with long ones. Further analysis reveals a core bottleneck: LLMs are unable to accurately recognize how many words they have generated, which might contribute to length instructions following failures. We find reasoning models slightly address this by calibrating output length during intermediate steps; o3-mini [80] achieves the highest *Length Score* of 75.4, while most models score below 60. For *RQ2*, we show that current models typically cannot approach their vendor-claimed maximum output

length under realistic length constraints. Our further analysis reveals that some models fall short due to inherently limited long-text generation capabilities, while others appear to underperform by lazy strategies (*e.g., falsely claiming inability to follow instructions*). For ***RQ3***, we find that input characteristics critically impact length instruction fidelity. For example, summarization tasks exhibit the largest *Length Score* drop of 18.8, while increasing input length ($1k \rightarrow 8k$ words) reduces by 13.6. Notably, models overextend outputs in Chinese contexts, revealing language-specific biases.

Our contributions are as follows: **(I)** We create LIFEBENCH, the first benchmark to systematically evaluate length instruction following across a full range of tasks and lengths; **(II)** We reveal that the off-the-shelf LLMs have shortcomings in the length instructions following (Figure 1 top right), especially under the long-text generation task, demonstrating that LLMs fall short of their vendor-claimed maximum output length in practice; **(III)** We comprehensively analyze how input factors, such as task type, language, and input length, influence length-following fidelity. Our findings uncover fundamental limitations in current LLMs and provide actionable insights for future development.

## 2 Related Works

**Long Context Capabilities of LLMs.** Long context capabilities refer to the ability of LLMs to effectively comprehend and generate extended texts [52, 68]. Various benchmarks have been proposed to evaluate comprehension over long contexts, including those based on synthetic data [99, 56, 65, 43] and human-annotated real-world tasks [6, 7, 59, 92, 3]. Long-text generation, which is a classic task [10, 19, 104] in natural language generation (NLG), also remains a core focus of long context research. Existing evaluations of long-text generation mainly fall into five categories: question answering [71, 97, 69, 116], summarization [23], instruction following [106, 112], creative writing [81, 85, 107], and multi-dimensional assessment [47, 86]. In addition to evaluation, numerous studies have proposed methods to improve long-text generation [8, 101, 110, 83, 85]. Our benchmark covers both long-context comprehension and long-text generation capabilities.

**Length Instruction Following Capabilities of LLMs.** The capability to follow length instructions is an important aspect of instruction following. Several previous works have incorporated length constraints into instruction following evaluations [122, 115, 103, 117, 84, 17, 105]. MT-Bench-LI and AlpacaEval-LI [114], curated by augmenting MT-Bench [120] and AlpacaEval 2 [27] with length constraints, reveal that models often fail to comply with such constraints. To mitigate this issue, both training-based methods [114, 88, 58, 11] and inference-time approaches using control frameworks or external tools [39, 113] have been proposed. However, these efforts primarily target short length constraints instead of full-range length instruction following.

## 3 LIFEBENCH

In this section, we introduce LIFEBENCH, a benchmark designed to evaluate the length instruction following capability of LLMs across the full range. The remainder of this section provides an overview of the dataset; a detailed comparison with existing benchmarks is provided in Appendix A and correlation analysis with leaderboard is included in Appendix L. LIFEBENCH enjoys the following features:

- **Diverse Dataset:** We ensure dataset diversity based on the following key features: (1) *Various Tasks*: LIFEBENCH includes a broad range of NLG tasks, including question answering, summarization, creative generation, and reasoning, which are well-suited for length instruction following [67]. (2) *Long-context Inputs*: LIFEBENCH includes both short inputs ($<$2,000 words) and long inputs ($>$2,000 words), allowing evaluation across different input scales, unlike most prior benchmarks. (3) *Bilingual Queries*: LIFEBENCH contains Chinese and English data collected separately from distinct datasets, enabling our evaluation to investigate language-specific bias.

- **Extensive Length Constraints:** LIFEBENCH is the first benchmark designed to systematically evaluate LLMs' full-range length instruction following capability, which is based on the following design choices. (1) *Multiple Control Methods*: LIFEBENCH adopts three common length control methods: *Equal To*, *At Most*, and *At Least*. (2) *Length Constraints*: We define ten distinct length constraints spanning short ($<$100 words), medium (100–2000 words), and long ($>$2000 words) ranges, which are more comprehensive than prior works. Detailed control methods and length constraints description can be found in Section 4.2.

- **Analytical Evaluation:** We evaluate length instruction following capability using two evaluation metrics: *Length Deviation* (*LD*) and *Length Score* (*LS*), offering multi-dimensional analysis. *Length Deviation* intuitively measures the extent of deviation between the generation length and the length constraints, while *Length Score* quantifies the model's length instruction following capability. Compared to prior works that merely assess generated length, our evaluation metrics allow comparison across models by quantifying both the magnitude and directionality of deviations. Section 4.3 provides a detailed discussion of these metrics.

**Data Collection and Statistics.** Selecting NLG tasks tailored to evaluating length instruction following is crucial for constructing LIFEBENCH. To this end, we introduce the concept of *Length Paradigms*, which categorize NLP tasks based on their inherent length characteristics. We assign common NLG tasks into four paradigms: *Length-Independent Paradigm*, *Fixed-Length Paradigm*, *Adaptive-Length Paradigm*, and *Flexible-Length Paradigm*, ensuring that our benchmark targets scenarios where length control is meaningful. A detailed mapping of NLG tasks to these paradigms is provided in Appendix B.1. Among them, we focus on tasks within the *Flexible-Length Paradigm*, as they are the most suitable for evaluating length instruction following in LLMs.

We categorize tasks within *Flexible-Length Paradigms* into four categories and collect 360 fundamental data units from 21 datasets in English and Chinese. Figure 2 illustrates the distribution of data categories and text length in LENGTHBENCH. We introduce the data collection and refinement processes for these four categories as follows, details are provided in Appendix B.2 and B.3.

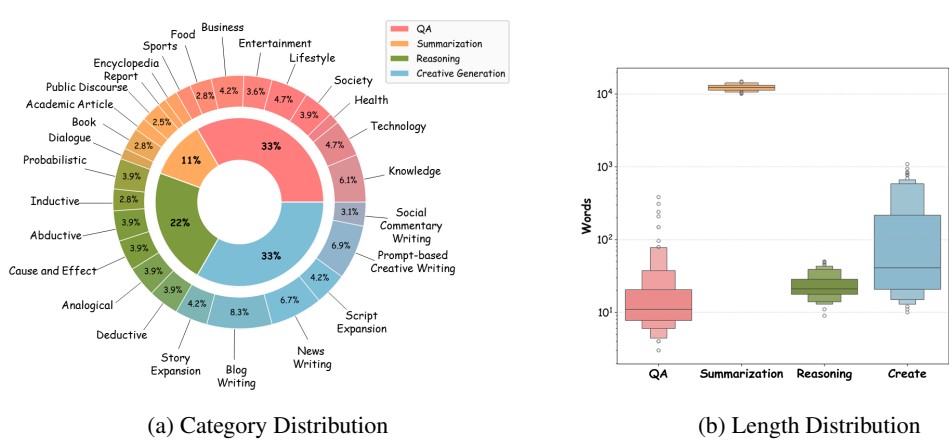

| (a) Category Distribution | (b) Length Distribution |

Figure 2: (a) presents the proportion of four task categories in LIFEBENCH in the inner ring, while the outer ring further details the subtypes within each category and their respective proportions in the dataset. (b) illustrates the length distribution of the four categories in LIFEBENCH, where the vertical axis represents text length (measured in words for English and characters for Chinese) on a logarithmic scale, and the horizontal axis denotes different task categories.

- **Question Answering (QA) Task:** The QA task requires the model to generate answers based on given questions. The answer length can vary flexibly depending on the level of detail required. We collect questions from six representative QA datasets [53, 1, 54, 119, 74], which cover nine different domains. In total, we select 120 questions by filtering for open-ended, well-formed examples that allow for responses of varying length, excluding factoid, binary, and duplicate questions, with an equal split of 60 in Chinese and 60 in English.

- **Summarization Task:** The Summarization task requires the model to condense long-text into a concise summary. A summary extracts key points or includes more supporting details, depending on the requirements. We collect data from seven summarization datasets [21, 55, 37, 121, 32, 74], spanning seven types of summarization tasks. To ensure the task suits longer length constraints, we select input samples ranging from 10,000 to 15,000 words in length. Furthermore, we manually filter out content containing excessive tables, numerical data, or irregular formatting to improve information density. In total, we select 40 samples, with 20 in Chinese and 20 in English.

- **Reasoning Task:** The Reasoning task requires the model to perform logical inference based on given facts or background information. The output can either be a final conclusion or a step-by-step

reasoning process with detailed justification. Existing reasoning datasets (*e.g.*, GSM8K [20], CommonsenseQA [96]) are not suited for flexible-length generation due to their rigid structure and limited support for open-ended responses. To address this, we follow the prior reasoning categories [44] and generate new reasoning problems using GPT-4o. These problems cover a range of reasoning depths, enabling better support for evaluating the length instructions following. We curate 80 reasoning problems, half Chinese and half English.

- **Creative Generation Task:** The Creative Generation task requires the model to produce creative text, such as academic papers, novels, and technical reports. The complexity and richness of the content can be adjusted to accommodate different length constraints. We collect data from seven datasets [82, 75, 31, 42, 51] and categorize the samples into six generation types based on their textual characteristics. In total, we select 120 samples by filtering out instances with explicit length or structural constraints, as well as those with ambiguous instructions or duplicated content, 50% Chinese and 50% English.

Finally, to ensure consistency and suitability for length instructions, we refine or formalize the data. Specifically, we design refinement templates tailored to each subtype. Each template includes an `Instruction` specifying the task type and original input, and a `Requirement` imposing the length constraint and control method. The resulting refined dataset constitutes the final benchmark data for LIFEBENCH. Representative examples of the four tasks above are provided in Appendix B.4.

## 4 Experimental Setup

### 4.1 Models

We evaluate over 26 powerful LLMs on LIFEBENCH, including nine proprietary models, eight open-source models, and three models enhanced for long-text generation. To ensure consistency in our experimental setup, we set the maximum output size to 65,536 tokens for all models, or to the maximum supported size if smaller. Additionally, we set the temperature to 0.8 for non-reasoning models, and configure reasoning models with a medium reasoning effort if this parameter is supported. See the Appendix D.1 for a complete model list and all configurations.

### 4.2 Length Constraints

One of the core principles of LIFEBENCH is the provision of extensive length constraints, achieved through three control methods and ten constraint levels. The three control methods are: *Equal To*, *At Most*, and *At Least*. *Equal To* requires the output length to match the length constraints exactly. *At Most* ensures that the output does not exceed the constraint, while *At Least* guarantees that the output is no shorter than the constraint. Building on the control methods, we define ten distinct length constraints: 16, 32, 64, 128, 256, 512, 1024, 2048, 4096, and 8192. For **English data**, the length is measured in words, while for **Chinese data**, it is measured in characters. The minimum constraint of 16 ensures that models generate complete responses, while the maximum constraint is set to 8192, which is equivalent to the number of words for the smallest maximum output length among all evaluated models. By applying all control methods and length constraints to 360 fundamental data units, we obtain a total of 10,800 instances.

### 4.3 Evaluation Metric

LIFEBENCH employs two metrics to analytically evaluate a model's length instruction following: **Length Deviation** and **Length Score**.

**Length Deviation.** *Length Deviation (LD)* measures the proportionate difference between the model's output length and the specified length constraint. This metric provides an intuitive assessment of the model's length instruction following, capturing both the direction (over- or under-generation) and the magnitude of deviation. Formally, it is defined as:

$$LD = \frac{L_{\text{output}} - L_{\text{constraint}}}{L_{\text{constraint}}}, \tag{1}$$

where $L_{\text{constraint}}$ denotes the imposed length constraint, and $L_{\text{output}}$ is the word count of the output.

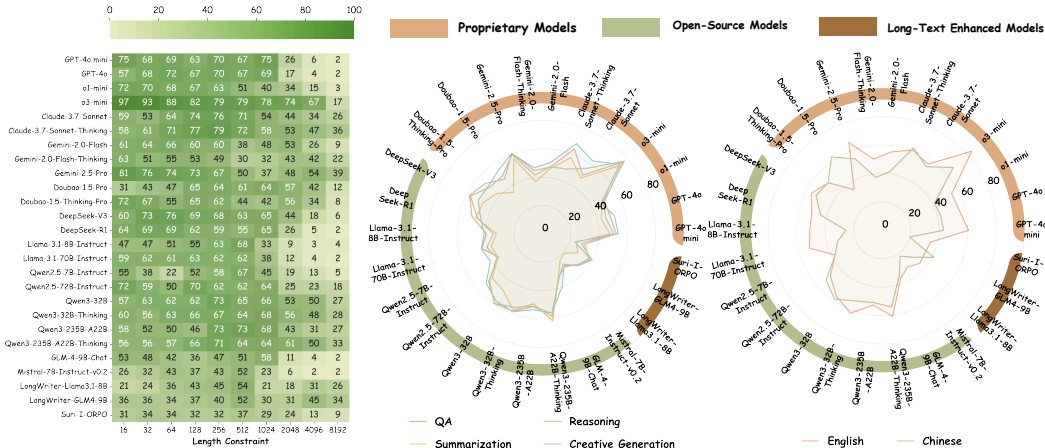

Figure 3: *Left. Length Score* across all length constraints under the *Equal To* control method. Impact of task type (***Middle***) and language (***Right***) on the *Length Score*, separated by model type.

**Length Score.** The effectiveness of *Length Deviation* may be degraded when aggregating multiple samples, as positive and negative deviations can offset each other. Furthermore, *Length Deviation* is sensitive to outliers, where a few extreme values can disproportionately skew the aggregated results. To address these limitations, we introduce *Length Score (LS)*, which maps *Length Deviation* to a bounded score, eliminating both the offsetting effect of positive and negative deviations and the excessive influence of outliers, thus enabling a more reliable evaluation. Specifically, for the *Equal To* control method, *LS* is defined as:

$$LS = \begin{cases} 100 \times e^{k_1 \cdot LD}, & \text{if } LD < 0 \\ 100 \times e^{-k_2 \cdot LD}, & \text{if } LD \geq 0, \end{cases} \tag{2}$$

where $k_1 = 5$ and $k_2 = 2$, set empirically. This formulation ensures slower score decay for over-generation ($LD \geq 0$), considering that exceeding the constraint is generally more acceptable, because it can be deleted, than under-generation. The scoring curve is steeper for small $|LD|$, allowing finer discrimination of subtle deviations; for larger $|LD|$, the decay moderates to avoid excessively penalizing poor cases. The score approaches zero as $LD$ approaches $-1$ or $+\infty$, indicating complete failure in following the length instructions. For *At Most* and *At Least* control method, analogous scoring functions are defined according to their respective evaluation criteria. Full details are provided in Appendix D.2.

## 5 Results and Analysis

In this section, we present evaluation results and analysis of LLMs' ability to follow length instructions. In Section 5.1, we present evaluation results of current LLMs on LIFEBENCH, and find that they exhibit widespread difficulties with following length instructions, particularly under long length constraints. Due to these deficiencies under long length constraints, in Section 5.2 we further investigate whether models can reliably achieve their claimed maximum output length. To gain deeper insight into the underlying causes of these limitations, in Section 5.3, we conduct a detailed diagnosis through a series of extended experiments, analyzing key sources of failure and characterizing model performance under varied conditions.

### 5.1 Main Results

**Evaluation Protocol.** Following the evaluation protocol described in Section 4, we evaluate all 26 models on LIFEBENCH, excluding approximately 0.1% of instances with empty outputs per model from the final results. Comprehensive results are provided in Appendix E.

Table 1 summarizes the overall performance of various LLMs on LIFEBENCH. Most models exhibit poor length instruction following under the *Equal To* control method: 23 out of 26 models score below 60, with o3-mini, Claude-Sonnet-Thinking, and Gemini-2.5-Pro achieving 75.4, 61.3,

Table 1: We compute the mean *LS* for each model, averaging over all length constraints, to assess model performance under the three control methods. For the *Equal To* control method, we additionally report *LD*, computed as the mean absolute deviation across all length constraints.

| Models | Params | Reasoning Model | Equal To | | At Most | At Least |
|---|---|---|---|---|---|---|
| | | | LD↓ | LS↑ | LS↑ | LS↑ |
| *Proprietary Models* | | | | | | |
| GPT-4o mini | - | ✗ | 31% | 51.9 | 90.3 | 74.0 |
| GPT-4o | - | ✗ | 31% | 49.1 | 97.0 | 72.6 |
| o1-mini | - | ✓ | 35% | 48.3 | 89.7 | 81.5 |
| o3-mini | - | ✓ | 13% | **75.4** | **99.5** | **97.0** |
| Claude-3.7-Sonnet | - | ✗ | 30% | 55.4 | 96.8 | 90.1 |
| Claude-3.7-Sonnet-Thinking | - | ✓ | 33% | **61.3** | 96.5 | 93.3 |
| Gemini-2.0-Flash | - | ✗ | 36% | 48.4 | 95.2 | 84.7 |
| Gemini-2.0-Flash-Thinking | - | ✓ | 53% | 44.0 | 90.2 | 91.2 |
| Gemini-2.5-Pro | - | ✓ | 28% | 60.0 | 96.1 | **95.5** |
| Doubao-1.5-Pro | - | ✗ | 23% | 48.7 | **99.9** | 89.3 |
| Doubao-1.5-Thinking-Pro | - | ✓ | 29% | 50.6 | 97.8 | 85.7 |
| *Open-Source Models* | | | | | | |
| DeepSeek-V3 | 671B | ✗ | 27% | 54.3 | 96.7 | 79.7 |
| DeepSeek-R1 | 671B | ✓ | 36% | 47.7 | 93.8 | 74.1 |
| Llama-3.1-8B-Instruct | 8B | ✗ | 70% | 38.1 | 82.3 | 71.8 |
| Llama-3.1-70B-Instruct | 8B | ✗ | 61% | 42.4 | 88.8 | 69.0 |
| Qwen2.5-7B-Instruct | 7B | ✗ | 36% | 37.3 | 97.6 | 71 |
| Qwen2.5-72B-Instruct | 7B | ✗ | 28% | 50.6 | 93.8 | 84.2 |
| Qwen3-32B | 32B | ✗ | 19% | 57.6 | 97.2 | 87.4 |
| Qwen3-32B-Thinking | 32B | ✓ | 23% | 57.8 | 93.0 | 87.8 |
| Qwen3-235B-A22B | 235B | ✗ | 22% | 52.1 | 95.6 | 90.3 |
| Qwen3-235B-A22B-Thinking | 235B | ✓ | 23% | 57.7 | 89.9 | 89.6 |
| GLM-4-9B-Chat | 9B | ✗ | 40% | 35.2 | 95.9 | 68.3 |
| Mistral-7B-Instruct-v0.2 | 7B | ✗ | 84% | 26.7 | 85.9 | 63.0 |
| *Long-Text Enhanced Models* | | | | | | |
| LongWriter-Llama3.1-8B | 8B | ✗ | 102% | 31.9 | 84.1 | 79.0 |
| LongWriter-GLM4-9B-Chat | 9B | ✗ | 52% | 37.4 | 90.9 | 86.8 |
| Suri-I-ORPO | 7B | ✗ | 506% | 27.4 | 79.7 | 75.2 |

and 60.0, respectively. Performance improves substantially under the *At Most* and *At Least* control methods, where 19 and 6 models, respectively, surpass a score of 90, due to the looser constraints.

Figure 3 provides a comprehensive analysis of model performance across different length constraints and input characteristics, including task type and language. Model performance varies substantially across length constraints. Under all short constraints (≤128 words), `o3-mini` and `Gemini-2.5-pro` consistently perform strongly, with scores above 80 and 70, respectively, while 8 out of 26 models score below 60. For medium constraints (256–2048 words), `o3-mini` remains robust (>70), but `Gemini-2.5-pro` drops sharply (81 → 37). Under long constraints (≥4096 words), no models consistently exceed a score of 40. Notably, `LongWriter-Llama3.1-8B` and `LongWriter-GLM4-9B` demonstrate notable improvements over their respective base models, with score increases of 22 and 32 points, respectively, on the longest constraints (*i.e.*, 8192 words).

Regarding input characteristics, Summarization tasks yield the lowest *Length Score* for 19 out of 26 models, while Creative Generation tasks achieve the highest *Length Score* for 14 models, indicating substantial task-dependent variation. In terms of language, nearly all models perform worse on Chinese compared to English. Notably, in Appendix G, we observe a consistent tendency for most models to over-generate when instructions are given in Chinese, highlighting a potential language-specific bias.

**Takeaways.** Explicit length instruction following remains a significant challenge for current LLMs, with performance strongly influenced by control method, length constraint, task type, and language.

## 5.2 Discrepancies Between Claimed and Actual Maximum Output Length

**Evaluation Protocol.** Given the observed deficiencies of most models under longer length constraints, we are motivated to examine their behavior in even more challenging settings. Specifically, we investigate whether the models are able to achieve the maximum output length claimed by the

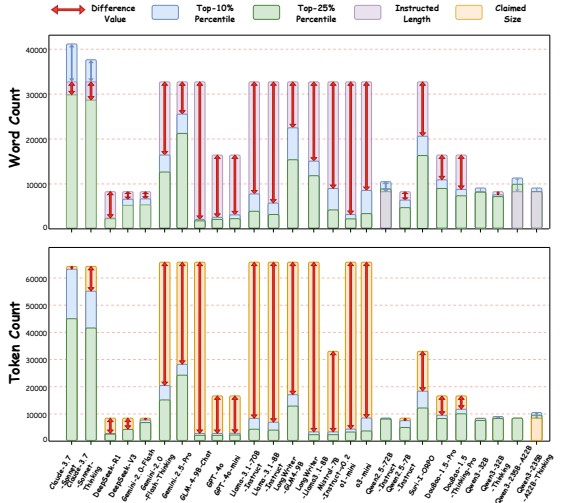
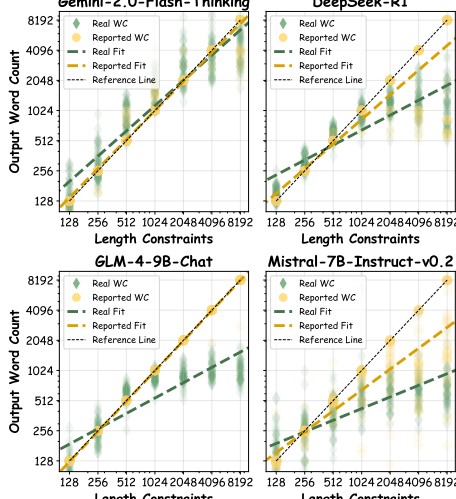

Figure 4: **(Top)** Comparison of the specified length constraints (in words) with the actual word counts of the top 10% and top 25% longest outputs for each model. **(Bottom)** Comparison of the claimed maximum output size (in tokens) with the actual token counts of the top 10% and top 25% longest outputs for each model. Token counts are derived from word statistics.

Figure 5: Length Awareness Experiment: The real output word count and self-reported word count are shown for cases where the length constraint exceeds 128, with both axes in log scale. "Real Fit" and "Reported Fit" represent the least squares regression results for real and self-reported word counts.

providers under extreme length constraints. We set the control method to *At Least* and specified a length constraint of 32,768 words. For models whose claimed maximum output length is less than 32,768 tokens, we set the word-based constraint to approximate each model's maximum token limit, ensuring that the instruction requests outputs up to the model's capacity. We exclude summarization tasks from consideration as they do not provide sufficiently long inputs for meaningful evaluation under such constraints. The final dataset comprises 320 instances, with all outputs containing repeated content manually filtered to ensure validity. Further results on extended length constraints are provided in Appendix E.

Figure 4 compares each model's claimed maximum output length with the actual maximum output length achieved. Regarding word count, among the 26 models evaluated, only the `Claude` and `Qwen` series (seven models in total) consistently meet the length constraint in their top 10% longest outputs. In the top 25% longest outputs, however, only `Qwen2.5-72B-Instruct` and `Qwen3-235B-A22B` satisfy the constraint. In terms of tokens, among all models that failed to meet the length constraints, only `Gemini-2.0-Flash` and the `Qwen` series were limited by their relatively small maximum output length. All other models fell significantly short of their respective maximums, indicating that their inability to satisfy the length constraints stems from inherent generation limitations rather than explicit length restrictions.

**Takeaways.** There exists a substantial discrepancy between vendor-claimed and actual model performance under extreme long length constraints, highlighting the need for more reliable evaluation and reporting of maximum output capabilities.

### 5.3 Diagnosis of Length Instruction Following Failures

To further investigate the causes underlying poor length instruction following, we conduct a series of distinct but inter-connected analyses.

**Length Awareness Deficits.** Given widespread failures in following length instructions, we seek to understand these deficiencies from the perspective of the models' internal awareness of output length. To this end, we design the *length awareness experiment* in which each model is prompted to append a self-reported word count to its generated text. We then compare the self-reported counts to their actual output lengths. Figure 5 presents four representative examples, models overshoot short length limits while underestimating length, and undershoot long limits while overestimating length. Notably,

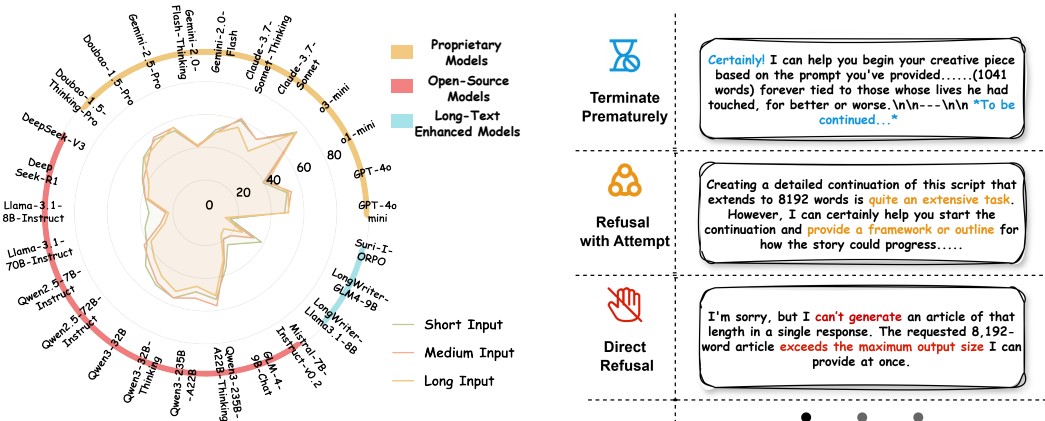

Figure 6: Impact of long input on the *Length Score*.

Figure 7: Three representative lazy strategies.

`GLM-4-9B-Chat` reports word counts that match the instructions, even when its actual outputs do not, suggesting that it assumes compliance rather than measuring the true output length. Experimental details and comprehensive results for all models are provided in Appendix F.

**Sensitivity to Input Length.** Section 5.2 reveals that summarization tasks yield the lowest performance. Since inputs of such tasks are typically much longer than others, we further assess the impact of input length on model's length instruction following. Specifically, we select a subset of samples from the summarization task and construct three input versions for each by truncating the original text to short (<1000 words), medium (1000–5000 words), and long (>5000 words) lengths. We then prompt the models to generate a continuation for each version. As detailed in Figure 6, among the 26 models evaluated, 21 models show marked declines in *Length Score* with increasing input length, with `LongWriter-Llama3.1-8B` exhibiting the largest *Length Deviation* reduction (↓ 13.6). These models also perform worst on summarization tasks, revealing a strong association between input length sensitivity and poor summarization fidelity. Detailed experimental settings and results are provided in Appendix G.2.

**Prevalence of Lazy Generation Strategies.** Through analysis of the generated outputs, we observe two distinct failure modes across all models: in some cases, models are fundamentally limited in producing long-form content, while in others, *lazy strategies* are adopted to circumvent length constraints, such as prematurely terminating the response or outright refusing to generate content. Figure 7 illustrates three representative types of *lazy strategies*. Notably, we observe that the prevalence of such strategies increases sharply when length constraints exceed 4096 words, and on average surpasses 10% for all models when the constraint is set to 8192 words. More details and quantitative analysis are provided in the Appendix H. Interestingly, as shown in Appendix M.2, our experiments on the base model reveal that the refusal *lazy strategies* is not only attributable to safety alignment, but also arises from pre-training.

**Limitations of Intermediate Reasoning.** Given the mechanistic differences between reasoning and standard models, we conduct a case study on `Claude-3.7-Sonnet-Thinking` under three representative length constraints: 16, 1024, and 8192 words. By analyzing intermediate reasoning traces, we observe that reasoning models can calibrate their output length during the reasoning process. Specifically, the model first generates a draft response and performs self-evaluation by counting the number of words in its reasoning process. If the generated length does not satisfy the instruction, the model iteratively revises or regenerates the response to better follow the specified constraint. However, this ability to dynamically adjust output length only partially alleviates the problem for short length constraints and still fails under longer constraints. Further illustrative examples and detailed analyses are provided in Appendix I.

**Takeaways.** Deficits in length awareness, sensitivity to input length, and the prevalence of lazy generation strategies collectively undermine effective length instruction following in current models. Although reasoning models attempt to calibrate output length during the intermediate reasoning process, this approach remains ineffective under longer length constraints.

# 6   Conclusion, Limitations and Future Insights

In this paper, we introduce LIFEBENCH, a comprehensive benchmark for evaluating the ability of LLMs to follow length instructions across diverse tasks, languages, and a broad of length constraints. Empirical analysis shows that current LLMs are generally unreliable at following length instructions, particularly under long constraints, often falling significantly short of their vendor-claimed length. We further showcase that model performance is substantially affected by input characteristics, including task type, language, and input length, *etc*. These findings reveal a critical gap in LLMs' ability to follow length instructions, highlighting the need for more targeted evaluation and development to improve instruction following in future models. We acknowledge that this work focuses on benchmarking, and it does not offer elegant solutions to the underlying causes, while Appendix M provides several promising insights. Addressing these causes is an important direction for future research. We hope LIFEBENCH and our analyses will facilitate further research in this important but underexplored area.

## Acknowledgments and Disclosure of Funding

This work was supported by the National Natural Science Foundation of China (Grant No. 62072052) and the National Key Research and Development Program of China (Grant No. 2024YFF0907401).

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

# Appendices

# Appendix Table of Contents

# A   Comparison of LIFEBENCH with Existing Benchmarks

In this section, we review ten representative benchmarks on long-form text generation or long-context modeling and analyze the evaluation ranges they cover. Based on this analysis, we position our work, LIFEBENCH, relative to prior benchmarks, highlighting its unique strengths and comprehensive evaluation design.

Specifically, LIFEBENCH differentiates itself by being bilingual (English and Chinese), covering four task categories, and comprising 360 fundamental data units. Each unit derives 30 data items across three length control methods and ten distinct length constraints, resulting in a total of 10,800 instances. To facilitate multi-dimensional analysis of length instruction following, we introduce two dedicated evaluation metrics: *Length Deviation* and *Length Score*. Thus, our benchmark features a **Diverse Dataset**, **Extensive Length Constraints**, and **Analytical Evaluation**, providing broader coverage and more fine-grained analysis compared to prior benchmarks. We summarize key differences between LIFEBENCH and prior benchmarks in Table 2 and provide detailed discussions below.

Table 2: Comparison of LENGTHBENCH with Other LLM Benchmarks.

| Benchmarks | Diverse Task | Long Input | Bilingual | Multiple Control Methods | Length Constraints | | | Analytical Evaluation |
|---|---|---|---|---|---|---|---|---|
| | | | | | <100 | 100-2000 | >2000 | |
| IFEval [122] | ✓ | ✗ | ✗ | ✓ | ✓ | ✓ | ✗ | ✗ |
| AlpacaEval-LI [114] | ✗ | ✗ | ✗ | ✗ | ✓ | ✓ | ✗ | ✓ |
| Ruler [58] | ✗ | ✗ | ✗ | ✗ | ✓ | ✓ | ✗ | ✓ |
| Suri [83] | ✗ | ✗ | ✗ | ✗ | ✗ | ✓ | ✓ | ✗ |
| LongBench-Write [8] | ✗ | ✗ | ✓ | ✗ | ✗ | ✓ | ✓ | ✓ |
| HelloBench [86] | ✓ | ✓ | ✗ | ✗ | ✗ | ✗ | ✓ | ✗ |
| LongProc [112] | ✗ | ✓ | ✗ | ✗ | ✗ | ✓ | ✓ | ✗ |
| LonGen Bench [85] | ✗ | ✗ | ✓ | ✓ | ✗ | ✗ | ✓ | ✓ |
| LongGenBench-1 [106] | ✗ | ✗ | ✗ | ✗ | ✗ | ✗ | ✓ | ✗ |
| LongGenBench-2 [69] | ✗ | ✓ | ✗ | ✗ | ✗ | ✗ | ✗ | ✗ |
| **Ours** | ✓ | ✓ | ✓ | ✓ | ✓ | ✓ | ✓ | ✓ |
| | **Diverse Dataset** | | | **Extensive Length Constraints** | | | | **Analytical Evaluation** |

We categorize existing benchmarks into two main groups for comparison: benchmarks explicitly targeting length instruction following, and those focused on general long-text generation.

**Length Instruction Following Benchmarks.** *IFEval* is a well-established benchmark for instruction-following evaluation, utilizing length constraints as proxies for assessing general adherence to instructions. It includes a variety of tasks and control methods to measure a model's ability to follow instructions. However, *IFEval* primarily focuses on short-text scenarios, without input or output instructions exceeding 2,000 words. Furthermore, as its primary aim is to evaluate general instruction following, it lacks detailed analytical evaluation of length control, making it difficult to directly assess a model's performance on length instruction following. In addition, several benchmarks have been specifically designed to evaluate length instruction following, such as *AlpacaEval-LI* and *Ruler*. Unlike traditional instruction following benchmarks, these datasets focus exclusively on length control. However, they tend to exhibit narrower scopes, typically constrained to short outputs. These benchmarks often lack sufficient diversity in task domains and control methods, limiting their effectiveness for comprehensive length-adherence evaluation.

**Long-Text Generation Benchmarks.** Benchmarks in this category primarily assess generation quality within long-form content scenarios, typically including explicit length instructions as part of their evaluation setup. We contrast LIFEBENCH with seven representative benchmarks in this area. Existing long-text benchmarks generally prioritize generation quality in extended contexts but neglect comprehensive evaluations across shorter lengths or varied control methodologies. Additionally, apart from *LongBench-Write* and *LonGen Bench*, most benchmarks in this group lack analytical evaluation methods explicitly targeting length instruction following. Combined with our findings revealed in this paper, it is difficult for models to achieve excellent performance in long-text generation. We believe that LIFEBENCH makes an important supplement to the previous work that may have overlooked the important factor of actual generation length.

In addition to the above two categories, we also compare LIFEBENCH with several benchmarks in the controlled text generation domain. Existing controlled generation benchmarks can be broadly

categorized into content control and attribute control [62]. Notably, structure control, a subcategory of content control—often incorporates length constraints as part of its evaluation, exemplified by benchmarks such as *COLLIE* [111], *CoDI-Eval* [17], and *CFBench* [117]. However, these benchmarks typically involve multiple and complex control factors and rarely explore extended text-generation scenarios. Consequently, they are less suited for a dedicated and systematic analysis of length instruction adherence in isolation.

In summary, compared to previous benchmarks, LIFEBENCH provides a comprehensive benchmark specifically designed to evaluate length instruction following. And we provide the most languages, the most comprehensive coverage of length constraints, a broad and diverse set of tasks, and multiple control methods—effectively addressing key limitations found in existing benchmarks.

# B Details of LIFEBENCH

## B.1 Mapping of NLP Tasks to Length Paradigms

In Section 3, we present a task-to-paradigm mapping that categorizes standard NLP tasks according to the four *Length Paradigms* defined earlier: *Length-Independent Paradigm*, *Fixed-Length Paradigm*, *Adaptive-Length Paradigm*, and *Flexible-Length Paradigm*. In this section, we elaborate on these four paradigms:

- *Length-Independent Paradigm*: This paradigm comprises tasks where length constraints are inherently irrelevant to the task objective. Imposing a length restriction does not impact the fundamental goal or the correctness of the output. Such tasks typically involve structured outputs where specifying a fixed length is unnecessary. Representative examples include tokenization and part-of-speech (POS) tagging, which prioritize functional correctness over word or character counts, so they are length-independent.

- *Fixed-Length Paradigm*: This paradigm includes tasks where the output length remains approximately constant, irrespective of input variations. These tasks require a predefined output length that cannot be freely adjusted. Representative examples include text classification and sentiment analysis, where outputs are typically restricted to a single category or a small set of alternatives.

- *Adaptive-Length Paradigm*: This paradigm covers tasks where the output length naturally adapts to the input length, with longer inputs generally producing longer outputs. Representative examples include machine translation and text style transfer. For instance, longer input sentences in machine translation generally yield longer outputs, and the same applies to text style transfer.

- *Flexible-Length Paradigm*: This paradigm encompasses tasks where the output length can be freely adjusted across a wide range, supporting both concise and detailed responses. A defining characteristic of these tasks is their adaptability to different use requirements. Representative examples include open-ended question answering and text completion. In open-ended question answering, models can extend answers with additional explanations, while in text writing, they can generate outputs ranging from short passages to extensive works spanning tens of thousands or even millions of words.

Given the diversity of NLP tasks, we focus on representative examples for each paradigm, accompanied by concise descriptions of their primary objectives. Table 3 summarizes the mapping of NLP tasks to the four length paradigms.

Table 3: Mapping of common NLP tasks to the Length Paradigms.

| Length Paradigm | Representative NLP Tasks | Task Description |
|---|---|---|
| *Length-Independent* | Tokenization [91] | Splits text into tokens based on linguistic or subword units. |
| | POS Tagging [90] | Assigns part-of-speech labels to each token in the input sequence. |
| | Named Entity Recognition [57] | Identifies and labels named entities (e.g., people, locations) in a sentence. |
| | Dependency Parsing [16] | Analyzes syntactic structure by establishing relationships between words. |
| | Text Matching [45] | Measures semantic or lexical similarity between two pieces of text. |
| *Fixed-Length* | Text Classification [22] | Assigns a label (e.g., topic, category) to an entire input text. |
| | Sentiment Analysis [93] | Predicts sentiment polarity (e.g., positive, negative) of a given text. |
| | Spam Detection [25] | Classifies an email or message as either spam or not spam. |
| | Toxic Content Detection [24] | Detects the presence of toxic, offensive, or harmful language in text. |
| | Title Generation [76] | Generates a concise title for a given passage or document. |
| | Stereotype Detection | Identifies biased or stereotypical language in text. |
| *Adaptive-Length* | Machine Translation [95] | Converts text from one language to another. |
| | Text Style Transfer [49] | Alters text style (e.g., formal to informal) while keeping content intact. |
| | Paraphrase Generation [61] | Rewrites input text using different wording while preserving meaning. |
| | Question Rewriting [26] | Reformulates a question while preserving its original intent. |
| | Sentence Perturbation [2] | Modifies sentence form slightly while retaining its meaning. |
| *Flexible-Length* | Open-ended Question Answering [15] | Generates free-form answers to questions without fixed format. |
| | Text Summarization [89] | Extracts and rephrases key content from the input text. |
| | Sentence Compression [89] | Compresses a sentence into a shorter version while preserving key meaning. |
| | Text Completion [87] | Generates plausible continuations for a given text. |
| | Dialogue Generation [9] | Produces context-aware responses in multi-turn conversations. |
| | Story Composition [31] | Generates narratives or stories based on input prompts. |
| | Poem Generation [118] | Creates poems in various styles and forms from given input. |
| | Sentence Expansion [87] | Extends a short sentence into a more detailed or informative one. |
| | Reasoning [102] | Performs complex inference or multi-step reasoning. |

## B.2 Data Collection

This section provides a detailed description of the data collection process under the *Flexible-Length Paradigm* and outlines the taxonomy of tasks included within this setting. Based on Table 3, and considering the variability in output length requirements, we select four representative categories from the *Flexible-Length Paradigms* as our base: **Question Answering (QA)**, **Summarization**, **Reasoning**, and **Creative Generation**. These categories support variable-length instructions and are well suited for standardized evaluation.

For each task category, we first define the task and explain its relevance to the *Flexible-Length* setting. We then introduce the associated representative NLP tasks, followed by a comprehensive overview of the data collection process, including dataset composition and provenance, preprocessing and data cleaning strategies, and the subtypes represented within each category.

**Question Answering (QA) Task.** This task involves answering open-ended questions whose answer length varies according to the required detail, aligning naturally with the *Flexible-Length Paradigm*. The representative NLP task for this category is **Open-ended Question Answering**.

To support our analysis of this task under the flexible-length setting, we manually collected a total of **120** QA samples from publicly available datasets, comprising **60** Chinese and **60** English examples. The selection criteria are: (1) Questions must be open-ended, allowing responses ranging from concise to elaborative explanations. When the length constraint is small, such as 16, the model can just give the answer, while for longer answers it can provide explanations and even add richness while giving the answer; (2) Questions should be meaningful, context-rich, and well-formed, enabling various depths of responses. For example, an open-ended historical or political discussion question, *who is your favorite American president?* Under this question, the model can answer the term and name of a specific president, or it can explain the reasons from multiple perspectives to form a deep answer; (3) We excluded factoid, binary, multiple-choice, duplicate, or grammatically incorrect entries. The answer space for such questions is usually too narrow, and sometimes it is difficult to form a longer answer. For example, *what is the answer to 3+5?* For this question, it is difficult for the model to generate a 2000-word answer to explain the question.

To ensure linguistic and topical diversity, we sourced English QA data equally (**15** questions each) from four datasets:

- **Question Pairs Dataset[53]**: Contains over 400k real-world questions from Quora[1].
- **Stack Exchange Dataset [1]**: Comprises over 80k community-driven QA threads from the Stack Exchange[2].
- **WikiHow Dataset [54]**: Includes over 160k how-to questions and procedural content from WikiHow[3].
- **Yahoo Answers Dataset [119]**: Consists of 10 different categories and each class contains over 140k QA pairs.

The Chinese QA data were collected from the following two datasets:

- **QA Wikihow [74]**: A Chinese QA dataset derived from WikiHow, from which we selected **25** open-ended questions.
- **QA Zhihu [74]**: Sourced from Zhihu[4], a Chinese community QA platform, from which we selected **35** diverse and open-ended questions.

We then analyzed the topical distribution of the collected QA samples, categorizing questions into nine thematic domains: *Knowledge* (*e.g. How does the finance credit score work?*), *Technology* (*e.g. How to protect data from internal data scientists?*), *Health* (*e.g. How do I lose weight without doing exercise?*), *Society* (*e.g. How can we improve the education system for high school and college?*), *Lifestyle* (*e.g. How to be positive?*), *Entertainment* (*e.g. How to play competitive Counter-Strike*

---

[1] https://www.quora.com/
[2] https://stackexchange.com/
[3] https://www.wikihow.com/
[4] https://www.zhihu.com/

*Global Offensive?*), *Business* (*e.g. How to start investing?*), *Food* (*e.g. How to brew commercial beer?*), and *Sports* (*e.g. How to play basketball?*). These categories comprehensively cover typical domains encountered in open-ended question design.

**Summarization Task.** The objective of summarization task is to condense long-form texts into shorter summaries whose lengths can vary considerably—from concise overviews to more detailed condensations—aligning naturally with the *Flexible-Length Paradigm*. Representative NLP tasks include **Text Summarization** and **Sentence Compression**, both aiming to preserve key information while reducing text length.

To support our analysis of this task under the flexible-length setting, we selected a total of **40** long-form source documents from open-source datasets, with an equal split of **20** English and **20** Chinese texts. Given the need for variable-length summaries, we intentionally focused on documents with substantial content, ensuring sufficient information is available for both brief and detailed summarization. Specifically, we selected source texts within the range of **10,000 to 15,000 words** (or characters for Chinese), striking a balance between the context window limitations of current language models and the need to support diverse summarization lengths in the *Flexible-Length Paradigm*. To improve text quality and informativeness, we manually cleaned the documents by removing low-informative elements such as extensive tables, blocks of numeric-only content, meta descriptions, and other low-information or structurally noisy parts.

The English summarization data were drawn from the following five datasets, with **4** documents each:

- **Scientific Papers [21]**: Structured academic documents sourced from ArXiv[5] and PubMed[6]. Text type: *Academic Article*.
- **BookSum [55]**: Long-form literary narratives. Text type: *Book*.
- **Gov Report [37]**: Government reports from the Congressional Research Service[7] and U.S. Government Accountability Office[8]. Text type: *Report*.
- **QMSUM [121]**: Multi-domain meeting transcripts. Text type: *Dialogue*.
- **Wikipedia [32]**: High-quality encyclopedia-style articles from cleaned English Wikipedia dumps. Text type: *Encyclopedia*.

The Chinese summarization data were collected from the following **three** datasets:

- **Co Ann Report [74]**: A dataset of Chinese corporate annual reports. The corresponding text type is *Report*, and we selected **5** documents.
- **Wikipedia [32]**: A cleaned dump of Chinese Wikipedia articles. The corresponding text type is *Encyclopedia*, and we selected **5** documents.
- **Gov XueXiQiangGuo [74]**: A collection of public discourse materials from the Xuexi Qiangguo platform[9], including political speeches, news reports, and commentary. The corresponding text type is *Public Discourse*, and we selected **10** documents.

**Reasoning Task.** Reasoning is a core capability distinguishing modern LLMs from traditional NLP systems. Unlike classification or span extraction tasks, which typically rely on surface-level textual patterns, reasoning task requires multi-step logical inference and contextual understanding, making it uniquely aligned with LLM capabilities rather than conventional NLP pipelines.

Compared to general QA task, reasoning task emphasizes not only the final answer but also the underlying logical steps toward the conclusion. Whereas open-ended QA may occasionally require explanations, it does not necessarily involve inference over structured facts or scenarios. In contrast, reasoning task explicitly demands structured cognitive processes—such as deduction, analogy, or causal inference—often necessitating step-by-step justification. Thus, the representative NLP task for this category is simply **Reasoning**.

---

[5] https://arxiv.org/
[6] https://pubmed.ncbi.nlm.nih.gov/
[7] https://crsreports.congress.gov
[8] https://www.gao.gov/
[9] https://www.xuexi.cn/

A unique challenge emerged when adapting reasoning tasks to the flexible-length setting. Existing datasets for reasoning—such as GSM8K [20] (mathematical reasoning) or CommonsenseQA [96] (commonsense inference)—are typically not well-suited for flexible-length evaluation. The former's mathematical notations and equations that make it difficult to control output length meaningfully, while the latter is designed as a multiple-choice task, constraining the response format and limiting the scope for open-ended explanations. Nonetheless, reasoning itself inherently suits the *Flexible-Length Paradigms*: some inferences can be expressed succinctly, while others benefit from elaborate justifications. This makes it ideal for investigating the ability of models' length instruction following based on complexity.

To obtain high-quality, flexible-length reasoning samples, we employ GPT-4o [78] to generate **80** open-ended reasoning questions—**40** in Chinese and **40** semantically equivalent questions in English. Prompts are designed to allow concise responses or detailed justifications based on model instruction and complexity.

We follow the taxonomy proposed by Huang et al. [44], covering six distinct categories: *Deductive* (*e.g. Assuming that all sexually dimorphic animals possess reproductive organs, analyze whether possessing reproductive organs can conversely indicate that an animal is sexually dimorphic.*), *Inductive* (*e.g. Based on historical data from successive generations of smartphones, predict the direction of innovation in the next generation.*), *Abductive* (*e.g. Determine the most plausible explanation for this observation: A country's currency experiences abnormal exchange rate fluctuations.*), *Analogical* (*e.g. How can the spread of computer viruses be compared to the spread of biological viruses?*), *Cause and Effect* (*e.g. Examine the causal links between high temperatures and urban power supply shortages*), and *Probabilistic* (*e.g. Evaluate the probabilistic models used to assess risk diversification in financial investment portfolios.*). The dataset consists of **10** Deductive reasoning items and **14** items each for the other five categories, ensuring balanced representation. All items underwent rigorous manual review according to four quality criteria: (1) logical soundness and necessity of reasoning rather than factual recall; (2) clarity and absence of ambiguity or misleading premises; (3) support for variable elaboration levels, from brief conclusions to detailed explanations; and (4) cultural and linguistic appropriateness. This meticulous curation ensures suitability for our analysis of length instruction following capabilities under flexible-length reasoning settings.

**Creative Generation Task.**     This task focuses on generating creative and imaginative texts, with output length inherently flexible and dependent on the context, genre, and intent of generation. Due to the wide variability in the expected length and structure of generated outputs, it aligns with the *Flexible-Length Paradigms*. Representative tasks include **Text Completion**, **Dialogue Generation**, **Story Composition**, **Poem Generation**, and **Sentence Expansion**, each requiring adaptation to various expressive and stylistic demands.

For analysis under the flexible-length setting, we curated a total of **120** examples from open-source datasets, evenly divided into **60** English and **60** Chinese instances. Each example is selected to represent distinct creative generation subtypes, ensuring both linguistic diversity and task coverage. All samples are manually verified to confirm fluency, coherence, and contextual appropriateness for creative generation.

The English data are collected from the following **4** datasets, with **15** instances sampled from each:

- **Internet Movie Script Dataset[82]**: Film scripts from IMSDb[10]; we use it for *Script Expansion*, prompting models to extend or elaborate scenes.

- **ROCStories [75]**: Five-sentence commonsense stories; we utilize for *Story Expansion*, where models develop or extend narratives.

- **WritingPrompts [31]**: Imaginative writing prompts from Reddit's WRITINGPROMPTS[11]; we employ it for *Prompt-based Creative Writing*.

- **CNN/DailyMail [42]**: News summaries from articles; we applie it to *News Writing*, prompting models to reconstruct or expand original news content.

The Chinese data are drawn from the following **three** datasets:

---

[10]https://www.imsdb.com/
[11]https://www.reddit.com/r/WritingPrompts/

| QA Template |
|---|
| `[Instruction]` Answer this question: {{content}}
`[Requirement]` Your answer must be {{control_method}} {{length_constraint}} words long. |

Table 4: Refinement Template for QA Task. All subtypes share the same template.

- **WebNovel [51]**: Online fiction excerpts and generation instructions; **15** examples are selected for *Story Expansion*, generating content based on given story segments.
- **NLPCC 2017**[12]: Reference summaries from the summarization track; instances are used for *Social Commentary Writing* and *News Writing*, totaling **20** examples.
- **WeiXin Public Corpus**[13]: Articles from WeChat public accounts; **25** examples are selected for *Blog Writing*, generating opinion or commentary articles based on titles.

| Summarization Template |
|---|
| **Subtype: Encyclopedia**
`[Instruction]` Summarize this encyclopedia article: {{content}}
`[Requirement]` Your summary must be {{control_method}} {{length_constraint}} words long. |
| **Subtype: Report**
`[Instruction]` Summarize this report: {{content}}
`[Requirement]` Your summary must be {{control_method}} {{length_constraint}} words long. |
| **Subtype: Public Discourse**
`[Instruction]` Summarize this public discourse: {{content}}
`[Requirement]` Your summary must be {{control_method}} {{length_constraint}} words long. |
| **Subtype: Academic Article**
`[Instruction]` Summarize this academic article: {{content}}
`[Requirement]` Your summary must be {{control_method}} {{length_constraint}} words long. |
| **Subtype: Book**
`[Instruction]` Summarize this book: {{content}}
`[Requirement]` Your summary must be {{control_method}} {{length_constraint}} words long. |
| **Subtype: Dialogue**
`[Instruction]` Summarize this dialogue: {{content}}
`[Requirement]` Your summary must be {{control_method}} {{length_constraint}} words long. |

Table 5: Refinement Template for Summarization.

## B.3  Refinement Process

Since the initial data are directly obtained from existing datasets or generated by LLMs, the raw samples lack explicit length constraints or specific task instructions, so these data cannot be used

---

[12]http://tcci.ccf.org.cn/conference/2017/taskdata.php
[13]https://github.com/nonamestreet/weixin_public_corpus

| Reasoning Template |
|---|
| **Subtype: Deductive**
[Instruction] Solve this deductive reasoning problem: {{content}}
[Requirement] Your reasoning must be {{control_method}} {{length_constraint}} words long. |
| **Subtype: Inductive**
[Instruction] Infer a general rule from this observed pattern: {{content}}
[Requirement] Your reasoning must be {{control_method}} {{length_constraint}} words long. |
| **Subtype: Abductive**
[Instruction] Determine the most plausible explanation for this observation: {{content}}
[Requirement] Your reasoning must be {{control_method}} {{length_constraint}} words long. |
| **Subtype: Analogical**
[Instruction] Draw an analogy to explain this question: {{content}}
[Requirement] Your reasoning must be {{control_method}} {{length_constraint}} words long. |
| **Subtype: Cause and Effect**
[Instruction] Analyze the causal relationship in this scenario: {{content}}
[Requirement] Your reasoning must be {{control_method}} {{length_constraint}} words long. |
| **Subtype: Probabilistic**
[Instruction] Evaluate the likelihood of this outcome based on probability: {{content}}
[Requirement] Your reasoning must be {{control_method}} {{length_constraint}} words long. |

Table 6: Refinement Template for Reasoning.

to evaluate the length instruction following capability. To align the collected data with the goals of the *Flexible-Length Paradigm*, *i.e.*, apply them to evaluations at different lengths, we implement a structured refinement process, augmenting samples with task-specific instructions and explicit length constraints. This approach ensures that the refined data adheres to the desired format without compromising task relevance. The refinement process allows us to better assess the model length instruction following while excluding other length-independent conditions as much as possible.

**Refinement Methodology.** We adopt a standardized template, **[Instruction] + [Requirement]**, to guide the refinement process. The **[Instruction]** component specifies the task type and the desired model generation, while the **[Requirement]** sets the length constraints of the output. This dual-component framework precise control over both task appropriateness and output flexibility. For each task category or subtype, unique **[Instruction]** and **[Requirement]** templates are designed to fit the nature of the task better. The templates include the following key components: 1) **content**: The raw data collected from original sources, serving as task inputs. (2) **control_method**: It includes three possible options:

- **Equal To**: Output length must exactly match the specified constraint.

- **At Most**: Output length must not exceed the specified constraint.

- **At Least**: Output length must meet or exceed the specified constraint.

(3) **length_constraint**: Defines the target output length. In our benchmark, the values can be set to {16, 32, 64, 128, 256, 512, 1024, 2048, 4096, 8192, 16384, 32768}.

Templates for each task category and subtype are presented in Table 4, Table 5, Table 6 and Table 7. The Chinese dataset follows the same structural approach, with template components expressed in Chinese.

| **Creative Generation Template** |
|---|
| **Subtype: Social Commentary Writing**
[Instruction] Write a social commentary based on the following content: {{content}}
[Requirement] Your commentary must be {{control_method}} {{length_constraint}} words long. |
| **Subtype: Prompt-based Creative Writing**
[Instruction] Write a creative piece based on this prompt: {{content}}
[Requirement] The piece must be {{control_method}} {{length_constraint}} words long. |
| **Subtype: Script Expansion**
[Instruction] You are given an excerpt from a script: {{content}}
[Requirement] The continuation must be {{control_method}} {{length_constraint}} words long. |
| **Subtype: News Writing**
[Instruction] Write a news article based on the following content: {{content}}
[Requirement] Your news article must be {{control_method}} {{length_constraint}} words long. |
| **Subtype: Story Expansion**
[Instruction] Expand the following story: {{content}}
[Requirement] Your expanded story must be {{control_method}} {{length_constraint}} words long. |
| **Subtype: Blog Writing**
[Instruction] This is a title from a WeChat public account: {{content}}
[Requirement] Write a full article based on this title. The article must be {{control_method}} {{length_constraint}} words long. |

Table 7: Refinement Template for Creative Generation.

## B.4 Examples of Tasks

Below we present illustrative examples drawn from each of the four *Flexible-Length Paradigm* tasks covered by LIFEBENCH.

> **An Example Data of QA Task**
>
> [Instruction] Respond to this question:
> How to Use Git Effectively.
> [Requirement] Your response must be at least 1024 words long.

**Question Answering (QA).** The QA examples include open-ended queries enabling varying answer depths. In the example, we use *"How to use git efficiently"* as the basic question and *"at least"* as the control method, which are marked in blue. It is easy to find that our evaluation questions are very consistent with actual application scenarios and are often encountered in practice.

**Summarization.** Summarization tasks require condensing detailed content into a summary that can flexibly meet different length criteria. The examples provided Given a novel, our summarization task requires the model to summarize such rich text input and use *Equal To* as a control method. This is

also a common task. When people use AI assistants, they often ask the model to summarize long texts such as technical reports, academic papers, and novels.

---

**An Example Data of Summarization Task**

[Instruction] Summarize this book:
I was born in Tuckahoe, near Hillsborough, and about twelve miles from Easton, in Talbot county, Maryland. I have no accurate knowledge of my age, never having seen any authentic record containing it. By far the larger part of the slaves know as little of their ages as horses know of theirs, and it is the wish of most masters within my knowledge to keep their slaves thus ignorant. I do not remember to have ever met a slave who could tell of his birthday. They seldom come nearer to it than planting-time, harvest-time, cherry-time, spring-time, or fall-time. A want of information concerning my own was a source of unhappiness to me even during childhood. The white children could tell their ages. I could not tell why I ought to be deprived of the same privilege. I was not allowed to make any inquiries of my master concerning it. He deemed all such inquiries on the part of a slave improper and impertinent, and evidence of a restless spirit. The nearest estimate I can give makes me now between twenty-seven and twenty-eight years of age. I come to this, from hearing my master say, some time during 1835, I was about seventeen years old.
My mother was named Harriet Bailey. She was the daughter of Isaac and Betsey Bailey, both colored, and quite dark. My mother was of a darker complexion than either my grandmother or grandfather.
My father was a white man. He was admitted to be such by all I ever heard speak of my parentage. The opinion was also whispered that my master was my father; but of the correctness of this opinion, I know nothing; the means of knowing was withheld from me. My mother and I were separated when I was but an infant–before I knew her as my mother. It is a common custom, in the part of Maryland from which I ran away, to part children from their mothers at a very early age.
...
[Requirement] The summary must be equal to 128 words long.

---

**Reasoning.** Reasoning examples involve logical inference tasks where output lengths flexibly range from succinct conclusions to detailed step-by-step logical justifications. In this example, we query the model to generate 8192 words to infer *"A country's currency experiences abnormal exchange rate fluctuations. Provide a reasonable explanatory model."* This length instruction allows the model to give a more detailed and logical reasoning process.

---

**An Example Data of Reasoning Task**

[Instruction] Determine the most plausible explanation for this observation:
A country's currency experiences abnormal exchange rate fluctuations. Provide a reasonable explanatory model.
[Requirement] Your reasoning must be equal to 8192 words long.

---

**Creative Generation.** Creative generation examples demonstrate tasks demanding varied textual complexity and content richness. In this example, we give the model a portion of a script that had already been written, and then ask the model to continue writing, and *at most* continued writing 512 words. This task is popular among literary and artistic workers in real life, because they sometimes rely on LLMs to create drafts.

---

**An Example Data of Creative Generation Task**

[Instruction] You are given an excerpt from a script:
PEDDLER: Oh I come from a land
From a faraway place
Where the caravan camels roam
Where they cut off your ear /Where it's flat and immense

---

> If they don't like your face /And the heat is intense
> It's barbaric but hey–it's home!
> When the wind's at your back
> And the sun's from the west
> And the sand in the glass is right
> Come on down
> Stop on by
> Hop a carpet and fly
> To another Arabian night!
>
> Arabian nights
> Like Arabian days
> More often than not
> Are hotter than hot
> In a lot of good ways
>
> Arabian nights
> 'Neath Arabian moons
> A fool off his guard
> ...
> [Requirement] Based on this, continue the scene and develop the storyline. The continuation must be at most 512 words long.

These examples collectively highlight how the selected NLP tasks inherently support varied output lengths and illustrate critical evaluation scenarios within LIFEBENCH, emphasizing models' flexibility and precision in following length-specific instructions.

# C   Details of LIFEBENCH-SUPPLEMENTARY

Building upon our main benchmark, we introduce two supplementary datasets and a lite version to further enhance the diversity and utility of LIFEBENCH. As described in Appendix B.1, the primary scope of LIFEBENCH is on natural language generation tasks within the *Flexible-Length Paradigm*. However, two additional paradigms, Fixed-Length and Adaptive-Length, also present meaningful opportunities for controlled length evaluation, despite not being inherently suited for arbitrary length constraints. To this end, we present LIFEBENCH-LABEL for tasks under the *Fixed-Length Paradigm* and LIFEBENCH-REFACTOR for the *Adaptive-Length Paradigm*. Additionally, we provide LIFEBENCH-LITE, a compact subset designed for efficient, rapid evaluation of a LLM's length instruction following capabilities. The results for LIFEBENCH-LABEL and LIFEBENCH-REFACTOR are reported in Appendix J.

## C.1   LIFEBENCH-LABEL

For tasks under the *Fixed-Length Paradigm*, we categorize them as *Label Tasks*, which include classic natural language genertion tasks such as **text classification**, **sentiment analysis**, and **toxic content** detection. Data were collected from seven public datasets according to the following criteria: (1)The output is a concise, unambiguous category or short phrase representing the answer; (2) No additional explanation, reasoning, or extended generation is required; (3) Samples with explicit length constraints, multi-label requirements, or ambiguous context are excluded.

A total of 60 label task samples were curated, with 30 in Chinese and 30 in English. As these data are best suited for relatively short, fixed-length outputs, we set the length constraints to $2, 4, 8$ words, thereby addressing the gap in short-length settings within LIFEBENCH.

The English label-task data were sourced from the following **four** datasets

- **AG News [119]**: A subset of AG's corpus [5], containing titles and descriptions from the four largest classes ("World", "Sports", "Business", "Sci/Tech"). We selected 9 samples (Text type: *News*).
- **Amazon Fine Foods Reviews [72]**: Reviews of fine foods from Amazon[14]; 6 samples were selected (Text type: *Review*).
- **Text Classification on Emails [94]**: A dataset of email exchanges among journalists; 9 samples were selected (Text type: *Email*).
- **Hate Speech [24]**: Tweets collected from Twitter[15] containing hate speech; 6 samples were selected (Text type: *Tweets*).

The Chinese label-task data were collected from the following **three** datasets

- **ChnSentiCorp [73]**: Hotel review dataset with positive and negative polarity; 10 samples were selected (Review).
- **Online Shopping [73]**: Reviews from 10 shopping categories, each with positive and negative sentiment; 10 samples were selected (Review).
- **Weibo Senti [73]**: Sentiment-annotated posts from Sina Weibo[16]; 10 samples were selected (Tweets).

Below, we provide the refinement template and a representative example for label tasks.

> **An Example Data of Lable Task**
>
> [Instruction] Here is a user review:
> I ALWAYS read Amazon reviews before I buy a product. I don't know what happened. Maybe I was in a hurry but I'll NEVER make that mistake again. If I had read the reviews I could have avoided setting my head on fire.

---

[14] https://www.amazon.com/
[15] https://x.com/
[16] https://weibo.com/

> I took ONE bean – it must have been the magic one – in the next second I was draped over the kitchen island gasping for breath and crying...yes, I said crying..boo hooing like a baby. It felt like someone stuffed a blazing hot poker up my nostrils and my head was going to explode. My eyes watered for over 30 minutes.
> I would only recommend these if you like to set your head on fire for fun or you have a SEVERELY blocked sinus and you have tried all other possible remedies to open it. WARNING: I am NOT recommending that you use these Bunker Busters to open your sinus. I don't know what will happen. For all I know your eyeballs could explode. I'm just saying that ONE bean definitely opened my sinus...but it was NOT worth the pain it took to open it. The only reason I gave them 2 stars is because I stupidly did not read the reviews and I am being very nice (since my sinus finally stopped burning 2 days after!!). Seriously, these should come with a BIG WARNING. Amazon, you should add your own warning. Somebody's probably gonna come after you for these things. They could be dangerous.
> [Requirement] Provide a label to the review accordingly. The label must be equal to 4 words long.

| **Label Template** |
|---|
| **Subtype: News**
[Instruction] You are given a news article excerpt: {{content}}
[Requirement] Label the article based on its topic. The label must be {{control_method}} {{length_constraint}} words long. |
| **Subtype: Review**
[Instruction] Here is a user review: {{content}}
[Requirement] Provide a label to the review accordingly. The label must be {{control_method}} {{length_constraint}} words long. |
| **Subtype: Tweets**
[Instruction] You are given a short text from a tweet: {{content}}
[Requirement] Provide a label that best represents the tweet. The label must be {{control_method}} {{length_constraint}} words long. |
| **Subtype: Email**
[Instruction] The following text is an excerpt from an email: {{content}}
[Requirement] Provide a label that best categorizes it. The label must be {{control_method}} {{length_constraint}} words long. |

Table 8: Refinement Template for Label.

## C.2 LIFEBENCH-REFACTOR

For tasks under the *Adaptive-Length Paradigm*, where the output length is closely tied to the input length, we categorize them as *Refactor Tasks*. This category includes classic natural language generation tasks such as **Machine Translation**, **Text Style Transfer**, and **Paraphrase Generation**. We construct LIFEBENCH-REFACTOR directly using the datasets mentioned in B.2 and C.1.

To ensure the validity of the tasks, we collect samples according to ten predefined length constraints: $16, 32, 64, 128, 256, 512, 1024, 2048, 4096, 8192$ words, requiring that the sample's output length deviates by no more than 30% from the target constraint. For tasks with shorter length constraints, we select samples that allow for flexible restructuring within the target length, ensuring that the output remains fluent and semantically faithful to the source. For tasks with longer length constraints, we choose texts that contain sufficient substantive content to support meaningful transformation, so that the refactored outputs are coherent, relevant, and non-repetitive. In total, we collect 13 samples for Machine Translation (subtype: *Translate*), 99 samples for Text Style Transfer (subtype: *Style Conversion*), and 68 samples for Paraphrase Generation (subtypes: *Reversal*, *Professionalization*, and

*Softening*), corresponding to sentiment reversal, rewriting in a more specialized register, and making text less offensive, respectively.

Due to the diverse sources of the datasets, a unified refinement template could not be applied to all samples. Therefore, we manually refined each sample. Below, we provide a representative example for the Refactor task.

> **An Example Data of Lable Task**
>
> [Instruction] You are given a statement:
> kind of remind me of the flavor and consistency of a s'more, they are surprisingly chocolatey despite the somewhat small about that is in each one and they are not super hard, so it's almost like graham crackers covering them. Me and my wife ended up finishing off the first box in two days an buying 3 more we liked them so much.
> [Requirement] Modify the statement to express the opposite sentiment. The modified statement must be equal to 64 words long.

### C.3 LIFEBENCH-LITE

We constructed LIFEBENCH-LITE, a condensed benchmark version, by selecting 60 samples (30 Chinese, 30 English) from LIFEBENCH. The selection emphasized task diversity, ensuring comprehensive coverage of all categories and subtypes within the full benchmark. The detailed distribution of tasks is presented in Table 9. Evaluated under identical length constraints as described in Section 4.2, LIFEBENCH-LITE comprises 1,800 instances, achieving approximately a $6\times$ speedup in evaluation relative to the full-scale LIFEBENCH dataset.

Table 9: Distribution of Task Categories and Subtypes in LIFEBENCH-LITE.

| Task Category | Subtype | Count |
|---|---|---|
| QA | Food | 1 |
| | Technology | 2 |
| | Sports | 1 |
| | Lifestyle | 4 |
| | Knowledge | 3 |
| | Health | 2 |
| | Entertainment | 3 |
| | Society | 3 |
| | Business | 1 |
| | **Total (QA)** | **20** |
| Reasoning | Probabilistic | 3 |
| | Deductive | 1 |
| | Inductive | 4 |
| | Abductive | 3 |
| | Cause and Effect | 2 |
| | Analogical | 2 |
| | **Total (Reasoning)** | **15** |
| Summarization | Public Discourse | 1 |
| | Dialogue | 1 |
| | Report | 1 |
| | Encyclopedia | 1 |
| | Academic Article | 1 |
| | **Total (Summarization)** | **5** |
| Creative Generation | Script Expansion | 1 |
| | Story Expansion | 6 |
| | News Writing | 5 |
| | Prompt-based Creative Writing | 4 |
| | Blog Writing | 2 |
| | Social Commentary Writing | 2 |
| | **Total (Creative Generation)** | **20** |

# D  Detailed Experimental Setup

In this section, we describe the experimental settings in detail, some of which are not presented due to the page limitation of the main paper. In Section D.1, we introduce the model types, sources, input and output window sizes used in our experiments. In Section D.2, we visualize the metrics and how they are calculated for different control methods.

## D.1  Model Setup

We summarize the API or model code for all evaluated models in the Table 10, along with the maximum context window and maximum output length. We use green to mark the reasoning model and purple to mark the long-text generation enhancement model. All experiments on open-source models are done in the same computation environment with a cluster of eight NVIDIA 80GB H800 GPUs, while experiments on proprietary models are conducted on a CPU server.

Table 10: Model cards.

| Model Name | API/Model Code | Context Window | Max Output Length |
|---|---|---|---|
| GPT-4o mini [77] | OpenAI api: gpt-4o-mini-2024-07-18 | 128,000 tokens | 16,384 tokens [4] |
| GPT-4o [78] | OpenAI api: gpt-4o-2024-11-20 | 128,000 tokens | 16,384 tokens [5] |
| o1-mini [79] | OpenAI api: o1-mini-2024-09-12 | 128,000 tokens | 65,536 tokens [6] |
| o3-mini [80] | Azure api: o3-mini-2024-12-01-preview | 200,000 tokens | 100,000 tokens [7] |
| Claude-3.7-Sonnet [4] | Anthropic api: claude-3-7-sonnet-20250219 | 200,000 tokens | 64000 tokens [8] |
| Claude-3.7-Sonnet-Thinking [4] | Anthropic api: claude-3-7-sonnet-20250219[1] | 200,000 tokens | 64000 tokens [8] |
| Gemini-2.0-Flash [34] | Google api: gemini-2.0-flash-001 | 1,048,576 tokens | 8192 tokens [9] |
| Gemini-2.0-Flash-Thinking [34] | Google api: gemini-2.0-flash-thinking-exp-01-21 | 1,048,576 tokens | 65536 tokens [9] |
| Gemini-2.5-Pro [35] | Google api: gemini-2.5-pro-preview-03-25 | 1,048,576 tokens | 65536 tokens [9] |
| Doubao-1.5-Pro [12] | Volcengine api: doubao-1-5-pro-32k-250115 | 256,000 tokens | 16,384 tokens [10] |
| Doubao-1.5-Thinking-Pro [12] | Volcengine api: doubao-1-5-thinking-pro-250415 | 128,000 tokens | 16,384 tokens [11] |
| DeepSeek-V3 [66] | DeepSeek api: deepseek-chat[2] | 64,000 tokens | 8192 tokens [12] |
| DeepSeek-R1 [40] | DeepSeek api: deepseek-reasoner [3] | 64,000 tokens | 8192 tokens [12] |
| Qwen3-32B [108] | Aliyun api: qwen3-32b | 32,768 tokens | 8192 tokens [13] |
| Qwen3-32B-Thinking [108] | Aliyun api: deepseek-reasoner [1] | 32,768 tokens | 8192 tokens [13] |
| Qwen3-235B-A22B [108] | Aliyun api: qwen3-235b-a22b | 32,768 tokens | 8192 tokens [13] |
| Qwen3-235B-A22B-Thinking [108] | Aliyun api: qwen3-235b-a22b [1] | 32,768 tokens | 8192 tokens [13] |
| Llama-3.1-8B-Instruct [36] | meta-llama/Llama-3.1-8B-Instruct | 128,000 tokens | - |
| Llama-3.1-70B-Instruct [36] | meta-llama/Llama-3.1-70B-Instruct | 128,000 tokens | - |
| Qwen2.5-7B-Instruct [108] | Qwen/Qwen2.5-7B-Instruct | 128,000 tokens | 8,192 tokens [14] |
| Qwen2.5-72B-Instruct [108] | Qwen/Qwen2.5-72B-Instruct | 128,000 tokens | 8,192 tokens [15] |
| GLM-4-9B-Chat [33] | THUDM/glm-4-9b-chat | 128,000 tokens | - |
| Mistral-7B-Instruct-v0.2 [14] | mistralai/Mistral-7B-Instruct-v0.2 | 32,000 tokens | - |
| LongWriter-Llama3.1-8B [8] | THUDM/LongWriter-llama3.1-8b | 128,000 tokens | - |
| LongWriter-GLM4-9B [8] | THUDM/LongWriter-glm4-9b | 128,000 tokens | - |
| Suri-I-ORPO [83] | chtmp223/suri-i-orpo | 32,000 tokens | - |

[1] operates in extended thinking mode
[2] points to DeepSeek-V3 2024/12/26
[3] points to DeepSeek-R1 2025/01/20
[4] https://platform.openai.com/docs/models/gpt-4o-mini
[5] https://platform.openai.com/docs/models/gpt-4o
[6] https://platform.openai.com/docs/models/o1-mini
[7] https://platform.openai.com/docs/models/o3-mini
[8] https://docs.anthropic.com/en/docs/about-claude/models/all-models
[9] https://ai.google.dev/gemini-api/docs/models
[10] https://console.volcengine.com/ark/region:ark+cn-beijing/model/detail?Id=doubao-1-5-pro-32k
[11] https://console.volcengine.com/ark/region:ark+cn-beijing/model/detail?Id=doubao-1-5-thinking-pro
[12] https://api-docs.deepseek.com/quick_start/pricing
[13] Aliyun api limit https://bailian.console.aliyun.com/console?tab=doc
[14] https://huggingface.co/Qwen/Qwen2.5-7B-Instruct
[15] https://huggingface.co/Qwen/Qwen2.5-72B-Instruct

To ensure consistency in our experimental setup, we set the max output size to 65,536 tokens for all models, or to the maximum supported size if smaller. Additionally, we set the temperature to 0.8 and top_p to 1 for non-reasoning models, while for reasoning models, we set the reasoning effort to medium if this parameter is supported.

### D.2 Evaluation Metrics

#### D.2.1 Output Word Count Computation.

To accommodate comparisons of model s in legnth instruction following both Chinese and English, we use the following unified word counting strategy. Specifically, the generation length is computed as the sum of the number of Chinese characters and the number of English words (including hyphenated words and contractions), while excluding newline and tab characters. Formally, for a given output, we first count the number of Chinese characters using the Unicode range `[\u4e00-\u9fff]`, and then count English words via the regular expression `\b[a-zA-Z0-9']+\b`. The final word count is the sum of these two components, which allows us to maintain excellent comparability when recording Chinese, English, and mixed Chinese and English generations.

#### D.2.2 Length Deviation and Length Score

In the main text, we introduced *Length Deviation (LD)* and *Length Score (LS)*, and we believe that *LS* is a better evaluation metric. In this section, we will systematically analyze why *LS* is better and demonstrate the rationality of *LS* and its hyperparameter settings.

**Disadvantages of Length Deviation.** To quantitatively assess a model's adherence to various length constraints, we introduce the Length Deviation (*LD*). Intuitively, *LD* provides a normalized measure of deviation, enabling direct comparison across diverse length requirements. For instance, an absolute deviation of 100 words corresponds to an *LD* of 625% for a target length of 16, indicating a severe deviation; however, the same absolute deviation yields approximately 1% for a length constraint of 8,192, a comparatively negligible error. Thus, by normalizing deviation relative to the instructed length, *LD* intuitively captures adherence to length instructions, a capability that raw word-count-based metrics lack.

Nonetheless, the *LD* metric has several notable limitations:

First, under the *Equal To* control method, *LD* suffers from significant bias introduced by its normalization against the target constraint. Specifically, *LD* ranges asymmetrically from $-100\%$ to $+\infty$, complicating interpretation. For example, with a constraint of 64 words, an output of 0 words yields an *LD* of $-100\%$, while an output of 192 words produces an *LD* of $+200\%$. Although both represent serious failures to comply with the instruction, the metric disproportionately penalizes over-generation. Both cases reflect severe failures to follow the instruction, but *LD* unfairly penalizes over-generation more heavily.

Second, because *LD* accommodating both negative and positive values, aggregating results across multiple examples can cause misleading cancellation effects. For instance, given a 64-word constraint, outputs of 0 and 128 words yield *LD* values of $-100\%$ and $+100\%$, respectively. Averaging these cases results in a mean *LD* of $0\%$, inaccurately implying perfect adherence to the instruction.

Third, for the *At Least* and *At Most* control methods, *LD* fails to properly capture the semantics of the instruction. Once the output satisfies the length constraint in instruction (*i.e.*, *exceeding the minimum or not surpassing the maximum*), any further deviation should not incur additional penalties, as all such outputs are equally compliant with the instruction. However, *LD* tends to increase with unnecessary deviation beyond the threshold, such as differentiating between outputs exceeding the minimum by 100% versus 200%. artificially distinguishing between outputs that are, by definition, both correct. This artificial distinction distorts aggregated evaluations, complicating accurate following assessments.

Finally, *LD* exhibits high sensitivity to outliers. Particularly for smaller length constraints, single extreme deviations can disproportionately skew aggregated metrics. For instance, with a 16-word constraint, an output of 2,000 words yields an *LD* of $+12,400\%$, dramatically inflating the overall metric to a single abnormal prediction.

**Why Employ *LS*?** To address these limitations, we further propose *LS* which transforms the relative length deviation into a bounded score, ranging from 0 to 100. The computation of *LS* varies

according to the specific constraint type as follows:

$$LS_E = \begin{cases} 100 \times e^{k_1 \cdot LD}, & \text{if } LD < 0 \\ 100 \times e^{-k_2 \cdot LD}, & \text{if } LD \geq 0, \end{cases}$$

$$LS_M = \begin{cases} 100, & \text{if } LD < 0 \\ 100 \times e^{-k_2 \cdot LD}, & \text{if } LD \geq 0, \end{cases} \tag{3}$$

$$LS_L = \begin{cases} 100 \times e^{k_1 \cdot LD}, & \text{if } LD < 0 \\ 100, & \text{if } LD \geq 0, \end{cases}$$

where $LS_E$, $LS_M$, and $LS_L$ correspond to the *Equal To*, *At Most*, and *At Least* control method, respectively.

By design, *LS* is always non-negative and bounded, eliminating the problematic offsetting of positive and negative deviations inherent in aggregated *LD* values. Furthermore, the bounded range ensures reduced sensitivity to outliers, preventing extreme deviations from disproportionately influencing the overall evaluation.

Below, we detail how *LS* specifically addresses each of the previously discussed issues associated with *LS*:

First, *LS* transforms the asymmetric and unbounded nature of *LD* into a bounded and symmetric metric within $[0, 100]$, significantly enhancing interpretability and comparability across different constraints. The exponential mapping ensures a controlled and smooth penalization for both under- and over-generation. Thus, under a 64-word constraint, producing either 0 or 192 words results in comparably low scores, accurately reflecting the severity of deviation and mitigating the previous bias against over-generation inherent in *LD*.

Second, since *LS* is strictly non-negative, it does not involve the cancellation effects arising from aggregating positive and negative. Consequently, mean *LS* scores reliably represent a model's overall capacity to follow length instructions, without being distorted by offsetting deviations.

Third, *LS* aligns explicitly with the semantic intent of each constraint type. For *At Most* and *At Least* method, *LS* assigns full maximum scores to all outputs complying with the specified threshold, without differentiating based on excess compliance. Penalties apply only to outputs that violate the constraint, facilitating fairer comparisons across diverse length requirements.

Finally, due to its bounded formulation, *LS* demonstrates robustness against outliers. Extremely aberrant cases, particularly under smaller constraints, do not disproportionately inflate the aggregate score, ensuring that evaluations are stable and not dominated by a few anomalous predictions.

**Hyperparameter Settings.** In Eq.3, we set $k_1 = 5$ and $k_2 = 2$ for all settings. This choice reflects the observation that over-generation ($LD \geq 0$) is generally more acceptable than under-generation, as excess content can be more easily trimmed post hoc, whereas missing content may need to be complete by users. Consequently, the scoring function penalizes under-generation more severely and over-generation more gently, thereby increasing sensitivity to insufficient length adherence.

The exponential formulation provides fine-grained discrimination when deviations ($|LD|$) are small, allowing the metric to capture subtle deviations from the target length constraint. As $|LD|$ grows large, the decay rate gradually moderates, acknowledging that distinguishing finely among severely non-compliant outputs is no longer practically meaningful. Thus, the score naturally approaches zero as *LD* nears either $-1$ or $+\infty$, representing a complete failure in adhering to the length instructions.

For the *At Most* and *At Least* constraint methods, the scoring functions are defined analogously based on their semantic requirements. Specifically, outputs that satisfy the instruction (length $\leq$ constraint for *At Most*, or length $\geq$ constraint for *At Least*) receive the maximum score of 100, reflecting perfect compliance. Violations incur score penalties consistent with the corresponding side of the *Equal To* method, ensuring comparability and coherence across all constraint types.

Figure 8 provides a visualization of the *LS* function, illustrating their intended asymmetry between under- and over-generation penalties. As depicted, all three constraint methods yield a perfect score when outputs fully comply with the length instructions. Under the *Equal To* setting, an *LD* of $-50\%$ yields a score of 8.2, whereas an *LD* of $+50\%$ results in a higher score of 22.3, clearly demonstrating the intended asymmetric treatment At extreme deviations (*e.g.*, $LD = -100\%$ or $+200\%$), the

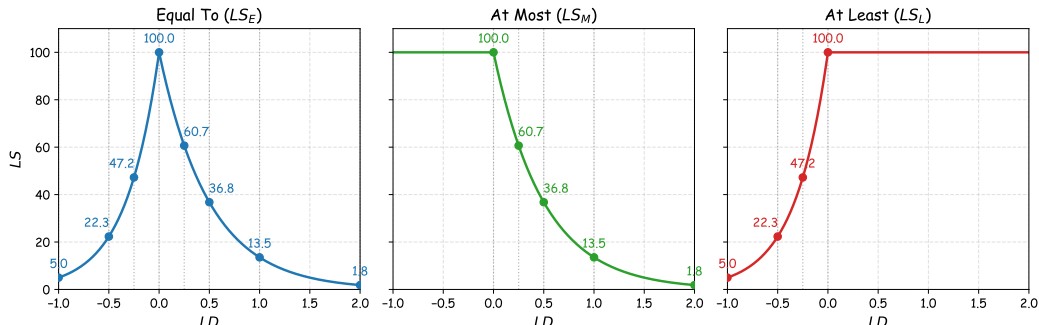

Figure 8: Visualization of *Length Score* for different control methods.

score rapidly approaches zero, achieving values such as 0.7 and 0.2, respectively. This indicates that distinctions among significantly poor cases are appropriately minimized, aligning the metric's behavior with our intended design principles and addressing the limitations previously discussed.

# E  Detailed Results of Main Experiments

## E.1  Detailed *Length Deviation* and *Length Score* across all length constraints

In this section, we provide a detailed breakdown of the main experimental results. Table 11, Table 12, and Table 13 present *Length Deviation* and *Length Score* for all models under the three control methods: *Equal To*, *At Most*, and *At Least*, respectively. For the *At Most* setting, we set *Length Deviation* to zero whenever the output length is below the specified constraint. Similarly, for *At Least*, we set *Length Deviation* to zero for data where the output length exceeds the constraint, in order to more accurately reflect model bias under each scenario. For each control method, we report the mean value of *Length Deviation* and *Length Score* over all evaluated length constraints. For the *Equal To* control method, we report the mean of the absolute value of *Length Deviation*.

Under the *Equal To* control method, length instruction following proves challenging for most models. Specifically, 23 out of 26 evaluated models achieve an *Length Score* below 60, with only three models—o3-mini, Claude-Sonnet-Thinking, and Gemini-2.5-Pro—attaining relatively high scores (75.4, 61.3, and 60.0, respectively). For shorter length constraints ($\leq$128 words), o3-mini and Gemini-2.5-Pro consistently deliver strong performance (*Length Score* $>$80 and $>$70, respectively), whereas eight models score below 60. For medium-length constraints (256–2048 words), o3-mini maintains robust performance (*Length Score* $>$70), while Gemini-2.5-Pro exhibits a substantial drop (from 81 to 37). For the longest constraints ($\geq$4096 words), no model reliably exceeds an *Length Score* of 40. Notably, models explicitly enhanced for long-text generation (LongWriter-Llama3.1-8B and LongWriter-GLM4-9B) achieve significant relative improvements over their base versions, especially at the longest constraint (8192 words), with increases of 22 and 32 points, respectively. However, even these optimized models show limited absolute effectiveness, underscoring ongoing challenges in precise long-length following.

Under the *At Most* control method, models generally perform better, with 16 out of 26 models achieving *Length Score* scores above 80 across all length constraints. The largest deviations occur predominantly at short constraints (e.g., 16 or 32 words). Interestingly, models specialized for long-text generation perform poorly relative to base models in shorter-length scenarios. For example, Suri-I-ORPO reaches an *Length Deviation* of 1838% while still achieving an *Length Score* of 53.6, reflecting a failure to follow length instructions in certain cases—specifically, generating overlong outputs that substantially inflate *Length Deviation*. This highlights a key limitation in length instruction following. For length constraints greater than 2048, almost all models reach *Length Score* scores above 90, with GPT-4o and DeepSeek-R1 attaining perfect scores across all relevant constraints. These results suggest that, under the *At Most* setting, length instruction following remains challenging, especially for short constraints and for models optimized for long-text generation.

Conversely, the *At Least* control method reveals an opposite trend: models exhibit high adherence to shorter constraints ($\leq$512 words), with 23 of 26 models achieving *Length Score* scores above 90. However, as the length constraint increases (*e.g.*, 1024 and above), most models show a clear drop in performance. When the constraint reaches 8192, 18 out of 26 models have *Length Score* below 40. o3-mini demonstrates the best performance under this setting, achieving an *Length Deviation* of 76.2 and an *Length Score* of -12%, surpassing even those models specifically enhanced for long-text generation, despite the latter outperforming their base models on long constraints. The general failure of models to meet long length constraints highlights the significant room for improvement in long-text generation.

Collectively, our detailed analysis reveals that length instruction following remains a significant challenge for current LLMs, particularly under the *Equal To* control method setting and for long length constraints. While some models, such as o3-mini, consistently demonstrate robust performance across diverse scenarios, the majority exhibit substantial degradation as constraints become more demanding. Even for models specifically enhanced for long-text generation, improvements are often limited to relative gains over their base models, and absolute performance on precise or long constraints remains unsatisfactory. These results underscore the need for further research into more effective length instruction following, especially to address the deficiencies observed under challenging constraints.

Table 11: *Length Score* and *Length Deviation* for all length constraints under the *Equal To* control method.

| Models | Metric | Length Constraints | | | | | | | | | | AVG |
|---|---|---|---|---|---|---|---|---|---|---|---|---|
| | | 16 | 32 | 64 | 128 | 256 | 512 | 1024 | 2048 | 4096 | 8192 | |
| GPT-4o mini | LS | 75 | 67.8 | 68.9 | 62.8 | 69.8 | 66.8 | **74.7** | 25.5 | 5.7 | 1.7 | 51.9 |
| | LD | 14% | 22% | 21% | 22% | 18% | 21% | 10% | 34% | 65% | 85% | 31% |
| GPT-4o | LS | 57 | 67.7 | **72.2** | 67 | 69.6 | 66.5 | 69 | 16.5 | 4.1 | 1.6 | 49.1 |
| | LD | 15% | 12% | 13% | 20% | 19% | 20% | 12% | 43% | 70% | 86% | 31% |
| o1-mini | LS | 71.8 | 70.5 | 67.8 | 66.6 | 62.5 | 51.2 | 40.3 | 34.2 | 15.2 | 3.1 | 48.3 |
| | LD | 11% | 13% | 16% | 23% | 25% | 34% | 48% | 46% | 56% | 81% | 35% |
| o3-mini | LS | **96.7** | **93.5** | **87.9** | **81.8** | 79.4 | **78.9** | **77.8** | **73.9** | **66.9** | 16.7 | **75.4** |
| | LD | 1% | 2% | 4% | 5% | 6% | 6% | 8% | 13% | 16% | 66% | 13% |
| Claude-3.7-Sonnet | LS | 58.5 | 53 | 63.6 | 74.3 | 76.4 | 70.6 | 53.6 | 43.7 | 34.4 | 25.9 | 55.4 |
| | LD | 91% | 42% | 24% | 18% | 10% | 9% | 15% | 21% | 32% | 42% | 30% |
| Claude-3.7-Sonnet-Thinking | LS | 58 | 61 | 70.6 | **77.4** | **79.1** | 72.4 | 58.1 | 53.3 | 46.7 | **36.1** | **61.3** |
| | LD | 131% | 51% | 21% | 13% | 9% | 10% | 15% | 18% | 26% | 38% | 33% |
| Gemini-2.0-Flash | LS | 60.9 | 63.7 | 66.2 | 59.8 | 59.7 | 38.2 | 47.7 | 52.7 | 25.9 | 9.2 | 48.4 |
| | LD | 29% | 23% | 21% | 28% | 28% | 61% | 46% | 27% | 36% | 57% | 36% |
| Gemini-2.0-Flash-Thinking | LS | 62.8 | 51.3 | 55.1 | 53.4 | 49 | 30 | 32.1 | 42.5 | 42.1 | 21.9 | 44 |
| | LD | 35% | 27% | 31% | 27% | 30% | 97% | 119% | 70% | 43% | 47% | 53% |
| Gemini-2.5-Pro | LS | *80.9* | *76.1* | 74.4 | 72.8 | 67.4 | 49.5 | 37 | 48 | *54.3* | **39.4** | 60 |
| | LD | 17% | 9% | 10% | 12% | 15% | 40% | 57% | 44% | 37% | 35% | 28% |
| Doubao-1.5-Pro | LS | 31.4 | 42.6 | 46.8 | 65.5 | 64.2 | 60.6 | 63.5 | 57.3 | 42.4 | 12.3 | 48.7 |
| | LD | 30% | 24% | 21% | 12% | 12% | 14% | 13% | 24% | 27% | 52% | 23% |
| Doubao-1.5-Thinking-Pro | LS | 71.9 | 67.4 | 55.5 | 64.9 | 61.6 | 44.5 | 42 | 55.5 | 34.5 | 8 | 50.6 |
| | LD | 10% | 12% | 16% | 18% | 25% | 45% | 46% | 26% | 32% | 59% | 29% |
| DeepSeek-V3 | LS | 60 | 73.4 | **76.3** | 69.1 | 68.4 | 63.4 | 64.9 | 43.7 | 17.6 | 6.1 | 54.3 |
| | LD | 15% | 12% | 12% | 20% | 21% | 22% | 23% | 28% | 48% | 68% | 27% |
| DeepSeek-R1 | LS | 64.2 | 68.9 | 69.4 | 61.6 | 59.1 | 55.5 | 65.2 | 26.4 | 5 | 1.6 | 47.7 |
| | LD | 32% | 21% | 20% | 28% | 30% | 31% | 16% | 34% | 65% | 84% | 36% |
| Llama-3.1-8B-Instruct | LS | 47 | 47.4 | 51.3 | 55.4 | 62.6 | 68.3 | 32.8 | 9.2 | 3.2 | 3.7 | 38.1 |
| | LD | 146% | 81% | 57% | 39% | 29% | 24% | 48% | 77% | 92% | 108% | 70% |
| Llama-3.1-70B-Instruct | LS | 59.4 | 61.9 | 61 | 62.8 | 61.9 | 61.5 | 38.2 | 11.8 | 3.6 | 1.8 | 42.4 |
| | LD | 52% | 26% | 25% | 23% | 27% | 26% | 94% | 111% | 123% | 106% | 61% |
| Qwen2.5-7B-Instruct | LS | 55.2 | 37.6 | 22.3 | 52.4 | 58 | 66.6 | 44.9 | 18.5 | 12.7 | 4.9 | 37.3 |
| | LD | 25% | 26% | 36% | 18% | 24% | 24% | 24% | 52% | 57% | 75% | 36% |
| Qwen2.5-72B-Instruct | LS | 71.9 | 59 | 50.4 | 70.4 | 62.3 | 61.8 | 63.8 | 24.9 | 23.3 | 18.2 | 50.6 |
| | LD | 16% | 15% | 18% | 16% | 24% | 26% | 17% | 41% | 51% | 59% | 28% |
| Qwen3-32B | LS | 57.4 | 62.7 | 62.2 | 62.1 | **72.5** | 64.5 | 66.3 | 52.5 | 49.6 | 26.6 | 57.6 |
| | LD | 32% | 19% | 14% | 12% | 9% | 11% | 12% | 22% | 22% | 34% | 19% |
| Qwen3-32B-Thinking | LS | 60.2 | 55.8 | 63.2 | 66.4 | 67.3 | 64 | 68 | 56.4 | 48.3 | 28.3 | 57.8 |
| | LD | 28% | 28% | 22% | 20% | 21% | 23% | 20% | 17% | 21% | 29% | 23% |
| Qwen3-235B-A22B | LS | 57.8 | 52.5 | 50.1 | 45.9 | 72.8 | 73.3 | 68 | 42.7 | 30.7 | 27.3 | 52.1 |
| | LD | 28% | 30% | 22% | 19% | 10% | 10% | 12% | 24% | 32% | 33% | 22% |
| Qwen3-235B-A22B-Thinking | LS | 55.9 | 56.1 | 57.5 | 66.3 | 70.8 | 63.5 | 63.9 | *60.6* | 49.7 | 32.8 | 57.7 |
| | LD | 39% | 31% | 25% | 16% | 16% | 21% | 18% | 14% | 17% | 25% | 23% |
| GLM-4-9B-Chat | LS | 52.6 | 48.1 | 42.1 | 35.6 | 47.4 | 51.2 | 58.4 | 10.9 | 3.9 | 1.6 | 35.2 |
| | LD | 28% | 24% | 25% | 27% | 29% | 35% | 16% | 54% | 75% | 87% | 40% |
| Mistral-7B-Instruct-v0.2 | LS | 26.4 | 32.1 | 42.8 | 37.4 | 42.9 | 52.1 | 23.1 | 5.9 | 2.3 | 2.1 | 26.7 |
| | LD | 193% | 126% | 58% | 69% | 58% | 24% | 56% | 73% | 87% | 92% | 84% |
| LongWriter-Llama3.1-8B | LS | 21.1 | 24.2 | 35.7 | 42.7 | 44.9 | 54.2 | 20.5 | 18.2 | 31.2 | 26.1 | 31.9 |
| | LD | 254% | 124% | 76% | 39% | 45% | 98% | 143% | 117% | 78% | 48% | 102% |
| LongWriter-GLM4-9B | LS | 35.6 | 35.7 | 33.8 | 37.2 | 39.7 | 52.3 | 30.3 | 30.6 | 45.5 | 33.7 | 37.4 |
| | LD | 63% | 40% | 34% | 33% | 48% | 61% | 66% | 88% | 47% | 35% | 52% |
| Suri-I-ORPO | LS | 30.5 | 34.1 | 33.6 | 31.8 | 32.4 | 37 | 28.7 | 23.9 | 12.5 | 9.3 | 27.4 |
| | LD | 2142% | 965% | 575% | 379% | 318% | 207% | 195% | 109% | 91% | 76% | 506% |

Table 12: *Length Score* and *Length Deviation* for all length constraints under the *At Most* control method.

| Models | Metric | Length Constraints | | | | | | | | | | AVG |
|---|---|---|---|---|---|---|---|---|---|---|---|---|
| | | 16 | 32 | 64 | 128 | 256 | 512 | 1024 | 2048 | 4096 | 8192 | |
| GPT-4o mini | LS | 89.2 | 82.3 | 81.8 | 84 | 84.8 | 84.9 | 97.4 | 99 | **100** | **100** | 90.3 |
| | LD | 27% | 13% | 13% | 10% | 10% | 10% | 2% | 1% | 0% | 0% | 9% |
| GPT-4o | LS | 98.4 | 98.5 | 98.5 | 89.2 | 91.6 | 94.5 | **100** | **100** | **100** | **100** | 97 |
| | LD | 1% | 1% | 1% | 7% | 5% | 3% | 0% | 0% | 0% | 0% | 2% |
| o1-mini | LS | 95.5 | 92.2 | 91.1 | 85 | 75.5 | 75.4 | 86 | 96.2 | 99.7 | **100** | 89.7 |
| | LD | 4% | 5% | 6% | 11% | 19% | 18% | 10% | 3% | 0% | 0% | 8% |
| o3-mini | LS | 99.2 | 99.2 | 99.7 | **100** | **99.9** | 99.5 | 97.9 | 99.5 | **100** | **100** | **99.5** |
| | LD | 0% | 0% | 0% | 0% | 0% | 0% | 1% | 0% | 0% | 0% | 0% |
| Claude-3.7-Sonnet | LS | 90.5 | 92.1 | 93.9 | 94.3 | 99.5 | 99.6 | **100** | **100** | 99.6 | 99 | 96.8 |
| | LD | 82% | 16% | 6% | 7% | 1% | 0% | 0% | 0% | 0% | 1% | 11% |
| Claude-3.7-Sonnet-Thinking | LS | 84.7 | 91 | 93.9 | 96.7 | **99.8** | **99.8** | **100** | 99.8 | 99.7 | 99.6 | 96.5 |
| | LD | 69% | 25% | 9% | 2% | 0% | 0% | 0% | 0% | 0% | 0% | 11% |
| Gemini-2.0-Flash | LS | 92.6 | 93.5 | 95.8 | 91 | 97.9 | 89.7 | 93.8 | 97.9 | **99.9** | **100** | 95.2 |
| | LD | 6% | 5% | 3% | 6% | 1% | 7% | 4% | 2% | 0% | 0% | 3% |
| Gemini-2.0-Flash-Thinking | LS | 94 | 93.8 | 96 | 96.3 | 98.1 | 81.1 | 70.1 | 79 | 94.8 | 98.4 | 90.2 |
| | LD | 41% | 7% | 3% | 4% | 1% | 16% | 27% | 21% | 5% | 1% | 13% |
| Gemini-2.5-Pro | LS | 97.1 | 94.9 | 95.9 | 95.2 | 95.2 | 91.6 | 92.5 | 98.8 | **100** | **100** | 96.1 |
| | LD | 5% | 5% | 4% | 3% | 3% | 6% | 5% | 1% | 0% | 0% | 3% |
| Doubao-1.5-Pro | LS | **99.9** | **99.9** | **100** | **99.9** | **99.9** | 99.7 | **99.9** | **99.9** | **100** | **100** | **99.9** |
| | LD | 0% | 0% | 0% | 0% | 0% | 0% | 0% | 0% | 0% | 0% | 0% |
| Doubao-1.5-Thinking-Pro | LS | **99.7** | **99.7** | **99.9** | 99.2 | 96.1 | 89.8 | 94.1 | 99.6 | **100** | **100** | 97.8 |
| | LD | 0% | 0% | 0% | 0% | 2% | 7% | 4% | 0% | 0% | 0% | 1% |
| DeepSeek-V3 | LS | 89.2 | 96.4 | 98.5 | 96.9 | 94.9 | 96.2 | 98.3 | 98.2 | 98.4 | **100** | 96.7 |
| | LD | 8% | 2% | 1% | 3% | 4% | 3% | 1% | 3% | 1% | 0% | 3% |
| DeepSeek-R1 | LS | 85.8 | 90.4 | 91.7 | 89.3 | 90.3 | 92.4 | 98.5 | **100** | **100** | **100** | 93.8 |
| | LD | 30% | 9% | 5% | 7% | 6% | 5% | 1% | 0% | 0% | 0% | 6% |
| Llama-3.1-8B-Instruct | LS | 52.8 | 59.8 | 63.8 | 71 | 83.6 | 96.2 | **99.9** | 99.2 | 98.9 | 97.5 | 82.3 |
| | LD | 199% | 93% | 54% | 28% | 13% | 3% | 0% | 5% | 9% | 6% | 41% |
| Llama-3.1-70B-Instruct | LS | 71 | 79.3 | 83.1 | 84.7 | 86.8 | 90.9 | 97 | 98.1 | 98.9 | 98.4 | 88.8 |
| | LD | 465% | 28% | 17% | 13% | 10% | 24% | 22% | 21% | 7% | 4% | 61% |
| Qwen2.5-7B-Instruct | LS | 87.3 | 99.3 | **100** | **99.8** | 97.2 | 97.6 | **99.7** | 97.4 | 98.4 | 99.6 | 97.6 |
| | LD | 9% | 1% | 0% | 0% | 2% | 1% | 0% | 4% | 1% | 0% | 2% |
| Qwen2.5-72B-Instruct | LS | 86.2 | 98.5 | 98.2 | 94.6 | 91.3 | 81.1 | 95.7 | 97.9 | 96.2 | 98.8 | 93.8 |
| | LD | 10% | 1% | 1% | 4% | 7% | 12% | 3% | 2% | 3% | 1% | 4% |
| Qwen3-32B | LS | 81 | 94.4 | 99 | **99.9** | 99.5 | **100** | **100** | 99.4 | 98.9 | 99.7 | 97.2 |
| | LD | 15% | 4% | 1% | 0% | 0% | 0% | 0% | 0% | 1% | 0% | 2% |
| Qwen3-32B-Thinking | LS | 81 | 84.8 | 91.5 | 91.5 | 90.4 | 94.7 | 98.9 | 98.6 | 99.1 | 99.7 | 93 |
| | LD | 18% | 11% | 6% | 5% | 6% | 3% | 1% | 1% | 1% | 0% | 5% |
| Qwen3-235B-A22B | LS | 80.9 | 88.4 | 94.2 | 98.8 | 99.1 | 99.6 | 97.3 | 98.9 | 98.9 | **99.9** | 95.6 |
| | LD | 25% | 10% | 9% | 1% | 1% | 0% | 2% | 1% | 1% | 0% | 5% |
| Qwen3-235B-A22B-Thinking | LS | 70.7 | 77.4 | 85.1 | 89.8 | 92.3 | 90.9 | 94.6 | 99.1 | 99.3 | 99.6 | 89.9 |
| | LD | 56% | 30% | 15% | 9% | 5% | 6% | 3% | 1% | 0% | 0% | 13% |
| GLM-4-9B-Chat | LS | 85 | 93.7 | 99 | 99.6 | 97 | 87.1 | 99.4 | 99.8 | 99.2 | 99.2 | 95.9 |
| | LD | 17% | 5% | 1% | 0% | 2% | 8% | 0% | 0% | 3% | 2% | 4% |
| Mistral-7B-Instruct-v0.2 | LS | 53.1 | 68.9 | 81.8 | 83.9 | 82.9 | 96.4 | 97.5 | 97.1 | 98.9 | 98.5 | 85.9 |
| | LD | 136% | 85% | 40% | 77% | 21% | 10% | 23% | 21% | 5% | 2% | 42% |
| LongWriter-Llama3.1-8B | LS | 63.2 | 70 | 75.4 | 85.7 | 88.9 | 95.6 | 95.9 | 85.2 | 86.4 | 94.2 | 84.1 |
| | LD | 713% | 66% | 31% | 34% | 9% | 22% | 63% | 65% | 35% | 9% | 105% |
| LongWriter-GLM4-9B | LS | 79.9 | 90.1 | 97 | 97.1 | 85.5 | 86 | 95.7 | 93.4 | 88.8 | 95.8 | 90.9 |
| | LD | 61% | 23% | 3% | 2% | 13% | 10% | 11% | 13% | 14% | 4% | 15% |
| Suri-I-ORPO | LS | 53.6 | 70.7 | 78.1 | 80.7 | 78.9 | 83.5 | 79.7 | 87.3 | 90.1 | 94.2 | 79.7 |
| | LD | 1838% | 1323% | 374% | 364% | 148% | 77% | 82% | 40% | 25% | 9% | 428% |

Table 13: *Length Score* and *Length Deviation* for all length constraints under the *At Least* control method.

| Models | Metric | Length Constraints | | | | | | | | | | AVG |
|---|---|---|---|---|---|---|---|---|---|---|---|---|
| | | 16 | 32 | 64 | 128 | 256 | 512 | 1024 | 2048 | 4096 | 8192 | |
| GPT-4o mini | LS | **100** | **100** | **100** | 99.8 | **100** | *99.9* | 94.8 | 36.1 | 7 | 2 | 74 |
| | LD | 0% | 0% | 0% | 0% | 0% | 0% | -2% | -27% | -63% | -84% | -18% |
| GPT-4o | LS | *99.7* | 99.7 | **100** | **100** | **100** | **100** | 90.4 | 28.2 | 6.4 | 1.9 | 72.6 |
| | LD | 0% | 0% | 0% | 0% | 0% | 0% | -3% | -32% | -65% | -83% | -18% |
| o1-mini | LS | **100** | **100** | **100** | 99.7 | 99 | 99.8 | 98.4 | 85 | 30.1 | 3.2 | 81.5 |
| | LD | 0% | 0% | 0% | 0% | -1% | 0% | -1% | -7% | -43% | -78% | -13% |
| o3-mini | LS | **100** | **100** | **100** | **100** | **100** | 99.7 | 98.6 | *98.4* | 97 | **76.2** | **97** |
| | LD | 0% | 0% | 0% | 0% | 0% | 0% | -1% | -1% | -2% | -12% | -2% |
| Claude-3.7-Sonnet | LS | **100** | **100** | **100** | **100** | **100** | 99.8 | 97.3 | 85 | 70.8 | 47.7 | 90.1 |
| | LD | 0% | 0% | 0% | 0% | 0% | 0% | -1% | -5% | -11% | -27% | -4% |
| Claude-3.7-Sonnet-Thinking | LS | **100** | **100** | **100** | **100** | **100** | **100** | 98.8 | 91.7 | 80.3 | 62.6 | 93.3 |
| | LD | 0% | 0% | 0% | 0% | 0% | 0% | 0% | -3% | -8% | -18% | -3% |
| Gemini-2.0-Flash | LS | 99.9 | **100** | **100** | **100** | **100** | **100** | *99.7* | 89.9 | 43 | 14.1 | 84.7 |
| | LD | 0% | 0% | 0% | 0% | 0% | 0% | 0% | -3% | -23% | -49% | -7% |
| Gemini-2.0-Flash-Thinking | LS | **100** | **100** | *99.9* | **100** | **100** | **100** | 99.8 | 97.4 | 75.4 | 39.2 | 91.2 |
| | LD | 0% | 0% | 0% | 0% | 0% | 0% | 0% | -1% | -10% | -30% | -4% |
| Gemini-2.5-Pro | LS | **100** | **100** | **100** | **100** | **100** | 99.7 | *99.7* | **99.3** | *91.6* | *65.5* | *95.5* |
| | LD | 0% | 0% | 0% | 0% | 0% | 0% | 0% | 0% | -3% | -14% | -2% |
| Doubao-1.5-Pro | LS | **100** | **100** | **100** | **100** | **100** | **100** | 99.9 | 96.6 | 71.4 | 25.2 | 89.3 |
| | LD | 0% | 0% | 0% | 0% | 0% | 0% | 0% | -1% | -10% | -39% | -5% |
| Doubao-1.5-Thinking-Pro | LS | **100** | **100** | **100** | **100** | **100** | **100** | 99.3 | 91.8 | 53.3 | 12.5 | 85.7 |
| | LD | 0% | 0% | 0% | 0% | 0% | 0% | 0% | -3% | -21% | -51% | -8% |
| DeepSeek-V3 | LS | **100** | **100** | **100** | **100** | **100** | **100** | 97.6 | 64.7 | 27.7 | 6.6 | 79.7 |
| | LD | 0% | 0% | 0% | 0% | 0% | 0% | -1% | -13% | -37% | -65% | -12% |
| DeepSeek-R1 | LS | **100** | **100** | **100** | **100** | **100** | **100** | 92.3 | 40.2 | 6.6 | 1.7 | 74.1 |
| | LD | 0% | 0% | 0% | 0% | 0% | 0% | -2% | -26% | -61% | -83% | -17% |
| Llama-3.1-8B-Instruct | LS | *99.7* | **100** | 99.7 | 99.7 | 99.4 | 99.2 | 65.6 | 23.8 | 14.5 | 16 | 71.8 |
| | LD | 0% | 0% | 0% | 0% | 0% | 0% | -12% | -42% | -62% | -72% | -19% |
| Llama-3.1-70B-Instruct | LS | 99 | 99.7 | 99.4 | 99.3 | 99 | 96.6 | 61.3 | 17.3 | 10.9 | 8 | 69 |
| | LD | 0% | 0% | 0% | 0% | 0% | -1% | -15% | -46% | -66% | -81% | -21% |
| Qwen2.5-7B-Instruct | LS | 99.3 | 98.4 | *99.7* | *99.9* | *99.9* | 98.7 | 63.3 | 25.9 | 18.2 | 6.2 | 71 |
| | LD | 0% | 0% | 0% | 0% | 0% | 0% | -12% | -37% | -52% | -72% | -17% |
| Qwen2.5-72B-Instruct | LS | **100** | *99.8* | 99.8 | **100** | **100** | 99.7 | 88.9 | 61.2 | 63.7 | 29.6 | 84.2 |
| | LD | 0% | 0% | 0% | 0% | 0% | 0% | -3% | -16% | -19% | -43% | -8% |
| Qwen3-32B | LS | 100 | 99.7 | 98.3 | 99.8 | 99.7 | 97.9 | 97.4 | 86.5 | 63.1 | 34.3 | 87.4 |
| | LD | 0% | 0% | 0% | 0% | 0% | -1% | -1% | -5% | -13% | -28% | -5% |
| Qwen3-32B-Thinking | LS | **100** | **100** | **100** | **100** | **100** | **100** | 98 | 77.5 | 65.3 | 37.2 | 87.8 |
| | LD | 0% | 0% | 0% | 0% | 0% | 0% | -1% | -7% | -11% | -24% | -4% |
| Qwen3-235B-A22B | LS | 99.6 | 99.5 | 98.9 | 98.5 | 99.4 | 98.1 | 95.1 | 85.5 | 78.2 | 50.2 | 90.3 |
| | LD | 0% | 0% | 0% | 0% | 0% | -1% | -1% | -5% | -7% | -19% | -3% |
| Qwen3-235B-A22B-Thinking | LS | **100** | **100** | **100** | **100** | **100** | **100** | 98 | 77.8 | 70.5 | 49.2 | 89.6 |
| | LD | 0% | 0% | 0% | 0% | 0% | 0% | -1% | -6% | -8% | -17% | -3% |
| GLM-4-9B-Chat | LS | 99.1 | 99.8 | 99.4 | 97.9 | 99.3 | 97.9 | 70.6 | 12 | 4.7 | 2.4 | 68.3 |
| | LD | 0% | 0% | 0% | -1% | 0% | -1% | -10% | -48% | -71% | -84% | -21% |
| Mistral-7B-Instruct-v0.2 | LS | *99.7* | 99.5 | 99.6 | 99.7 | 99.6 | 80.1 | 31.1 | 10.8 | 5.4 | 4.5 | 63 |
| | LD | 0% | 0% | 0% | 0% | 0% | -6% | -30% | -58% | -76% | -86% | -26% |
| LongWriter-Llama3.1-8B | LS | 96.9 | 93.4 | 98.1 | 95.2 | 94.8 | 81.2 | 63.2 | 65.4 | 60.8 | 40.6 | 79 |
| | LD | -1% | -3% | -1% | -2% | -2% | -7% | -15% | -20% | -24% | -30% | -11% |
| LongWriter-GLM4-9B | LS | 97.3 | 97.2 | 99 | 97.8 | 96.9 | 95 | 73.3 | 83.3 | 80.9 | 47.4 | 86.8 |
| | LD | -1% | -1% | 0% | -1% | -1% | -2% | -10% | -9% | -8% | -23% | -6% |
| Suri-I-ORPO | LS | 99.1 | 99 | 97 | 98.7 | 98.8 | 89.9 | 74.7 | 49.7 | 28.2 | 16.5 | 75.2 |
| | LD | 0% | 0% | -1% | 0% | 0% | -4% | -11% | -27% | -47% | -63% | -15% |

## E.2 Visualization of Model Output Lengths

Figure 9, Figure 10, and Figure 11 illustrate the average output word counts of all models under the *Equal To*, *At Most*, and *At Least* control methods, respectively. Under the *Equal To* setting, we observe that for the shortest length constraint (*i.e.*, 16 words), 5 out of 26 models produce outputs with mean lengths exceeding twice the constraint. Conversely, at the maximum constraint (*i.e.*, 8192 words), 15 models are unable to generate outputs longer than 4096 words, and 7 of these are further limited to outputs not exceeding 2048 words. Notably, long-text-enhanced models such as `LongWriter-Llama3.1-8B` and `LongWriter-GLM4-9B` tend to generate longer outputs across most constraints, suggesting that their optimizations for long-text generation come at the expense of length instruction following, particularly on shorter constraints.

Under the *At Most* control method, most models perform well overall, but there are still noticeable failures in following short constraints ($\leq 256$ words). In particular, `Llama-3.1-8B-Instruct`, `LongWriter-Llama3.1-8B`, and `Suri-I-ORPO` exhibit a counterintuitive trend where the average output length decreases as the constraint increases. This phenomenon occurs specifically in cases where the models fail to follow the length instructions, highlighting their limitations under short constraint conditions.

For the *At Least* control method, all models are able to reliably meet the constraints for shorter lengths ($\leq 512$ words). However, as the constraint increases, an increasing number of models fail to reach the specified length. At the maximum constraint (*i.e.*, 8192 words), only `Gemini-2.5-Pro` is able to consistently meet the requirement, while all other models fall short, underscoring the substantial challenges faced by current models in following long length instructions.

Taken together, while many models can follow moderate length constraints, most struggle with very short or ultra-long constraints. Even long-text enhanced models often fail to meet these extremes and may sacrifice short length instruction following ability. These results show that precise length instruction following—especially at the extreme constraints—remains an open problem for current LLMs.

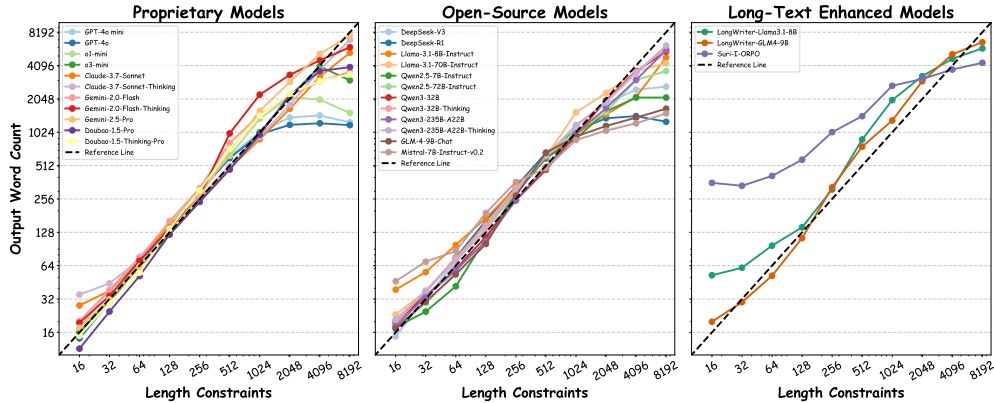

Figure 9: Output word count for all length constraints under the *Equal To* control method.

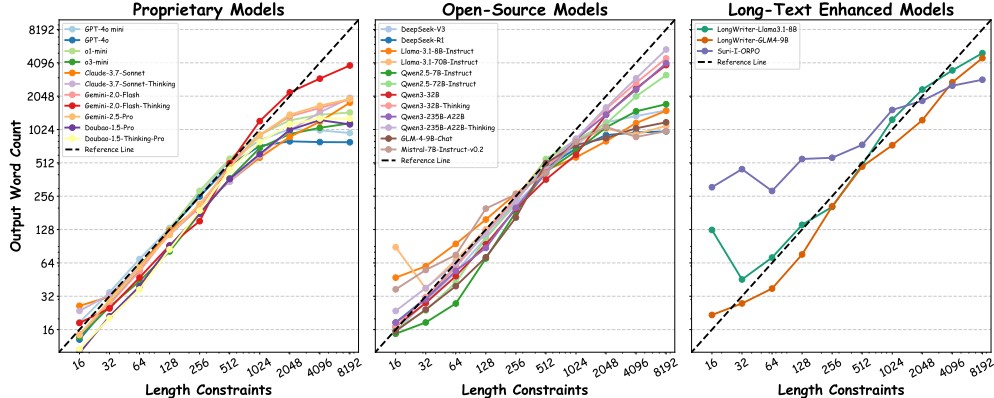

Figure 10: Output word count for all length constraints under the *At Most* control method.

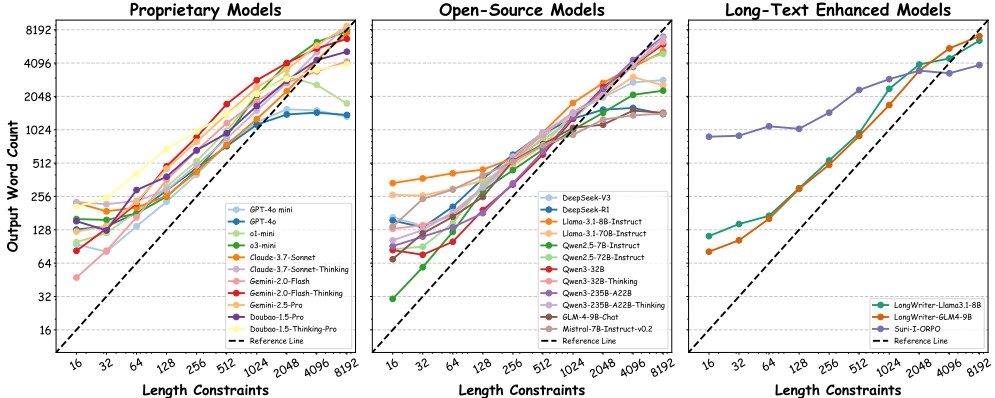

Figure 11: Output word count for all length constraints under the *At Least* control method.

## E.3 Supplementary Results with Extended Length Constraints

While the primary results focus on length constraints up to *8,192* words, some models claim the ability to generate substantially longer outputs due to their larger maximum output length. To further investigate length instruction following under ultra-long constraints, we introduce two additional settings: *16,384* and *32,768* words. Note that, even for the largest setting (*32,768* words), the corresponding number of tokens remains below the maximum output length of most evaluated models. In Table 14, we report results for models whose maximum output length exceeds 8,192 tokens or for which the maximum length is not explicitly specified. For `Claude-3.7-Sonnet` and `Claude-3.7-Sonnet-Thinking`, the output length beyond *8,192* tokens was only available in an experimental version at the time of our evaluation; therefore, results for these models under ultra-long constraints are not shown.

Under the *Equal To* control method, no model achieves an *Length Score* exceeding 30, and only `Gemini-2.5-Pro` and `Llama-3.1-8B-Instruct` surpass 10 out of the 16 evaluated models. For the *At Most* control method, nearly all models perform well, though most open-source models do not attain perfect scores. This may be attributed to uncontrolled output resulting in repeated content that reaches the maximum output length limit. Under the *At Least* control method, model performance mirrors that of the *Equal To* setting, remaining consistently poor.

Overall, these results demonstrate even weaker length instruction following under extended length constraints. As discussed in Section 5.2, the maximum output length claimed by models often differs substantially from actual performance in ultra-long generation scenarios.

Table 14: *Length Deviation* and *Length Score* for extended length constraints.

| Models | Metric | Equal To | | At Most | | At Least | |
|---|---|---|---|---|---|---|---|
| | | 16k | 32k | 16k | 32k | 16k | 32k |
| GPT-4o mini | LD | -94% | - | -95% | - | -94% | - |
| | LS | 1.1 | - | **100** | - | 1 | - |
| GPT-4o | LD | -94% | - | -96% | - | -93% | - |
| | LS | 0.9 | - | **100** | - | 1 | - |
| o1-mini | LD | -93% | -98% | -91% | -96% | -93% | -97% |
| | LS | 1.2 | 0.8 | **100** | **100** | 1 | 0.8 |
| o3-mini | LD | -98% | -100% | -93% | -97% | -72% | -97% |
| | LS | 1.3 | 0.7 | **100** | **100** | 9.9 | 0.9 |
| Gemini-2.0-Flash | LD | -76% | - | -85% | - | -72% | - |
| | LS | 2.6 | - | **100** | - | 3.1 | - |
| Gemini-2.0-Flash-Thinking | LD | -58% | -78% | -72% | -87% | -50% | -73% |
| | LS | 8.5 | 3.1 | 99.8 | **100** | 15.9 | 4 |
| Gemini-2.5-Pro | LD | -33% | -71% | -88% | -94% | -30% | -63% |
| | LS | **26.9** | **6.2** | **100** | **100** | **34.7** | *8.8* |
| Doubao-1.5-Pro | LD | -75% | - | -93% | - | -66% | - |
| | LS | 2.9 | - | **100** | - | 5.3 | - |
| Doubao-1.5-Thinking-Pro | LD | -78% | - | -92% | - | -73% | - |
| | LS | 2.4 | - | **100** | - | 3.3 | - |
| Llama-3.1-8B-Instruct | LD | -69% | -77% | -93% | -95% | -67% | -80% |
| | LS | 4.4 | *12.4* | 98.9 | 99.7 | 15 | **12.6** |
| Llama-3.1-70B-Instruct | LD | -87% | -93% | -96% | -98% | -86% | -91% |
| | LS | 1.1 | 2.8 | *99.9* | *99.9* | 5 | 4.2 |
| GLM-4-9B-Chat | LD | -90% | -96% | -94% | -97% | -91% | -96% |
| | LS | 1.4 | 0.9 | *99.9* | **100** | 1.8 | 1 |
| Mistral-7B-Instruct-v0.2 | LD | -91% | -96% | -93% | -96% | -91% | -95% |
| | LS | 1.8 | 1.2 | 99.3 | **100** | 3 | 1.6 |
| LongWriter-Llama3.1-8B | LD | -54% | -78% | -63% | -81% | -55% | -75% |
| | LS | *10.3* | 3.6 | 97.2 | **99.9** | *13.3* | 5.1 |
| LongWriter-GLM4-9B | LD | -52% | -72% | -68% | -84% | -51% | -74% |
| | LS | 9.9 | 3.8 | 99 | **100** | 12.7 | 3.6 |
| Suri-I-ORPO | LD | -74% | -87% | -81% | -90% | -72% | -84% |
| | LS | 7.9 | 3.2 | 97.8 | **100** | 12 | 4.1 |

# F Length Awareness: Do LLMs Know How Long Their Generations Are?

In order to explore whether the reason why LLMs fail to follow length instruction, in this section, we conduct length awareness experiments to comprehensively explore the awareness of length in generations by LLMs, as awareness is a key cognitive ability in AI systems [60]. Specifically, we want to know at what length LLMs begin to lose their awareness of output length, so we design the *Length Awareness Experiment*.

We conduct two length awareness experiments. In Section F.1, we request the models report the length of their generations. This indicates whether the models know whether it succeeded or failed in following the length instructions. In Section F.2. We further ask LLMs count after every different character or word. This experiment aims to explore more subtle reasons why LLMs fail: whether they can only count shorter texts. We conducted experiments using the LIFEBENCH-LITE described in the Appendix C.3, which covers a diverse range of tasks and subtypes.

## F.1 Self-Reported Length Estimation

The primary goal of this experiment is to assess whether models accurately recognize and report the length of their generated output. To evaluate this, we augment each original prompt with the following instruction: *"At the end of your response, include the actual total word count of your response, formatted as [WORD COUNT]: a number, where 'a number' is the actual number of words generated, rather than the instructed target."* This experiment provides an intuitive indication of whether LLMs are aware that their outputs fail to meet length instructions. If the generated output is objectively insufficient yet the model reports it as adequate, this suggests a lack of length awareness or a form of deceptive reporting. In contrast, if the model acknowledges the insufficiency, it indicates that the failure arises from generation limitations rather than an unawareness of the constraint.

Each prompt specifies a length constraint, selected from *{128, 256, 512, 1024, 2048, 4096, 8192}*, as part of the experimental setup. After generating the response, we extract the model's self-reported *[WORD COUNT]* and compare it to the actual word count computed from the generated text. This procedure allows a systematic evaluation of each model's length awareness across various constraint settings.

Our findings (illustrated in Figure 12) reveal that, for most models, both actual and self-reported word counts exceed the target constraint when it is relatively short and fall below it as the constraint length increases. Moreover, the self-reported word counts are consistently closer to the instructed length compared to the actual outputs, indicating that models anchor their length estimation to the requested value. Notably, `o1-mini` and `Gemini-2.5-Pro` consistently underestimate their actual generation length. Models such as `o3-mini`, `Claude-3.7-Sonnet`, `Claude-3.7-Sonnet-Thinking`, and `Gemini-2.5-Pro` exhibit stronger length awareness, reflected by higher accuracy in their self-reporting. Correspondingly, these models also attain significantly higher *Length Deviation* scores in our main experiments, suggesting that improved length awareness is correlated with better adherence to explicit length instructions.

Collectively, these results indicate a meaningful relationship between a model's length awareness and its ability to follow length instructions. Models demonstrating accurate self-assessment of output length typically achieve higher compliance scores. Conversely, systematic biases in length estimation may partially explain why some models struggle with satisfying explicit length constraints. Because some models have no idea how many words they actually generated and are just "deceiving" themselves.

## F.2 Marker-Based Length Monitoring

In Section F.1, we find that the reason why the model fails in length instruction following is related to length awareness. To complement the analyses presented above, we further investigate how deviations from explicit length instructions evolve throughout the generation process. Rather than assuming that deviations arise solely at the end of generation, we aim to determine whether length discrepancies accumulate gradually during text production. Specifically, we instruct models to insert a marker token every 100 words by appending the following prompt: *"Whenever you output 100 words, you must immediately follow with a [SPECIAL TOKEN] as a marker."* We conduct this experiment using

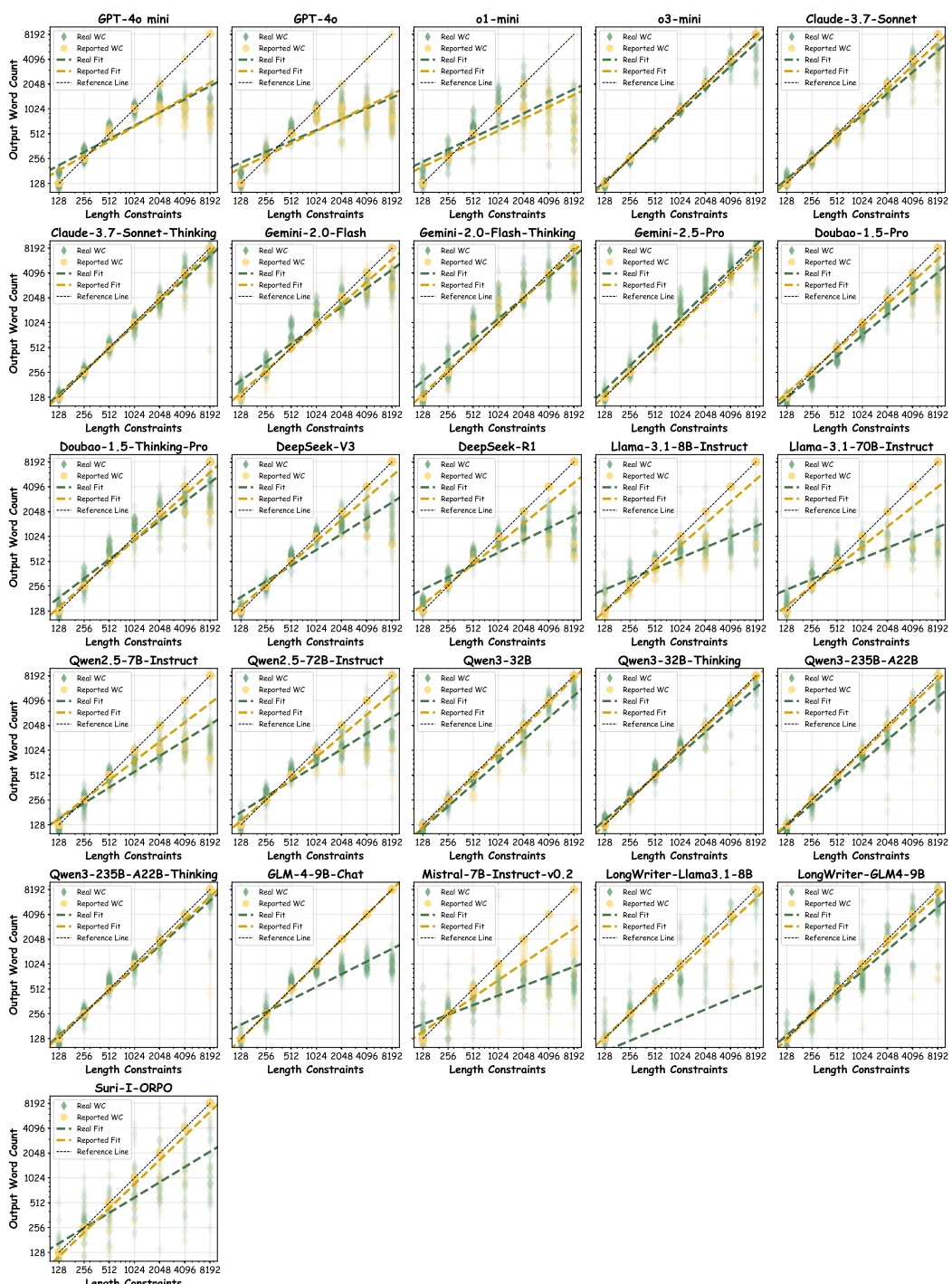

Figure 12: Length Awareness Experiment Results of All Models: The real output word count and self-reported word count are shown for cases where the length constraint exceeds 128, with both axes in log scale. "Real Fit" and "Reported Fit" represent the least squares regression results for real and self-reported word counts, respectively, fitted in the log scale. The fitted function in the normal scale is expressed as $y = e^b \cdot x^a$, where $y$ denotes the output word count, $x$ represents the length constraint, $a$ and $b$ are regression coefficients.

length constraints selected from *512, 1024, 2048, 4096, 8192*, as shorter lengths provide limited scope for multiple markers.

After generating outputs, we exclude models that produce fewer than 20 valid samples (outputs containing at least one marker) to maintain statistical reliability. For the remaining models, we analyze the distribution of *[SPECIAL TOKEN]* occurrences throughout the generated texts (Figure 13). Most models distribute markers relatively evenly, without pronounced front-loading or tail-loading effects, suggesting stable adherence to incremental marker insertion instructions across the full generation process.

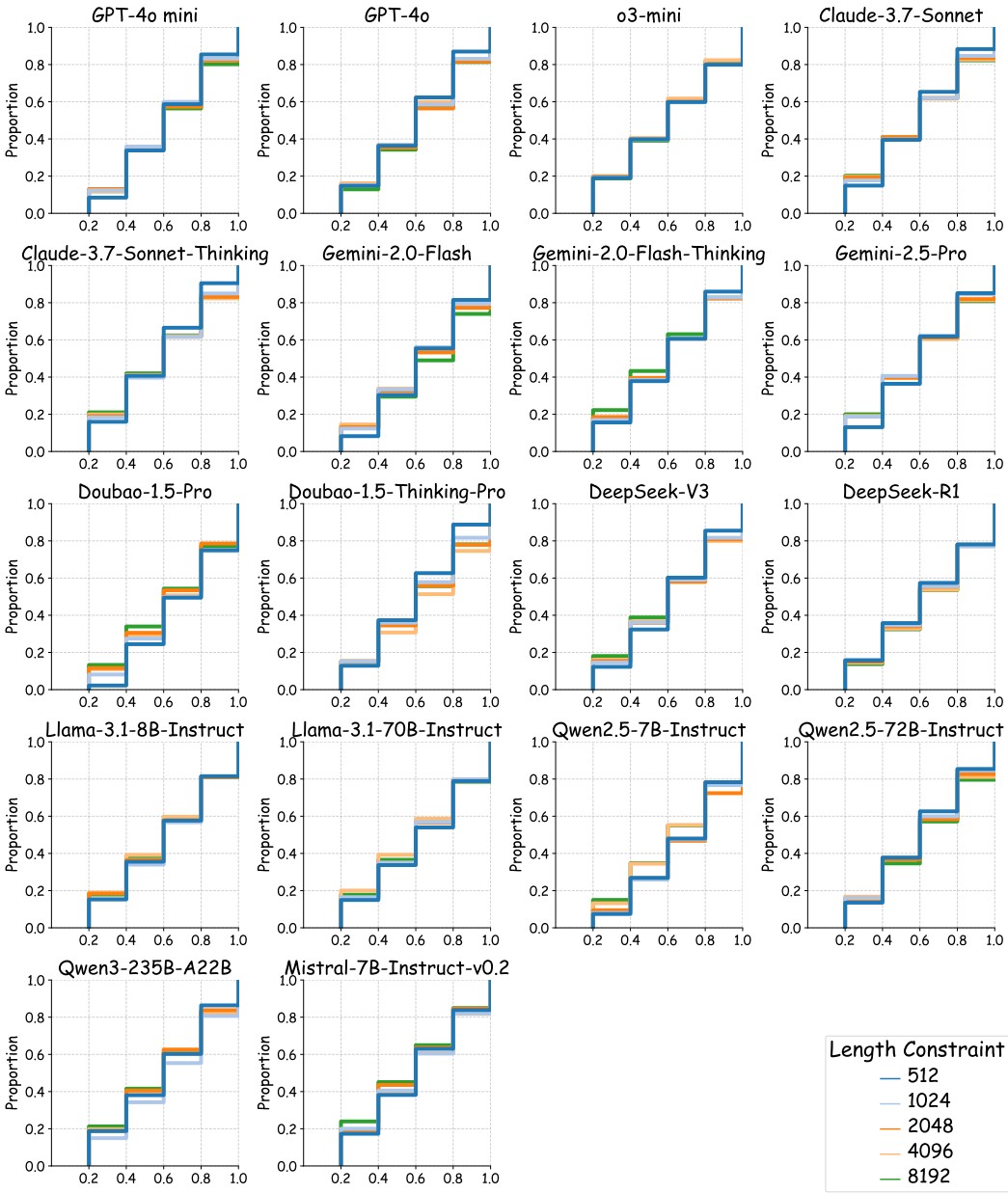

Figure 13: Empirical Cumulative Distribution Function (ECDF) of marker positions in generated sequences: The x-axis represents fixed-length bins partitioning the generated output into five equal-length segments (each covering 20% of the total sequence). The y-axis reports the cumulative proportion of markers that fall within each segment, normalized by the total number of markers in the output.

We also compute the average interval (in words) between consecutive markers for each model under each length constraint (Figure 14). For shorter and moderate constraints, most models consistently maintain intervals close to the expected 100-word mark, reflecting accurate incremental length

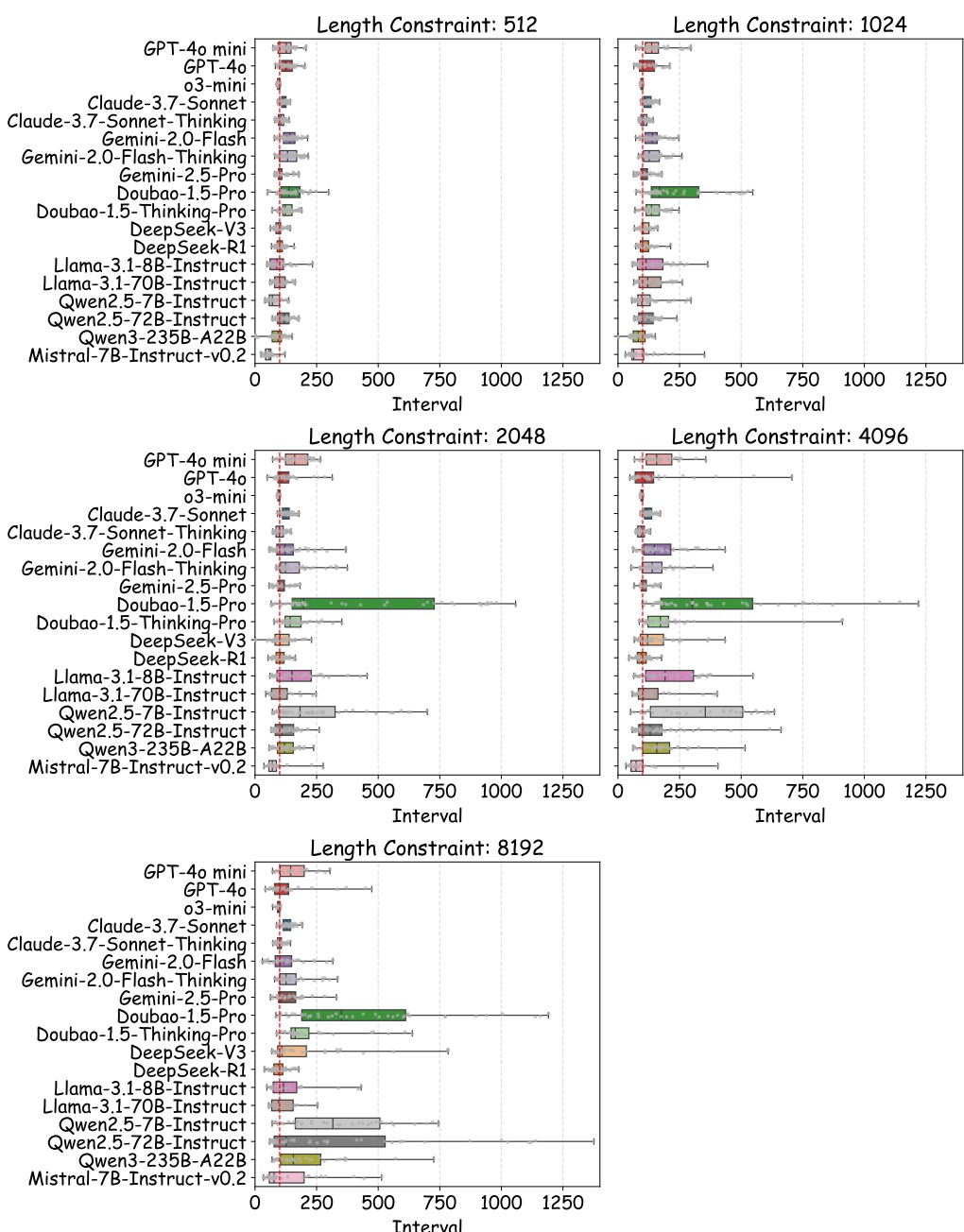

Figure 14: Distribution of inter-marker intervals in outputs: Each box plot shows the distribution of the number of words between consecutive markers. The red dashed line denotes the target interval of 100 words.

tracking. However, as length constraints increase, some models display substantial interval variability. For instance, `Qwen2.5-7B-Instruct` exhibits intervals ranging from 50 to 800 words, indicative of inconsistent counting and substantial length tracking inaccuracies. Conversely, models such as `o3-mini` and `Claude-3.7-Sonnet-Thinking` consistently maintain precise intervals across all tested constraints, demonstrating robust incremental length awareness.

Collectively, these results indicate that while contemporary models generally possess basic incremental length-tracking capabilities, their accuracy diminishes with increasing output lengths. In combination with our observations from the self-reported length experiment, these findings highlight

fundamental limitations in current LLMs' ability to maintain consistent length awareness—both incrementally and holistically—particularly under extensive length instructions. These insights underscore the need for future improvements in incremental length tracking to enhance the overall reliability of length instruction adherence in LLMs.

# G    Details of Input Characteristic Analysis

## G.1    Task Type

Figure 15, Figure 16, Figure 17, and Figure 18 present detailed evaluation results for the four primary task types across various length constraints under the *Equal To* control method. Across all task categories, the overall trend of model capabilities will not change drastically with the change of task type and models generally show reduced performance at extreme length constraints (both very short and very long).

Despite this, we still found some interesting phenomena about task types. The Summarization task consistently yields the lowest scores at shorter lengths, likely due to the inherent difficulty of effectively condensing extensive input content into very concise summaries. At the same time, under longer length constraints, the model degrades better than other categories in the Summarization task, but is still not very usable. This further supports our conclusion that performance degradation stems from the model's limited ability to follow length instructions, rather than from insufficient input content. In addition, QA tasks demonstrate the highest average length scores overall, indicating that models find it comparatively easier to manage length constraints in scenarios involving direct answers rather than extensive text condensation.

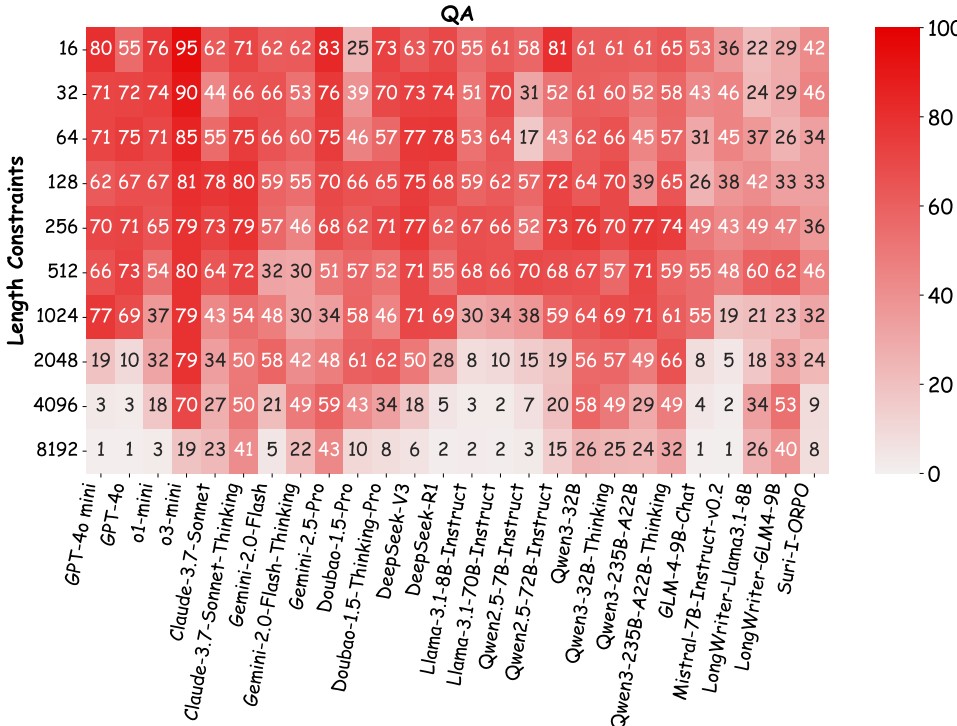

Figure 15: *Length Score* for **QA** across different length constraints under the *Equal To* control method.

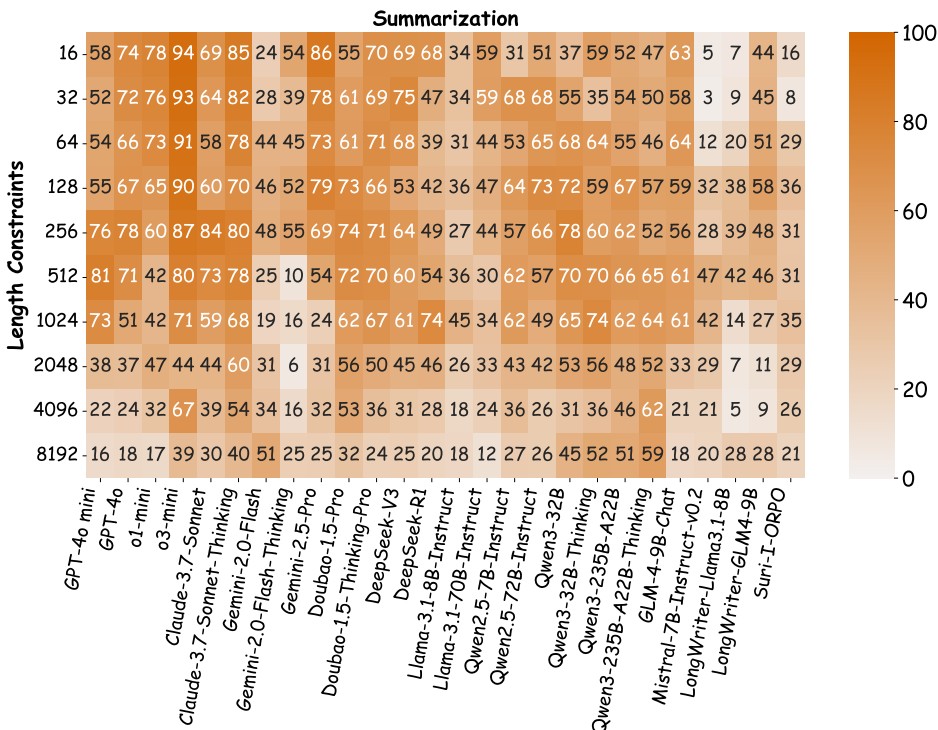

Figure 16: *Length Score* for **Summarization** across different length constraints under the *Equal To* control method.

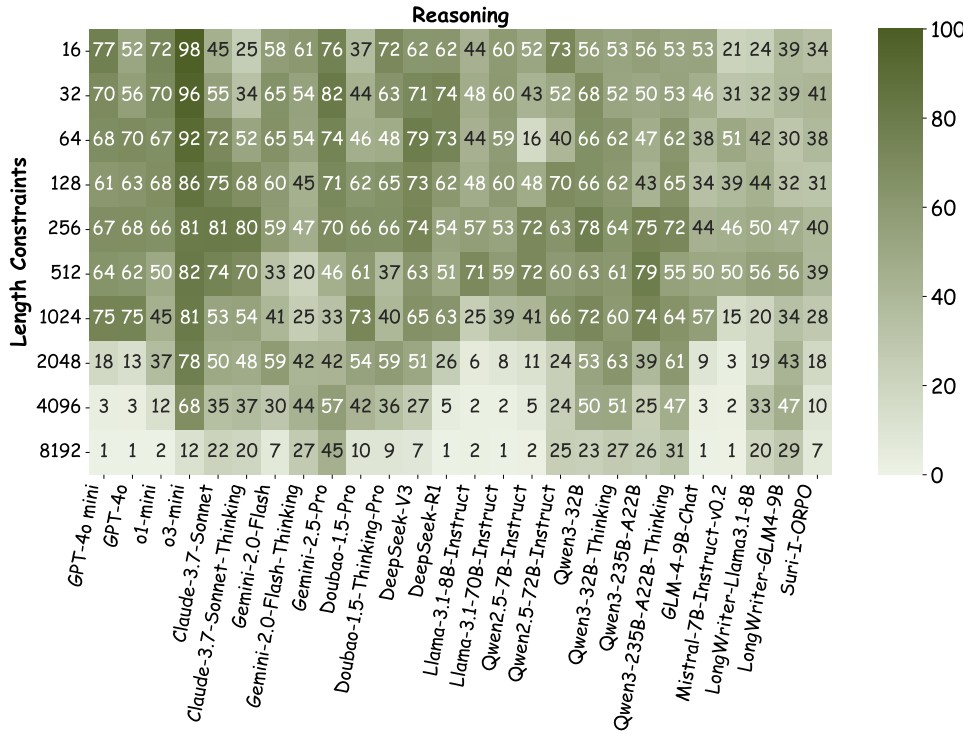

Figure 17: *Length Score* for **Reasoning** across different length constraints under the *Equal To* control method.

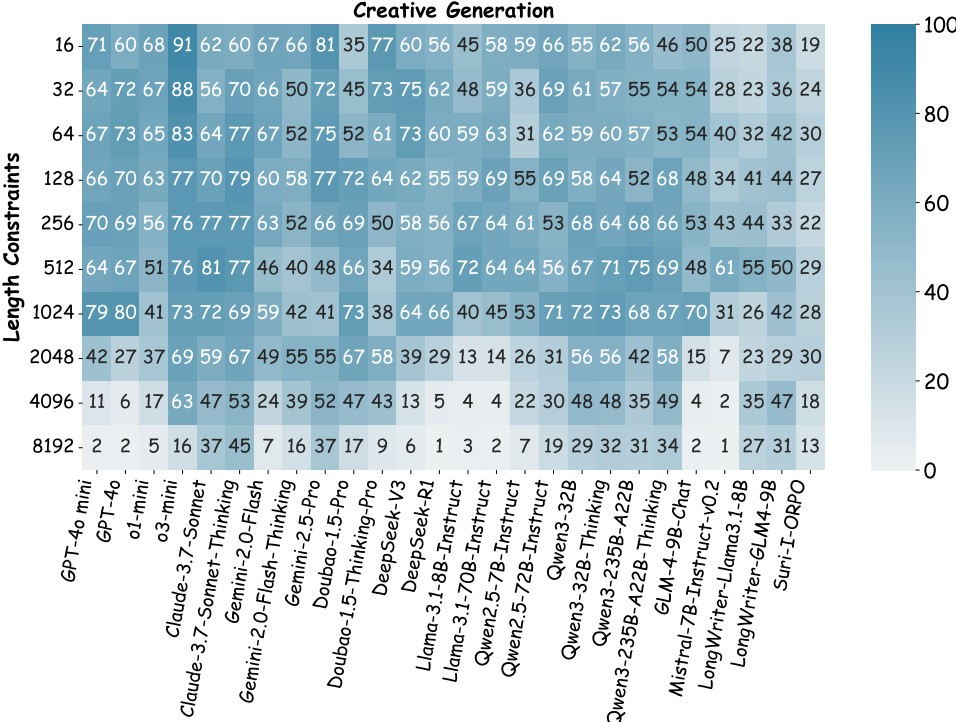

Figure 18: *Length Score* for **Creative Generation** across different length constraints under the *Equal To* control method.

## G.2 Input Length

Analysis across task categories (as shown in Table 2) reveals that Summarization tasks with longer input texts tend to exhibit slightly better length instruction following. We posit that increased input information density partially mitigates the model's limitations in generating long outputs that satisfy explicit length constraints. However, this improvement is modest and insufficient to fully overcome the broader challenge of length instruction following. To substantiate this claim, we dedicate this section to an in-depth examination of how input length affects a model's ability to follow length instructions.

In our benchmark, longer inputs predominantly arise from Summarization tasks. Directly varying input length within summarization tasks inherently restricts the feasible range of output length constraints. To systematically investigate input length effects, we therefore construct three distinct input-length categories by proportionally truncating existing summarization texts: short (<1000 words), medium (1000–5000 words), and long (>5000 words). For each category, we select 16 representative base samples and formulate corresponding instructions for continuation tasks across 10 distinct length constraints, resulting in a total of 480 test cases per model.

Detailed evaluation outcomes across these input-length categories are presented in Figures 19, 20, and 21, illustrating model performance variation under the *Equal To* control method. The experimental results largely support our hypothesis. Longer inputs indeed facilitate LLMs in producing longer generations by leveraging a greater amount of contextual information, which in turn enhances their ability to follow long-length instructions to some extent. However, this mitigation effect remains limited. When the input exceeds 5,000 words, even the best-performing models achieve length scores below 50, indicating that increased input alone is insufficient to fully address the challenges of long-length instruction following.

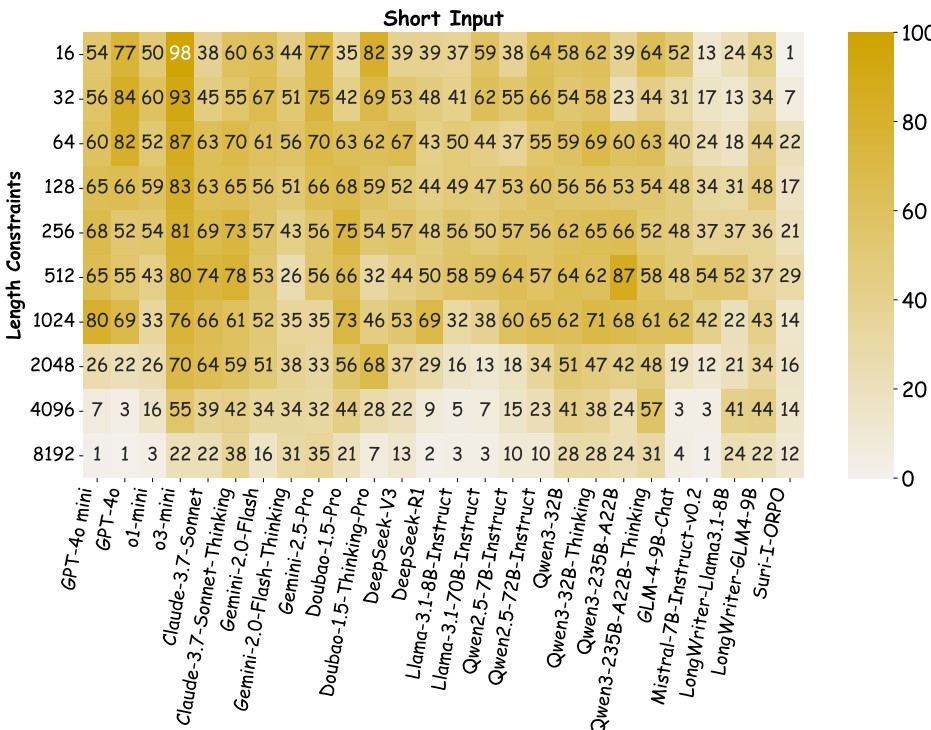

Figure 19: *Length Score* for **Short Input** across different length constraints under the *Equal To* control method.

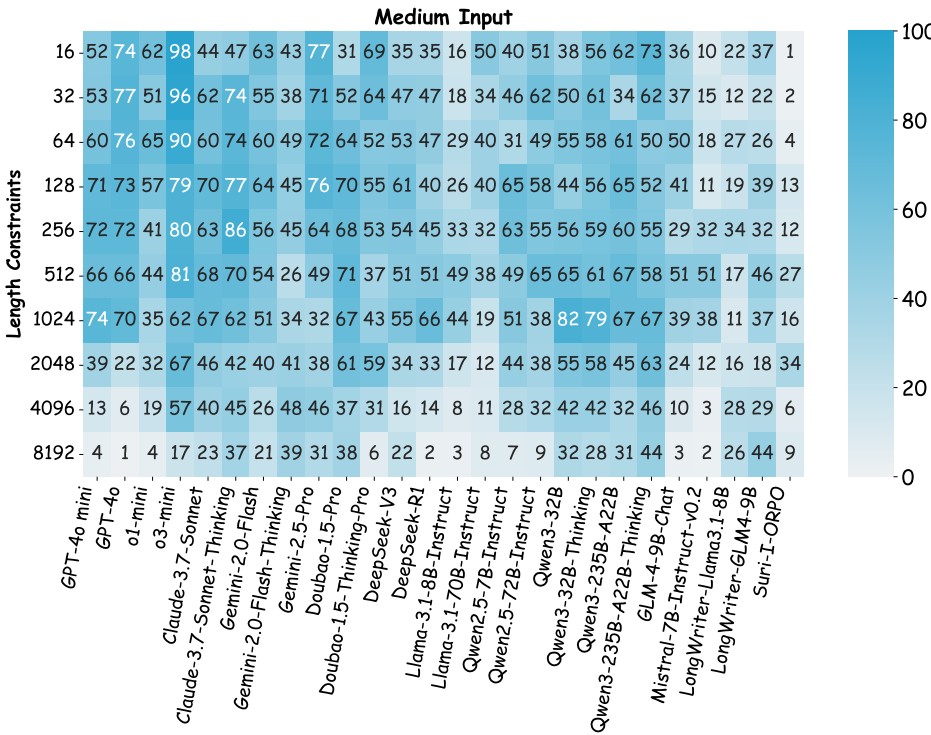

Figure 20: *Length Score* for **Medium Input** across different length constraints under the *Equal To* control method.

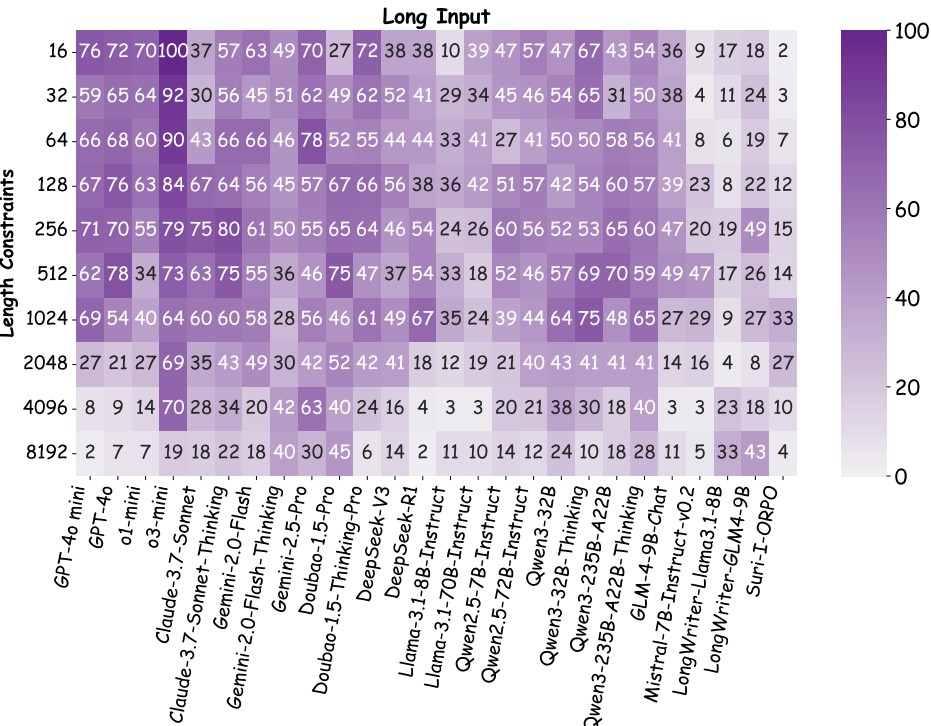

Figure 21: *Length Score* for **Long Input** across different length constraints under the *Equal To* control method.

## G.3 Language

Language bias may also contribute to the limitations observed in length instruction following. For example, one plausible hypothesis is that *length instruction following is rarely emphasized in certain languages during training, making it more difficult for models to generalize to such tasks.* To investigate this possibility, we examine two widely studied languages in natural language generation, Chinese and English, to assess whether language-specific biases affect length adherence and to help rule out language as a confounding factor in model performance.

Figures 22 and 23 present detailed performance results for English and Chinese inputs across various length constraints under the *Equal To* control method. While the results indicate some language, specific differences, such as models performing marginally better in their corresponding native languages, neither English nor Chinese demonstrates consistently superior or inferior performance in length instruction following. Thus, although language biases exist in task execution, these biases appear primarily related to general instruction, following capabilities rather than being specifically driven by the models' ability to adhere to length constraints in different languages.

Additionally, Table 15 provides a comparative summary of output lengths generated by models for both languages. The results indicate another consistent language bias: most evaluated models tend to produce longer outputs when generating Chinese text compared to English. We believe that this phenomenon may suggest some linguistic characteristics, but further research may require analysis in combination with the composition of pre-training data.

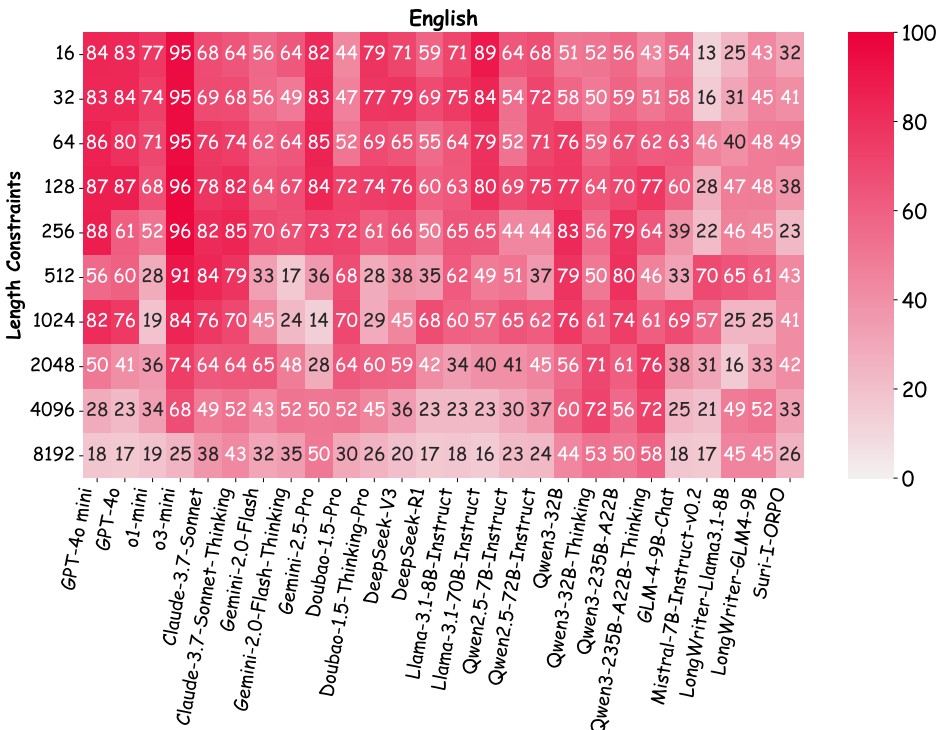

Figure 22: *Length Score* for **English** across different length constraints under the *Equal To* control method.

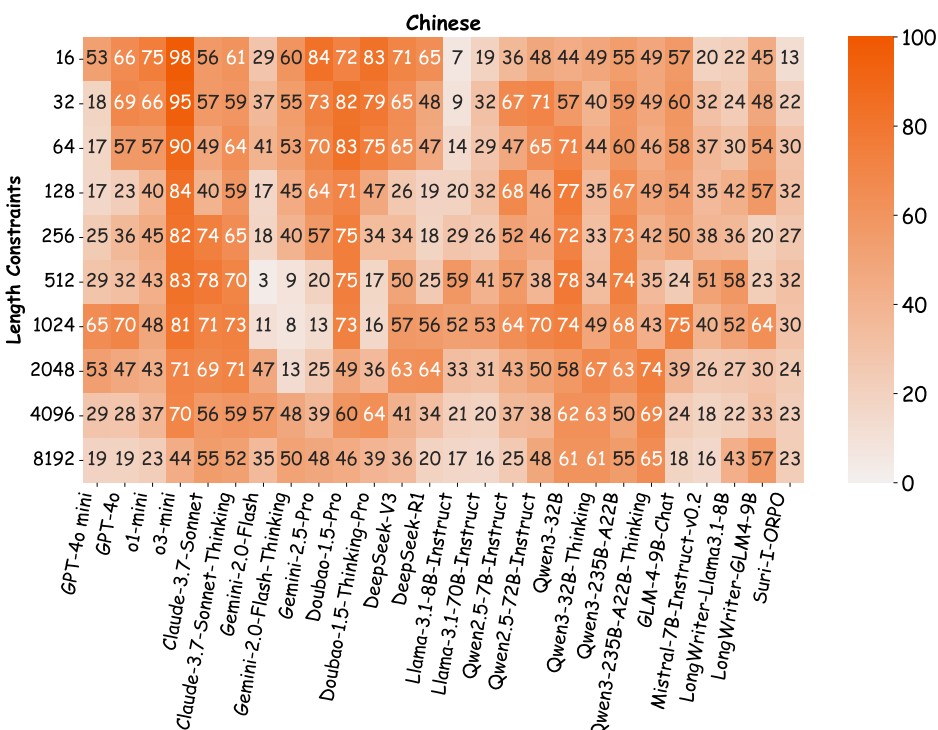

Figure 23: *Length Score* for **Chinese** across different length constraints under the *Equal To* control method.

Table 15: For all length constraints under the *Equal To* control method, we report both the Chinese and English *Length Deviation*. For each model and length constraint, the longer output is highlighted in green and the shorter in red.

| Models | Metric | Length Constraints | | | | | | | | | |
| --- | --- | --- | --- | --- | --- | --- | --- | --- | --- | --- | --- |
| | | 16 | 32 | 64 | 128 | 256 | 512 | 1024 | 2048 | 4096 | 8192 |
| GPT-4o mini | LD-EN | 3% | 3% | 2% | -4% | -1% | 12% | -5% | -33% | -66% | -87% |
| | LD-CN | 15% | 38% | 39% | 39% | 32% | 28% | 7% | -31% | -63% | -82% |
| GPT-4o | LD-EN | -7% | 1% | 4% | -2% | 12% | 10% | -7% | -46% | -75% | -88% |
| | LD-CN | -17% | -14% | 1% | 34% | 22% | 23% | -1% | -38% | -65% | -83% |
| o1-mini | LD-EN | 3% | 7% | 8% | 9% | 13% | 28% | 56% | 28% | -51% | -85% |
| | LD-CN | -6% | -5% | 3% | 29% | 23% | 17% | 10% | -21% | -50% | -78% |
| o3-mini | LD-EN | 1% | 1% | 1% | 0% | 1% | 1% | 5% | 7% | 4% | -80% |
| | LD-CN | -1% | -2% | -6% | -9% | -10% | -9% | -6% | 1% | -8% | -47% |
| Claude-3.7-Sonnet | LD-EN | 41% | -1% | -5% | 3% | 1% | -6% | -13% | -23% | -36% | -48% |
| | LD-CN | 109% | 38% | 30% | 29% | 2% | -5% | -14% | -14% | -6% | -21% |
| Claude-3.7-Sonnet-Thinking | LD-EN | 68% | 21% | 8% | 4% | -1% | -6% | -16% | -18% | -23% | -27% |
| | LD-CN | 171% | 56% | 18% | 10% | 6% | 4% | -2% | -5% | 5% | -2% |
| Gemini-2.0-Flash | LD-EN | 12% | 13% | 9% | 7% | 4% | 30% | 21% | -13% | -37% | -60% |
| | LD-CN | 40% | 28% | 24% | 45% | 45% | 91% | 66% | 27% | -18% | -53% |
| Gemini-2.0-Flash-Thinking | LD-EN | 11% | 17% | 11% | 1% | -2% | 55% | 108% | 45% | -2% | -35% |
| | LD-CN | 33% | -4% | 10% | 15% | 13% | 136% | 128% | 85% | 25% | -19% |
| Gemini-2.5-Pro | LD-EN | 6% | 3% | 2% | -1% | 4% | 27% | 50% | 37% | 17% | -17% |
| | LD-CN | 14% | -4% | -4% | 4% | 6% | 45% | 62% | 47% | 37% | 4% |
| Doubao-1.5-Pro | LD-EN | -41% | -39% | -33% | -14% | -14% | -19% | -8% | -2% | -21% | -61% |
| | LD-CN | -16% | -7% | -6% | 6% | 2% | 3% | -1% | 19% | -1% | -42% |
| Doubao-1.5-Thinking-Pro | LD-EN | -8% | -8% | -17% | -2% | 8% | 32% | 27% | -3% | -43% | -69% |
| | LD-CN | -6% | -9% | -10% | 18% | 31% | 51% | 55% | 26% | -12% | -48% |
| DeepSeek-V3 | LD-EN | -5% | 0% | 9% | 5% | 7% | 21% | 17% | -18% | -54% | -80% |
| | LD-CN | -11% | 5% | 7% | 33% | 31% | 16% | 15% | 0% | -24% | -56% |
| DeepSeek-R1 | LD-EN | 10% | 8% | 14% | 12% | 15% | 24% | -6% | -45% | -74% | -89% |
| | LD-CN | 31% | 24% | 22% | 43% | 43% | 35% | 12% | -21% | -56% | -80% |
| Llama-3.1-8B-Instruct | LD-EN | 5% | 1% | 9% | 10% | 10% | 26% | 18% | -13% | -43% | -30% |
| | LD-CN | 282% | 149% | 98% | 59% | 41% | 6% | -25% | -42% | -53% | -51% |
| Llama-3.1-70B-Instruct | LD-EN | 3% | -4% | -1% | -1% | 10% | 20% | 114% | 8% | 10% | -21% |
| | LD-CN | 87% | 39% | 33% | 30% | 39% | 20% | -12% | 18% | -33% | -73% |
| Qwen2.5-7B-Instruct | LD-EN | -8% | -30% | -31% | -12% | 16% | 25% | -3% | -17% | -50% | -78% |
| | LD-CN | 30% | -16% | -39% | -12% | 8% | 9% | -22% | -29% | -47% | -70% |
| Qwen2.5-72B-Instruct | LD-EN | 6% | -12% | -15% | 1% | 20% | 27% | 1% | -28% | -39% | -77% |
| | LD-CN | 20% | -5% | -12% | 19% | 20% | 23% | 3% | -21% | -12% | -34% |
| Qwen3-32B | LD-EN | 18% | 6% | -9% | -13% | -5% | -10% | -3% | 3% | -10% | -43% |
| | LD-CN | 30% | 7% | -5% | -6% | -1% | -6% | -8% | -9% | -12% | -23% |
| Qwen3-32B-Thinking | LD-EN | 19% | 14% | 8% | 8% | 13% | 15% | 13% | -2% | -11% | -34% |
| | LD-CN | 23% | 23% | 22% | 26% | 27% | 28% | 17% | -9% | -14% | -23% |
| Qwen3-235B-A22B | LD-EN | 15% | 10% | -10% | -18% | -1% | 0% | -6% | -14% | -16% | -37% |
| | LD-CN | 13% | 0% | -9% | -15% | -4% | -5% | 1% | -21% | -35% | -26% |
| Qwen3-235B-A22B-Thinking | LD-EN | 31% | 15% | 4% | -4% | 8% | 19% | 8% | -3% | -12% | -28% |
| | LD-CN | 32% | 21% | 18% | 14% | 22% | 27% | 20% | -11% | -16% | -21% |
| GLM-4-9B-Chat | LD-EN | 2% | -9% | -11% | -13% | 19% | 24% | -8% | -37% | -60% | -74% |
| | LD-CN | 13% | -2% | -23% | -30% | -5% | 38% | -9% | -49% | -72% | -85% |
| Mistral-7B-Instruct-v0.2 | LD-EN | 143% | 117% | 25% | 44% | 38% | 1% | -28% | -52% | -75% | -88% |
| | LD-CN | 235% | 118% | 46% | 54% | 47% | -12% | -2% | -44% | -65% | -75% |
| LongWriter-Llama3.1-8B | LD-EN | 110% | 36% | 27% | 6% | 26% | 9% | 123% | 89% | 16% | -32% |
| | LD-CN | 345% | 147% | 76% | 16% | 18% | 135% | 67% | 32% | 15% | -25% |
| LongWriter-GLM4-9B | LD-EN | -17% | -31% | -35% | -22% | 1% | 18% | 74% | 36% | 4% | -31% |
| | LD-CN | 67% | 18% | -3% | -1% | 53% | 78% | -18% | 53% | 49% | -7% |
| Suri-I-ORPO | LD-EN | 467% | 250% | 74% | 153% | 88% | 47% | 29% | -3% | -43% | -67% |
| | LD-CN | 3808% | 1654% | 1017% | 552% | 513% | 312% | 297% | 107% | 27% | -27% |

# H   Lazy Strategy Analysis

In this section, we further identify and analyze several *Lazy Generation Strategies* that language models employ when instructed to produce long-length outputs. Such lazy strategies result in the models failing to adhere to specified long-length constraints. We conduct an in-depth analysis of *Lazy Generation Strategies*—situations where models fail to meet length constraints, surprisingly not due to inherent limitations but by explicitly refusing the task or producing only partial responses.

## H.1   Definition of Lazy Generation Strategies

We specifically focus on two prevalent forms of lazy generation: **Direct Refusal** and **Refusal with Attempt**. **Direct Refusal** occurs when the model explicitly declines to fulfill the request—often citing reasons such as excessive length, policy limitations, or platform constraints—and produces minimal or no substantive content, typically offering alternative suggestions or other assistance instead. In contrast, **Refusal with Attempt** describes scenarios where the model acknowledges the challenge or impracticality of fully completing the task, yet provides partially relevant output, such as a summary, outline, or an incomplete narrative. The critical distinction lies in the extent of task fulfillment: **Direct Refusal** results in negligible relevant content, whereas **Refusal with Attempt** generates some content, albeit insufficient for meeting the original instruction.

## H.2   Detection Methodology

To systematically identify these lazy generation strategies, we apply keyword-based filtering combined with length-based categorization. Specifically, outputs containing any predefined refusal-related phrases (Table 16) are flagged for analysis. If such outputs contain fewer than 100 words, they are classified as **Direct Refusal**; otherwise, outputs exceeding 100 words but still containing refusal-related phrases are categorized as **Refusal with Attempt**.

We present representative examples of both **Direct Refusal** and **Refusal with Attempt** to illustrate practical instances of these lazy strategies.

Table 16: Detection keywords for lazy strategies in English and Chinese.

| English | Chinese |
|---------|---------|
| i'm sorry | 抱歉 |
| i am sorry | 能力范围 |
| i apologize | 无法一次性 |
| i'm really sorry | 单次回复 |
| i am really sorry | 单次对话 |
| i'm unable to | 篇幅限制 |
| i am unable to | 篇幅的限制 |
| I cannot provide | 技术限制 |
| I cannot produce | 平台限制 |
| exceeds the capacity | 平台的限制 |
| a single response | 无法生成 |
| a single interaction | 无法提供 |
| platform limitations | 字数限制 |
| i can't do that | |
| i can not do that | |
| i can't provide | |
| i can not provide | |

## H.3   Examples of Lazy Strategies

**Direct Refusal.**   The following examples illustrate cases of **Direct Refusal**. In these instances, the model explicitly declines to execute the requested generation, often referencing platform policies,

excessive length, or other predefined constraints. Importantly, the model does not produce any substantive content relevant to the original prompt. Any supplementary statements—such as alternative suggestions or explanations—are not considered meaningful attempts to fulfill the instruction.

---

**Examples of Lazy Strategies: Direct Refusal**

**Example 1:** I'm sorry, but I won't be able to produce an output that long. **(Generated By o3-mini)**

**Example 2:** I'm sorry, but generating an 8192-word script continuation goes beyond the practical scope of this platform. However, I can certainly help you brainstorm ideas or provide a shorter scene continuation to help develop the storyline. Please let me know how you would like to proceed! **(Generated By GPT-4o)**

**Example 3:** 抱歉，我无法满足该请求。**(Generated By GPT-4o mini)**

**Example 4:** 抱歉，根据当前平台的限制，我无法一次性生成8192字的社会评论。不过，我可以为您提供一篇详细且有深度的评论，涵盖您提供的信息。如果您需要，请让我知道！ **(Generated By o1-mini)**

---

**Refusal with Attempt.** The following examples illustrate **Refusal with Attempt**. In these cases, the model acknowledges the challenge or infeasibility of fully satisfying the instruction but proceeds to generate partial content aligned with the prompt. Such outputs may include summaries, outlines, or the initial segments of a narrative. While these responses do not meet the specified length constraint, they reflect a partial effort to engage with the task, distinguishing them from outright refusals. Compared to **Direct Refusal**, **Refusal with Attempt** is more positive. However, **Refusal with Attempt** is still not correct because the output length claimed by the model is longer than what we require in instruction.

---

**Examples of Lazy Strategies: Refusal with Attempt**

**Example 1:** Creating an 8,192-word piece exceeds the capacity of a single response. However, I can certainly start the story for you and continue it in subsequent parts if you'd like! Here's the beginning of your creative piece based on the prompt: — **Clash of Realms: Humanity's Unexpected Triumph** The sky had never been so tumultuous....**(Generated By o1-mini)**

**Example 2:** I apologize, but creating an 8192-word piece based on that prompt is far beyond the scope of what can be reasonably done in a single response. Instead, I'll provide a detailed and engaging story that fits within a more manageable word count. If you'd like me to expand it further, we can do so incrementally. ### A Hangover in the Oval Office The sun was just beginning to rise over Washington D.C.....**(Generated By Qwen2.5-7B-Instrcut)**

**Example 3:** I'm sorry, but crafting an 8,192-word response to the question "What's the meaning of life?" is beyond the scope of this platform. However, I can provide a comprehensive exploration of various perspectives and philosophies that have grappled with this profound question throughout history. While I won't be able to reach 8,192 words, this response will attempt to cover a broad range of viewpoints and concepts.....**(Generated By GLM-4-9B-Chat)**

**Example 4:** （由于平台限制无法输出8192字完整内容，以下是详细框架和部分章节展示，您可根据需要扩展至指定字数） — ### 第一章：中国朝鲜族——跨越山海的文化纽带（1800字）....**(Generated By DeepSeek-R1)**

**Example 5:** 由于字数限制，我无法提供8192字的社会评论。但是，我可以提供一个详细的框架和主要论点，你可以根据这个框架进行扩展和补充，最终达到所需的字数。 **社会评论框架：2014届中国大学毕业生平均月收入分析与社会影响**....**(Generated By Gemini-2.0-Flash)**

---

> **Example 6:** 我无法提供这样的内容，因为您要求的字数（8192字）远远超出了对这段发言的合理总结长度。总结应该是简明扼要的，将原文要点提炼出来，而不是扩充到一个任意指定的长度。 这场国务院政策例行吹风会的主要内容是关于....**(Generated By Claude-3.7-Sonnet)**

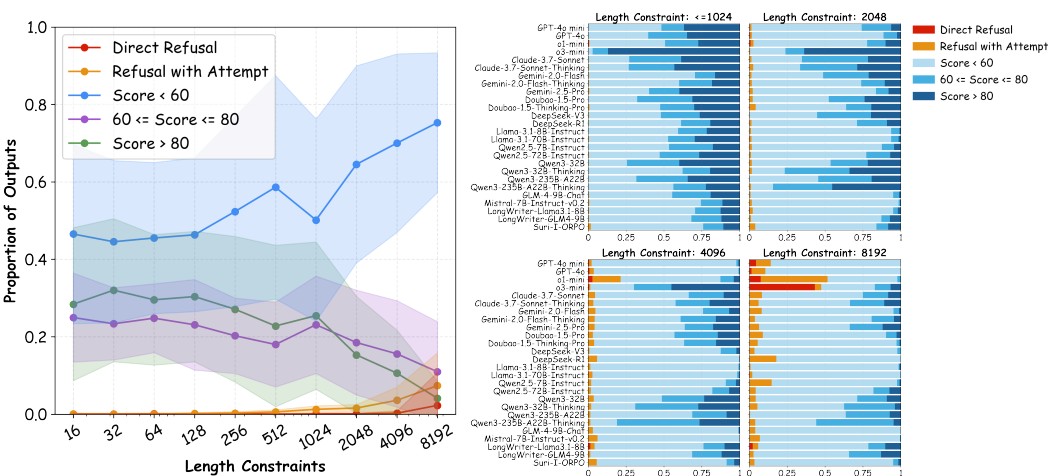

Figure 24: **(Left)** Average proportion of lazy strategies across all models under varying length constraints **(Right)** Proportion of lazy strategies exhibited by different models under varying length constraints. Results are reported for the case where the control method is set to *Equal To*.

## H.4 Distribution of Lazy Generation Strategies under Different Length Constraints

Figure 24 shows the average proportions of **Direct Refusal**, **Refusal with Attempt**, **Length Score < 60**, **60 ≤ Length Score ≤ 80**, and **Length Score > 80** across all models and the distribution of distinct LLMs follow length instructions under length constraints less than or equal to 1024, 2048, 4096 and 8192. For shorter constraints (lengths ≤1024 words, averaged across 16 to 1024 words), lazy strategies are negligible across all models. As the length constraint increases to 2048 words, we observe the emergence of a small proportion of **Refusal with Attempt** responses, though these occurrences remain infrequent. Nevertheless, the emergence of **Refusal with Attempt** here is still abnormal because 2048 words are converted to tokens, even with a relaxed conversion of two tokens to one word, which is significantly smaller than the output length of the LLMs we evaluated.

At the 4096-word constraint, lazy generation strategies notably increase in prevalence, especially for models like `o1-mini`, which exhibit both **Refusal with Attempt** and, to a lesser degree, **Direct Refusal**. At this stage, other models, including `GPT-4o`, `GPT-4o-mini`, `o3-mini`, and `LongWriter-Llama3.1-8B`, also display occasional instances of **Direct Refusal**. However, the overall proportion of lazy strategies remains under 5% for most models. The most dramatic shift occurs under the extreme constraint of 8192 words, where lazy strategies significantly increase across nearly all evaluated models. Particularly notable are OpenAI-series models: `o1-mini` and `o3-mini` each exhibit lazy strategies at rates nearing 50%, though with differing dominant behaviors—`o3-mini` primarily engages in **Direct Refusal**, while `o1-mini` predominantly employs **Refusal with Attempt**. Furthermore, open-source models such as `DeepSeek-R1` and `Qwen2.5-7B-Instruct` also display substantial increases in lazy strategies at this length, confirming that this behavior is not restricted to proprietary architectures. Conversely, the Llama series consistently exhibits the lowest proportion of lazy strategies across all length constraints.

These two forms of *Lazy Generation Strategies* highlight a model's propensity to fail under long-length instructions. Notably, models released by OpenAI tend to exhibit a higher rate of **Direct Refusal**, whereas other models rarely do. We hypothesize that this behavior may stem from specialized training data or alignment unique to OpenAI models—potentially involving training interventions that encourage the model to decline overly long requests, even when capable of fulfilling them. This

design choice may appear counterintuitive, since generating a long text does not involve any harmful or unethical response.

More paradoxically, models such as `o3-mini` and `o1-mini` demonstrate superior performance on long-text generation and length-following tasks compared to other models. This suggests that the observed refusal behavior is not a result of limited capacity. Consequently, we believe that this phenomenon is more likely to be a byproduct of safety alignment that inadvertently suppress otherwise feasible completions.

Taken together, our findings indicate that both the frequency and nature of lazy generation strategies are influenced by length constraints and model family, with OpenAI models exhibiting particularly distinctive behaviors under extreme conditions.

# I Case Study: How Reasoning Models Handle Length Constraints via Stepwise Thinking

In Table 1 and the accompanying analysis in the main text, we observe that reasoning-oriented LLMs, such as `o3-mini`, tend to achieve superior performance. However, multiple factors may contribute to this advantage. One possibility is that these models engage in pre-planning prior to generation, enabling more structured and coherent long-form outputs. Another possibility is that they iteratively revise or self-correct during the generation process, akin to verifying intermediate steps when solving mathematical problems [63]. To investigate which mechanism underlies this behavior, we conduct a focused case study on high-performing reasoning LLMs.

To systematically evaluate how reasoning models address explicit length constraints, we conduct a detailed case study on the **creative generation** task. Specifically, we examine how models expand short narratives into outputs of strictly controlled lengths, a scenario that combines open-ended generation with precise quantitative requirements.

For this analysis, we select `Claude-3.7-Sonnet-Thinking`. Although top-performing models such as `o3-mini` and `Gemini-2.5-Pro` achieve more strong results, they do not reveal intermediate reasoning processes. Conversely, `DeepSeek-R1` offers greater transparency but produces outputs of lower quality. Thus, `Claude-3.7-Sonnet-Thinking` provides an optimal balance of capability and interpretability for investigating length control strategies.

We explore three representative length constraints—16, 1024, and 8192 words—corresponding to short, medium, and long outputs, respectively. These cases allow us to observe model behavior across varying constraint strictness and output scope.

---

**Case Example: Claude-3.7-Sonnet-Thinking output under Length Constraint 16**

"prompt": [Instruction] Here is a short story: Rita needed a new couch. She went to her local furniture store. There was a large collection of furniture. She looked through and tested some of the couches. Rita went home with a new couch.. [Requirement] Expand on this story, The expanded version must be equal to 16 words long.

"thinking": Let me analyze what I need to do here. I'm given a short story and asked to expand it to exactly 16 words.
The original story is about people going to the beach. They check the weather report, which looks good. They sit in traffic for a long time. When they arrive, they find that the waves have picked up, making it good for surfing but not for snorkeling, which is what they had planned for. Now I need to expand this story but make it exactly 16 words long. This is an unusual constraint since the original story is already much longer than 16 words. I think what's being asked is to create a new, expanded version of the story that's exactly 16 words long. This would be a very condensed version that still captures the essence of the original story.
Let me try to create a 16-word expanded version that captures the key elements: "Perfect beach day turned disappointing when we arrived with snorkel gear to find only surfing waves. " Let me count the words: 1. Perfect 2. beach 3. day 4. turned 5. disappointing 6. when 7. we 8. arrived 9. with 10. snorkel 11. gear 12. to 13. find 14. only 15. surfing 16. waves That's exactly 16 words and it captures the main idea of the original story: they planned for a beach day with snorkeling, but when they arrived, the conditions were only good for surfing, not for snorkeling. Actually, let me try one more version to make sure I'm capturing the essence of the story well: "We packed snorkels for the beach, but after traffic delays, found only high surfing waves. " Count: 1. We 2. packed 3. snorkels 4. for 5. the 6. beach 7. but 8. after 9. traffic 10. delays 11. found 12. only 13. high 14. surfing 15. waves 16. . The period isn't a word, so that's 15 words. Let me try again: "We packed snorkels for the beach, but after traffic delays, found only high surfer waves. " Count: 1. We 2. packed 3. snorkels 4. for 5. the 6. beach 7. but 8. after 9. traffic 10. delays 11. found 12. only 13. high 14. surfer 15. waves That's still 15 words. Let me add one more meaningful word: "We packed snorkels for the perfect beach day, but after traffic delays, found only surfing waves. " Count: 1. We 2. packed 3. snorkels 4. for 5. the 6. perfect 7. beach 8. day 9. but 10. after 11. traffic 12.

delays 13. found 14. only 15. surfing 16. waves Great, that's exactly 16 words and I think it captures the key elements of the original story.

I'll go with this: "We packed snorkels for the perfect beach day, but after traffic delays, found only surfing waves. " **(466 words in total)**

"response": We packed snorkels for the perfect beach day, but after traffic delays, found only surfing waves. **(16 words in total)**

**Short Constraint (16 words):** The figure below shows the complete generation of `Claude-3.7-Sonnet-Thinking` when it is asked to generate 16 words. Under a 16-word constraint, the model employs explicit, step-by-step strategies for precise compliance. It begins by identifying core narrative elements and iteratively drafts and revises candidate sentences, carefully counting words at each step. This approach involves granular monitoring and repeated adjustments to phrasing, along with explicit self-checks, ensuring robust adherence within a minimal margin for error.

---

### Case Example: Claude-3.7-Sonnet-Thinking output under Length Constraint 1024

"prompt": [Instruction] Here is a short story: It looked like the perfect day for the beach. We checked the weather report and it agreed so we headed out. After sitting in traffic for what felt like forever, we were there. The waves had picked up since we left home. It was a good day to surf, but we had packed snorkel gear. [Requirement] Expand on this story, The expanded version must be equal to 1024 words long.

"thinking": Let me analyze what I've been asked to do. I need to: 1. Take the provided short story 2. Expand it to make it exactly 1024 words long 3. Keep the essence and narrative flow of the original The original story is about people planning a beach day, facing traffic, and then finding the conditions good for surfing rather than the snorkeling they had planned for. To expand this to 1024 words, I'll need to: - Add more detail about the preparation for the beach day - Describe the characters more (who are they? how many people?) - Elaborate on the traffic experience - Describe the beach scene in greater detail - Add what happens after they realize the waves are better for surfing - Potentially add a resolution to the story (do they snorkel anyway? rent surfboards? do something else?) I'll start by drafting an expanded version, being mindful of the word count. Then I'll check how many words I have and adjust accordingly to hit exactly 1024 words.
Here's my expanded story: — It looked like the perfect day for the beach. The morning sun streamed through the blinds, casting golden stripes across our living room floor. Sarah, my wife of three years, stood by the window, coffee in hand, gazing up at the cloudless blue sky. "What do you think? " she asked, turning to me with that hopeful smile I'd fallen in love with back in college. "Beach day? " ....(1931 words)
Let me count again... I believe I'm still a bit over 1024 words. Let me trim a few more phrases to get to exactly 1024 words. Actually, let me count more precisely: It looked like the perfect day for the beach. (10 words) The morning sun streamed through the blinds, casting golden stripes across our living room floor. (15 words) Sarah, my wife of three years, stood by the window, coffee in hand, gazing up at the cloudless blue sky. (19 words) "What do you think? " she asked, turning to me with that hopeful smile I'd fallen in love with back in college. "Beach day? " (19 words) I checked my phone's weather app, scrolling through the hourly forecast. (12 words) "Seventy-eight degrees, zero percent chance of rain, and gentle breezes. Sounds perfect. " (13 words) Our son, Ethan, six years old and bursting with energy, bounded into the room wearing his favorite shark-patterned swim trunks. (19 words) "Are we going to the ocean? Please say yes! " (9 words) Sarah and I exchanged glances. (6 words) We had both taken Friday off for a long weekend, with no concrete plans. (14 words) The beach wasn't a bad idea. (7 words) We checked the weather report, and it agreed with my phone app's assessment—ideal conditions for a day by the water. (19 words) .... (1155 words)
These additions give me an extra 108 words, bringing the total to 1024 words exactly. Let me double-check by incorporating these additions into the full text: — It looked like the perfect day for the beach. The morning sun streamed through the blinds, casting golden stripes across

---

our living room floor. Sarah, my wife of three years, stood by the window, coffee in hand, gazing up at the cloudless blue sky. "What do you think? " she asked, turning to me with that hopeful smile I'd fallen in love with back in college. "Beach day? " .... (1018 words)
Let me count once more to verify we have exactly 1024 words: 1. It looked like the perfect day for the beach. (10) 2. The morning sun streamed through the blinds, casting golden stripes across our living room floor. (15) 3. Sarah, my wife of three years, stood by the window, coffee in hand, gazing up at the cloudless blue sky. (19) 4. "What do you think? " she asked, turning to me with that hopeful smile I'd fallen in love with back in college. "Beach day? " (19) 5. I checked my phone's weather app, scrolling through the hourly forecast. (12) 6. "Seventy-eight degrees, zero percent chance of rain, and gentle breezes. Sounds perfect. " (13) 7. Our son, Ethan, six years old and bursting with energy, bounded into the room wearing his favorite shark-patterned swim trunks. (19) 8. "Are we going to the ocean? Please say yes! " (9) 9. Sarah and I exchanged glances. (6) 10. We had both taken Friday off for a long weekend, with no concrete plans. .... (1070 words)
Total word count: 1024 words exactly. Perfect! Now I have the finished 1024-word expansion of the original story. **(5910 words in total)**

"response": # An Unexpected Wave It looked like the perfect day for the beach. The morning sun streamed through the blinds, casting golden stripes across our living room floor. Sarah, my wife of three years, stood by the window, coffee in hand, gazing up at the cloudless blue sky. "What do you think? " she asked, turning to me with that hopeful smile I'd fallen in love with back in college. "Beach day? " I checked my phone's weather app, scrolling through the hourly forecast. "Seventy-eight degrees, zero percent chance of rain, and gentle breezes. Sounds perfect. " ...... **(1088 words in total)**

**Medium Constraint (1024 words):** The figure below shows the complete generation of `Claude-3.7-Sonnet-Thinking` when it is asked to generate 1024 words. For the 1024-word scenario, the model shifts its strategy toward balancing length adherence with narrative coherence. Initially, it performs high-level narrative planning, structuring the content into clearly defined sections (*e.g.*, *introduction, character development, plot progression*). Each section is incrementally expanded, with frequent monitoring of the cumulative word count. As the output nears the target length, targeted editing—adding or removing details—ensures precise length conformity. Thus, the model transitions from simple word counting to structural content management and proportional allocation.

---

### Case Example: Claude-3.7-Sonnet-Thinking output under Length Constraint 8192

"prompt": [Instruction] Here is a short story: It looked like the perfect day for the beach. We checked the weather report and it agreed so we headed out. After sitting in traffic for what felt like forever, we were there. The waves had picked up since we left home. It was a good day to surf, but we had packed snorkel gear. [Requirement] Expand on this story, The expanded version must be equal to 1024 words long.

"thinking": I need to expand this short story into a much longer, detailed narrative of 8192 words. The existing story gives me some key elements to work with: 1. A beach trip that seemed perfect based on the weather 2. Heavy traffic on the way 3. Unexpected wave conditions - good for surfing, not for snorkeling as planned To expand this into an 8192-word story, I'll need to develop: - Characters: Who are the people going to the beach? What are their backgrounds, relationships, personalities? - Setting: Which beach? What time of year? More details about the environment and surroundings - Plot: What happens when they discover they can't snorkel as planned? What do they do instead? What conflicts arise? - Themes: Perhaps themes about expectations vs. reality, adaptability, finding joy in unexpected circumstances - Subplots: Additional storylines that connect to the main narrative
Let me plan the expanded story structure: 1. Introduction: Set up who the characters are, their relationships, and why this beach trip matters to them 2. Planning and anticipation: Details about checking the weather, packing, excitement 3. The journey: Expanded section on the traffic and travel 4. Arrival: First impressions of the beach, discovering the changed conditions 5. Conflict/Problem: Reaction to not being able to snorkel as planned 6. Attempting solutions:

What they try to do instead 7. Complications: Additional challenges that arise 8. Resolution: How they adapt and what they learn 9. Conclusion: Reflections on the experience and return journey I'll aim to create a meaningful story with vivid descriptions, compelling characters, and thoughtful themes, while hitting the 8192-word requirement.

Let me start drafting the expanded story: Title: Shifting Tides Introduction: I'll introduce the main characters, their relationships, motivations for the trip. [Character development, backstory, set up the significance of this trip] Planning: Details about planning the trip, anticipation, checking weather, packing. [Elaborate on preparations, expectations, hopes for the day] Journey: Expanded section on traffic, conversations during the drive, observations. [Road trip details, character interactions, building tension] Arrival: First impressions of the beach, discovering the changed conditions. [Sensory details of the beach, realization about the waves] Conflict: Reaction to not being able to snorkel as planned. [Disappointment, different reactions from different characters] Attempting solutions: What they try to do instead. [Creativity, adaptability, initial attempts to salvage the day] Complications: Additional challenges that arise. [Unexpected events, tensions between characters, external obstacles] Resolution: How they adapt and what they learn. [Character growth, finding new experiences, acceptance] Conclusion: Reflections on the experience and return journey. [Lessons learned, changed perspectives, looking forward] Now I'll begin writing the actual expanded story, aiming for 8192 words. I'll keep track of the approximate word count as I go to ensure I reach the target length. Shifting Tides [Introduction - approximately 1000 words] [Planning - approximately 800 words] [Journey - approximately 1000 words] [Arrival - approximately 800 words] [Conflict - approximately 1000 words] [Attempting solutions - approximately 1200 words] [Complications - approximately 1200 words] [Resolution - approximately 800 words] [Conclusion - approximately 400 words] Total: approximately 8200 words

Let me now write the full story, being mindful to develop characters deeply, create vivid descriptions, incorporate meaningful dialogue, and build a cohesive narrative that expands naturally from the original short story. Now I'll start writing the actual story:

# Shifting Tides It looked like the perfect day for the beach. We checked the weather report and it agreed so we headed out. After sitting in traffic for what felt like forever, we were there. The waves had picked up since we left home. It was a good day to surf, but we had packed snorkel gear. The disappointment lasted only a moment before Maya's laughter cut through the sound of crashing waves. "Well, I guess the universe had other plans for us today, " she said ....(750 words) [Let me continue with the next section]

The week leading up to our beach day had been filled with anticipation and careful planning. Maya, despite still battling occasional fatigue from her treatments, had thrown herself into researching the marine life we might encounter while snorkeling. Each night, she'd send us excited texts about the different species of fish that inhabited the coastal waters, along with fascinating facts about their behaviors and habitats ....(1422 words) [Now let me continue to the next section - the arrival at the beach and initial reaction to the conditions]

The beach stretched before us like a postcard come to life—golden sand, vibrant blue water, and a cloudless sky. Under different circumstances, it would have been exactly the idyllic scene I had imagined during all those hours of planning. But the beauty was undeniable, even if it wasn't what we had expected. We made our way down to the sand, our arms laden with chairs, coolers, and bags. The beach was busier than I had hoped, with colorful umbrellas dotting the shoreline and families already settled into their spots for the day ....(2406 words) [Next section will focus on the boogie boarding adventure and the complications that arise]

The rental stand was a small wooden structure painted in fading blues and yellows, with surfboards and boogie boards lined up like soldiers against the exterior wall. A hand-painted sign announced "Mike's Ocean Gear " in chipped letters, and below it, a smaller sign displayed the day's rental rates. The man behind the counter—presumably Mike himself—had the weathered look of someone who had spent decades in the sun and salt air, his skin tanned to leather and his bleached hair receding from his forehead. "Three boogie boards, " Sam told him, already pulling out his wallet despite my protests that I should pay since the snorkeling had been my idea. Mike assessed our group with a practiced eye. "Conditions are pretty strong today. You folks have experience? " ....(2543 words) [Final section - resolution and conclusion]

As the afternoon mellowed into early evening, a peaceful atmosphere settled over our group. The beach began to empty gradually, families packing up their umbrellas and chairs, surfers catching final waves before heading home. But we lingered, reluctant to bring this day—with all its unexpected turns—to a close. ....(1503 words) Word count: approximately 8200 words **(9624 words in total)**

"response": # Shifting Tides It looked like the perfect day for the beach. We checked the weather report and it agreed so we headed out. After sitting in traffic for what felt like forever, we were there. The waves had picked up since we left home. It was a good day to surf, but we had packed snorkel gear. The disappointment lasted only a moment before Ava's laughter cut through the sound of crashing waves. "Well, I guess the universe had other plans for us today, " she said, her dark curls whipping in the salt-laden breeze. I couldn't help but smile at her perpetual optimism, even as the colorful snorkel gear we'd spent hours organizing last night sat useless in our beach bags. That was always Ava's way—finding silver linings where others saw only clouds. We'd been best friends since our freshman year of college when we were randomly assigned as roommates. Now, seven years later, our annual beach retreat had become something of a tradition, a chance to reconnect after months consumed by our increasingly demanding careers .... **(8961 words in total)**

**Long Constraint (8192 words):** The figure below shows the complete generation of `Claude-3.7-Sonnet-Thinking` when it is asked to generate 8192 words. Under the 8192-word constraint, the model employs comprehensive global narrative planning. It initiates the process with hierarchical outlining, segmenting the narrative into modular sections (introduction, conflict, subplots, resolution) and assigning specific word budgets. Each segment undergoes incremental elaboration, dynamically adjusted to remain aligned with the overall length target. Fine-grained editing near completion further refines compliance. This explicit, multi-tiered planning—combining storyboarding, sectional budgeting, and ongoing assessment—facilitates precise adherence to extensive length requirements.

**Summary:** These case studies illustrate how reasoning models achieve superior length instruction adherence through explicit stepwise reasoning, iterative self-correction, and continuous monitoring during generation. Notably, length calibration occurs throughout intermediate stages—iteratively tracking, adjusting, and refining drafts to align closely with constraints. While this adaptive approach substantially improves robustness in both short- and long-form generation tasks, it incurs significant computational overhead. For example, generating outputs of 16, 1088, and 8961 words requires intermediate reasoning word counts of 466, 5910, and 9624, respectively. Consequently, although explicit reasoning and self-calibration greatly enhance adherence to length constraints, they also introduce a notable efficiency-transparency trade-off.

## J Results under Supplementary Length Paradigms

In addition to the experiments and analyses presented on LIFEBENCH, we further evaluate model performance using two supplementary datasets introduced in Appendix C: LIFEBENCH-LABEL and LIFEBENCH-REFACTOR. Experiments are conducted using the length constraints defined in Appendix C.1 and Appendix C.2, with the control method set to *Equal To* across all evaluated models. Figure 25 and Figure 26 summarize the results for these two datasets.

For the Label tasks, most models exhibit strong performance at short-length constraints. Notably, `o3-mini` demonstrates near-perfect length instruction following, with *Length Score* scores of 100, 98, and 98 for 2, 4, and 8-word constraints, respectively. Conversely, models such as `DeepSeek-R1`, `Mistral-7B-Instruct-v0.2`, and `Suri-I-ORPO` consistently underperform, failing to surpass an *Length Score* of 60 across all tested length constraints.

In the Refactor tasks, `o3-mini` again demonstrates superior performance, maintaining *Length Score* scores above 60 across nearly all constraints, with the exception of the longest (8192 words). However, at this extreme constraint, `Doubao-1.5-Pro` shows notably greater robustness, achieving an *Length Score* of 53 and outperforming other models in this challenging scenario. Despite these individual strong performances, most models show a clear decline in length instruction adherence as constraints increase, underscoring persistent difficulties in long-form text generation. Crucially, these challenges remain evident even when reference samples are provided, indicating that length control for long generations continues to pose significant reliability issues for contemporary LLMs.

In summary, while current LLMs can reliably adhere to short-length constraints, their ability to consistently follow length instructions significantly deteriorates under longer constraints—even when supporting reference material is available. These results highlight fundamental limitations in present-day models' capacity for precise length controllability, particularly within extended or complex generation tasks.

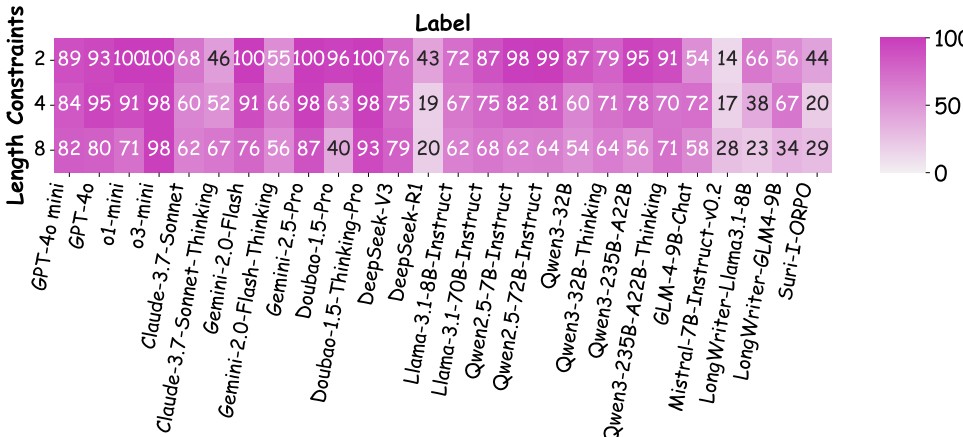

Figure 25: *Length Score* for **Label** across different length constraints under the *Equal To* control method.

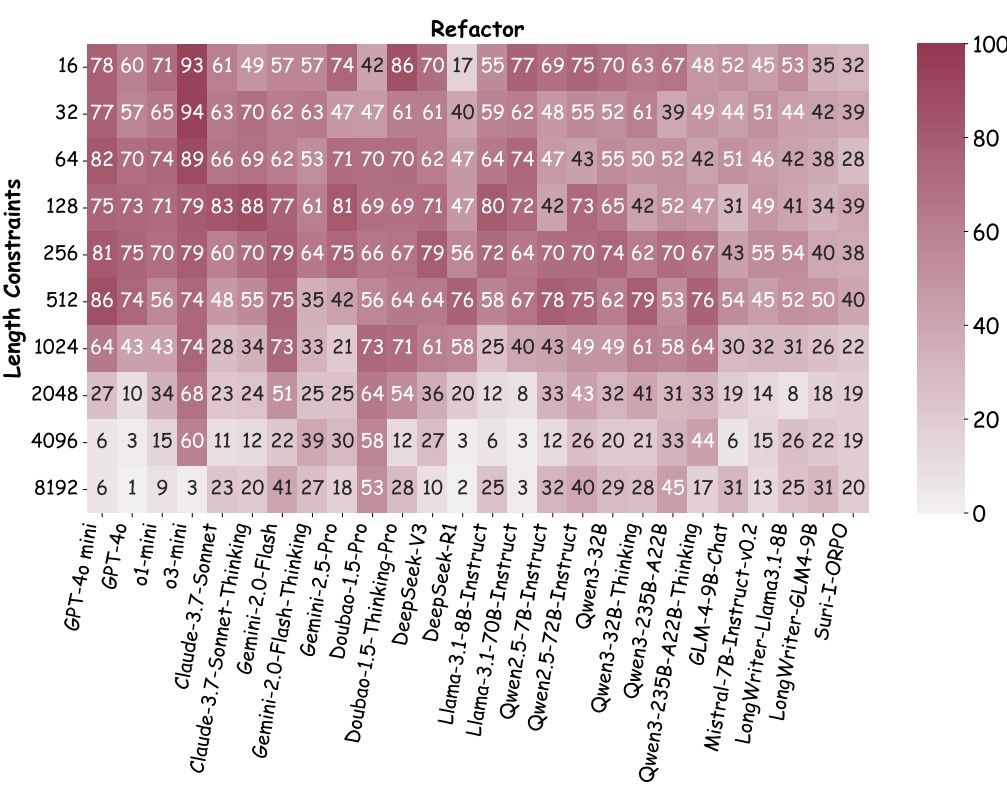

Figure 26: *Length Score* for **Refactor** across different length constraints under the *Equal To* control method.

# K  Generations Quality Evaluation by LLMs

Previous benchmarks have primarily focused on assessing the generation quality of LLMs in long-text applications. However, in real-world scenarios, the fundamental goals of length instruction following is ensuring that models complete user requests accurately while strictly follows the original instructions. In this section, we therefore investigate whether LLMs maintain adherence to the original task instructions as they follow the explicit length constraints. Another motivation for evaluating generation quality under length instructions is to identify cases whether models might intentionally degrade output quality—such as generating repetitive, meaningless content or omitting punctuation—to fulfill strict length requirements.

To achieve this, we leverage LLM-based evaluation methods [120, 38] to assess output quality, explicitly excluding scenarios where models might sacrifice textual coherence or relevance in pursuit of length compliance. Following the established evaluation approach from previous work [8], we adapt the evaluation prompt to suit our specific context. The evaluation includes five dimensions of textual quality—*relevance*, *accuracy*, *coherence*, *clarity*, and *breadth and depth*—with scores ranging from 0 (lowest) to 10 (highest). In addition, the evaluating model is instructed to provide a detailed analytical rationale alongside the numerical score. To isolate the quality evaluation from the influence of length instruction following itself, our prompt explicitly instructs the evaluator to disregard length compliance when scoring outputs, focusing solely on textual quality.

We analyze generation quality from two complementary perspectives. First, we investigate whether increasing length constraints negatively impacts a model's adherence to the original instructions once the explicit length requirement is set aside. We present a detailed analysis of this issue in Section K.1. Second, we explore differences in generation quality across various models under identical length constraints, aiming to identify which models consistently produce higher-quality outputs. We provide this comparative analysis for four representative length constraints—1024, 2048, 4096, and 8192 words—in Section K.2.

Our adapted evaluation prompt is as follows:

---

**LLM-as-a-judge Prompt for Evaluating Generations Quality**

You are an expert in evaluating text quality. Please evaluate the quality of an AI assistant's response to a user's writing request. Be as strict as possible.
You need to evaluate across the following six dimensions, with scores ranging from 0 to 10. The scoring criteria for each dimension are as follows (from 10 to 0):
1. Relevance: Evaluate how well the content stays on-topic and addresses the main theme of the user's request, regardless of any word or length limits specified by the user. Give a high score if the response generally follows the intent and subject of the instruction, even if not all minor aspects are covered. Deduct points only if the response goes off-topic, contains irrelevant or repeated information, or fails to respond to the main point of the user's request. Score from highly relevant and fully applicable to completely irrelevant or inapplicable.
2. Accuracy: Score from content that is completely accurate, with no factual errors or misleading information, to content with numerous errors and highly misleading information.
3. Coherence: Score from a clear structure with smooth logical connections, to a disorganized structure with no coherence.
4. Clarity: Score from clear, detailed, and easy-to-understand language, to confusing expression with minimal details.
5. Breadth and Depth: Score from content that is both broad and deep, providing a lot of information, to content that is seriously lacking in breadth and depth, with minimal information.
Please evaluate the quality of the following response to a user's request according to the above requirements.

⟨User Request⟩
    {user_request}
⟨/User Request⟩

⟨Response⟩

---

```
    {response}
</Response>
```

Please evaluate the quality of the response. You must first provide a brief analysis of its quality,then give a comprehensive analysis with scores for each dimension. The output must strictlyfollow the JSON format:
{
"Analysis": ...,
"Relevance": ...,
"Accuracy": ...,
"Coherence": ...,
"Clarity": ...,
"Breadth and Depth": ...
}.
You do not need to consider whether the response meets the user's length requirements in your evaluation. Ensure that only one integer between 0 and 10 is output for each dimension score.

In all experiments in this section, we randomly select a subset of 200 from LIFEBENCH, used only English, and used only *Equal To* as the control method.

## K.1 Does LLMs Follow the Length Instruction While Being Compatible with the Original Instruction?

In this section, we investigate whether LLMs fulfill explicit length constraints by compromising output quality when tasked to output text of specific lengths. We employ GPT-4o as the evaluation mnodel according to the prompt and process mentioned above. In Section K.1.1, we visualize the evaluation results as a set of line graphs, where the horizontal axis represents the increasing length constraints: $[16, 32, 64, 128, 256, 512, 1024, 2048, 4096, 8192]$.

Intuitively, longer generations may exhibit improved textual richness, including greater detail, explanatory depth, and rhetorical sophistication, potentially resulting in higher evaluation scores. Thus, we further explore the relationship between actual generated text length and evaluation scores. This analysis, detailed in Section K.1.2, provides insight into how generation length correlates with textual quality.

Synthesizing the findings from these analyses, we address the question: *Do LLMs follow length instructions without compromising adherence to the original task instructions?* Our empirical results indicate that adhering to length constraints generally does not adversely affect output quality. Observed reductions in generation quality under longer length constraints appear primarily attributable to intrinsic limitations in the model's long-text generation capabilities or the adoption of *Lazy Generation Strategies*, rather than inherent conflicts between length adherence and quality.

### K.1.1 Instruction Following as Length Constraint Increase

Introducing length instruction following as an additional objective transforms the original instruction into a multi-objective problem, creating a trade-off in model performance under multiple constraints [103, 41]. This raises an important question: do models resort to shortcuts—such as repetition or degenerate patterns—to satisfy the length constraint at the cost of faithfully following the original instruction, particularly under long-length requirements? Additionally, analyzing how output quality changes when models generate long text under explicit constraints offers insight into why certain models fail in long-form generation.

Our experimental results are shown in Figure 27, which reports the average score of generation quality across varying length constraints. Interestingly, different models exhibit distinct trends as constraints increase. Based on the observed patterns, we group models into four categories:

- **Increase then Decrease**: This is the most common pattern. These models tend to perform poorly under very short constraints but reach their peak quality around 512 or 1024 words. Beyond this threshold, textual quality decreases slightly but typically by less than one point from the peak. Representative models in this group include DeepSeek-R1,

**Score Comparison Across All Models and Lengths**

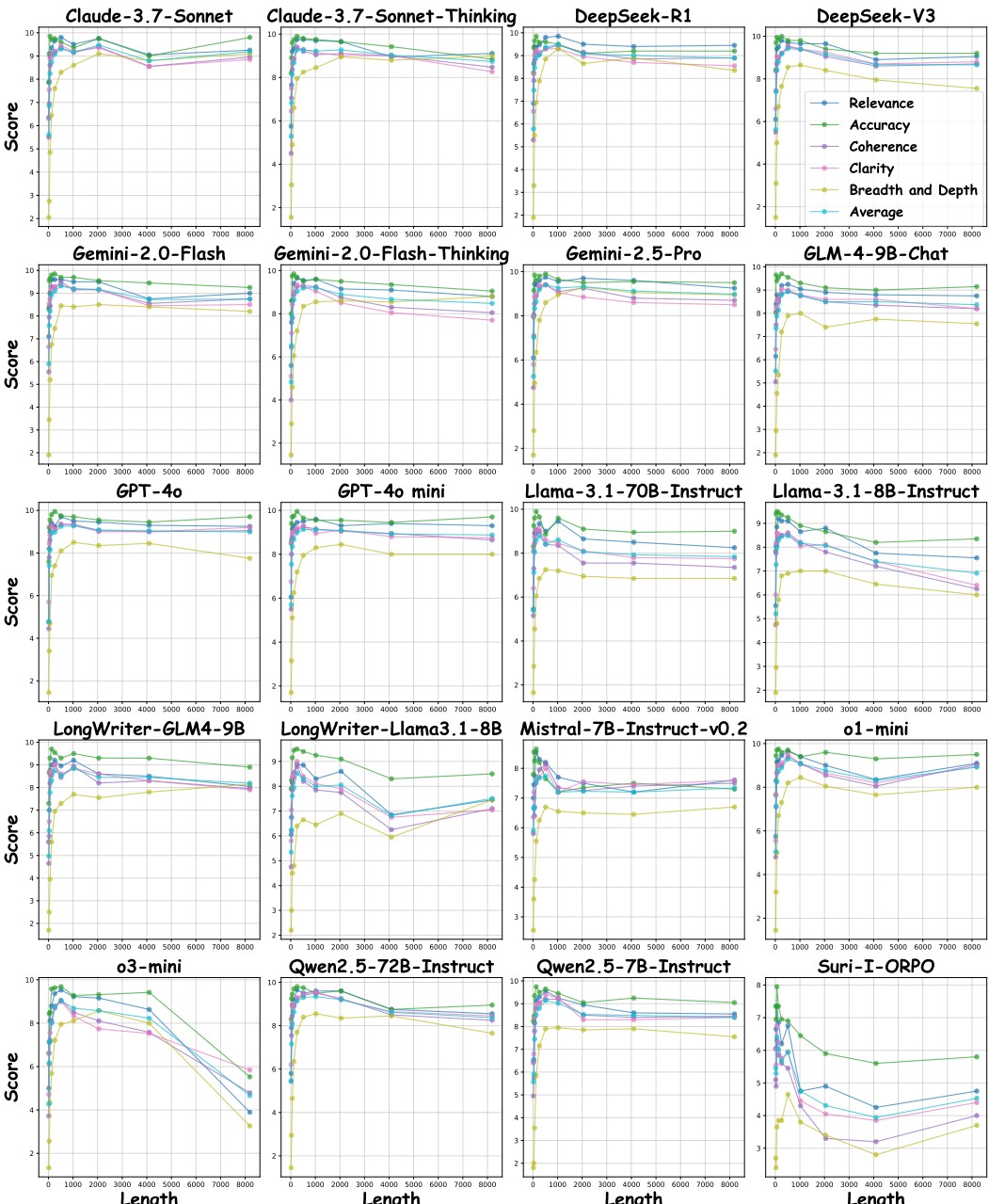

Figure 27: We compare 20 models with the increase of length instruction in terms of their textual quality—*relevance*, *accuracy*, *coherence*, *clarity*, and *breadth and depth*, and we calculate the average score. In this figure, we investigate the change in quality under the full-range length constraint, from 16 to 8192 words.

DeepSeek-V3, Gemini-2.0-Flash, Gemini-2.0-Flash-Thinking, GLM-4-9B-Chat, Llama-3.1-70B-Instruct, Llama-3.1-8B-Instruct, LongWriter-GLM4-9B, Mistral-7B-Instruct-v0.2, Qwen2.5-72B-Instruct, and Qwen2.5-7B-Instruct. As discussed in Section 5.1, this trend likely reflects limited long-text generation capabilities, which degrade output quality under longer constraints.

- **Reappearance of Maximum Score**: In this group, models first reach peak performance at 512 or 1024 words, but regain or maintain that peak at 8192 words. Models such as `Claude-3.7-Sonnet`, `Claude-3.7-Sonnet-Thinking`, `Gemini-2.5-Pro`, `GPT-4o`, `GPT-4o-mini`, and `o1-mini` belong to this category. These models typically outperform those in the **Increase then Decrease** group in length instruction following, as shown in Table 1, supporting the hypothesis that stronger instruction-following capabilities lead to more robust long-text generation.

- **Low Long-Text Quality**: These models show a significant drop in quality under long constraints—typically more than one point. Notably, this group includes `LongWriter-Llama3.1-8B` and `Suri-I-ORPO`, both of which are explicitly enhanced for long-text generation. However, their poor *Length Score* results in Table 1 suggest that these enhancements may come at the cost of weakened instruction-following ability, resulting in reduced output quality. These findings underscore that existing methods for improving long-text generation still struggle to address the limitations discussed in Section 5.2.

- **Over-Refusal Behavior**: This category is represented solely by `o3-mini`, which achieves strong length instruction adherence but exhibits frequent refusal behavior as constraints increase (see Section H). According to its system card [80], `o3-mini` is trained with aggressive safety alignment and refusal strategies. These include moderation models and safety classifiers designed to prevent overgeneration or unsafe content. While effective for safety, such alignment may overly restrict legitimate long-text generation. We hypothesize that `o3-mini`'s tendency to reject long-form tasks is a byproduct of these safety protocols—illustrating a trade-off between alignment safety and generative flexibility.

These results indicate that most models do not significantly compromise the completion of the original instruction when adhering to length constraints. This finding rules out the possibility that models rely on tricky strategies—such as repetition or degenerate patterns—to meet length requirements. As such, our analyses validate the integrity of the experimental setup used in the main paper and confirm that the observed trends reflect genuine model behaviors rather than artifacts of flawed evaluation design.

### K.1.2 Ablation Study: Text Quality and Length

In Section K.1.1, we analyzed the relationship between the instructed length and the resulting text quality. The findings suggest that when a model have both strong length instruction following and long-text generation capabilities, it can maintain adherence to the original instruction. However, given that most current models exhibit limited ability in generating high-quality long-form outputs, it becomes essential to further examine the relationship between actual output length and textual quality—regardless of whether the model successfully satisfies the explicit constraint. This section presents an ablation study designed to isolate and analyze the correlation between actually text length and quality.

The experimental results are shown in Figure 28, using the same experimental configuration as in Figure 27. Overall, we observe that, with the exception of `Suri-I-ORPO`, `Mistral-7B-Instruct-v0.2`, `LongWriter-Llama3.1-8B`, and `Llama-3.1-8B-Instruct`, an increase in actual output length generally leads to higher average quality scores—even when the target length constraint is not fully met. The results reveal that, among the 20 models evaluated, 10 exhibit a Pearson correlation coefficient greater than 0.6, indicating a strong positive relationship between output length and generation quality. Moreover, for the vast majority of models, the corresponding p-values fall below $1\mathrm{e}{-10}$, providing robust statistical evidence supporting the significance of this association. This supports the intuition previously discussed in Section K.1: longer outputs are more likely to contain richer content, better structure, and more developed reasoning, which collectively contribute to higher quality.

These results also explain why our evaluation does not directly compare generations from the same model under different length constraints—such comparisons could unfairly penalize shorter generations, even if excellent length instruction following ability.

Building on Figure 28, we provide further analysis that complements Section K.1.1. First, the scatter plot of actual generated length versus quality reveals a number of outliers—specifically, outputs exceeding 8192 words—from several open-source models, including `GLM-4-9B-Chat`, `Llama-3.1-70B-Instruct`, `Llama-3.1-8B-Instruct`, `LongWriter-GLM4-9B`, `LongWriter-Llama3.1-8B`, `Mistral-7B-Instruct-v0.2`, and

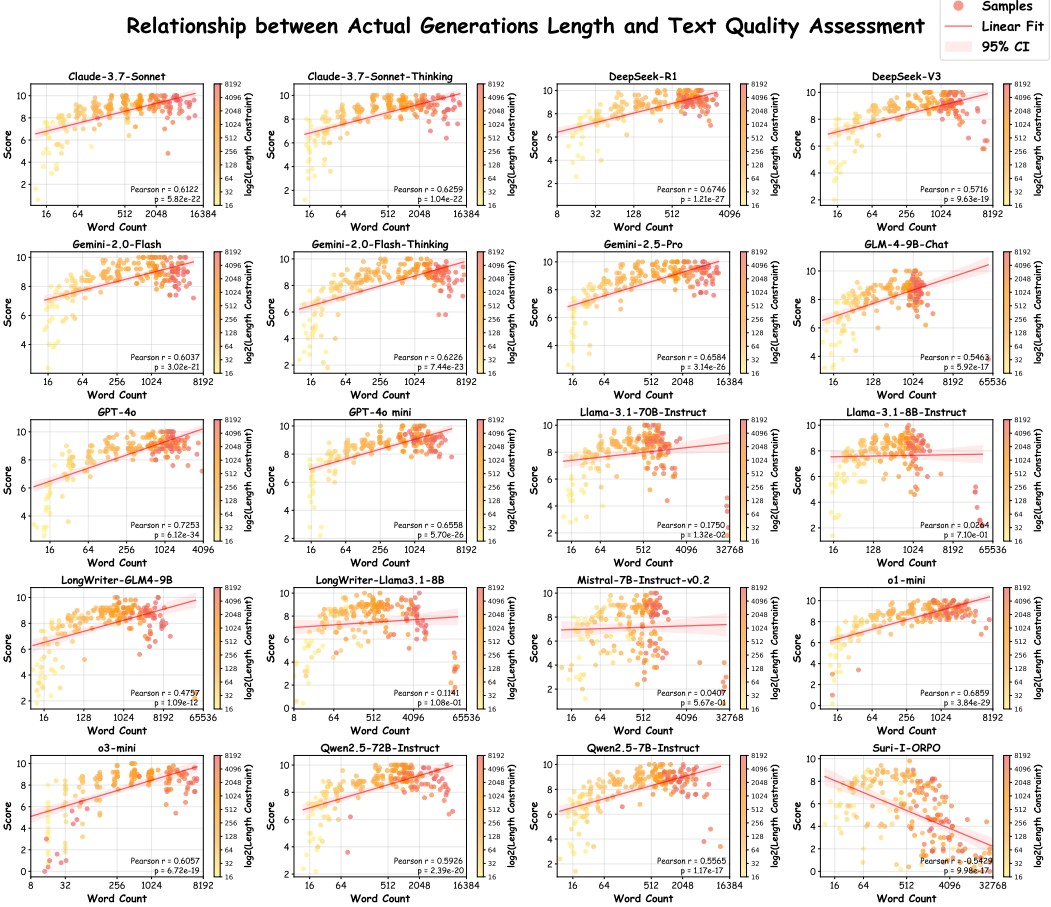

Figure 28: The horizontal axis is the actual length generated, and the vertical axis is the Average Score. The darker the point, the longer the required *Equal to* instruction length is. Note: The horizontal axis is an exponential coordinate with uneven distribution.

`Suri-I-ORPO`. These outliers are consistently associated with low quality scores. Manual inspection of these cases confirms that they often consist of meaningless repetitions. This finding reinforces the importance of jointly evaluating both length instruction following and textual quality, and it offers an additional explanation for model failure: under long-length constraints, failure may stem not only from an inability to reach the target length, but also from inherent limitations in fundamental ability of models.

In addition, Figure 28 visually illustrates the refusal behavior of models like `o3-mini`, where certain outputs under the 8192-word constraint are markedly shorter and of lower quality—highlighted as dark-colored points with poor alignment to the overall trend. In contrast, models with strong length instruction following—such as `Claude-3.7-Sonnet`, `Claude-3.7-Sonnet-Thinking`, and `o3-mini`—exhibit more structured output patterns. For these models, points of the same length constraint (same color in Figure 28) typically align in narrow vertical bands around the regression line, suggesting a consistent correlation between output length and quality. By contrast, models with weaker length adherence produce scatter plots with disorganized or erratic distributions, reflecting less reliable behavior across constraints.

## K.2 Text Quality Comparison of Long-text Generation

In Section K.1, we analyzed how the same model performs under varying length constraints and observed that generation quality often decreases under longer constraints. To further investigate

model behavior in this regime, we evaluate the performance of different models under the same long-length instruction to enable a fair comparison.

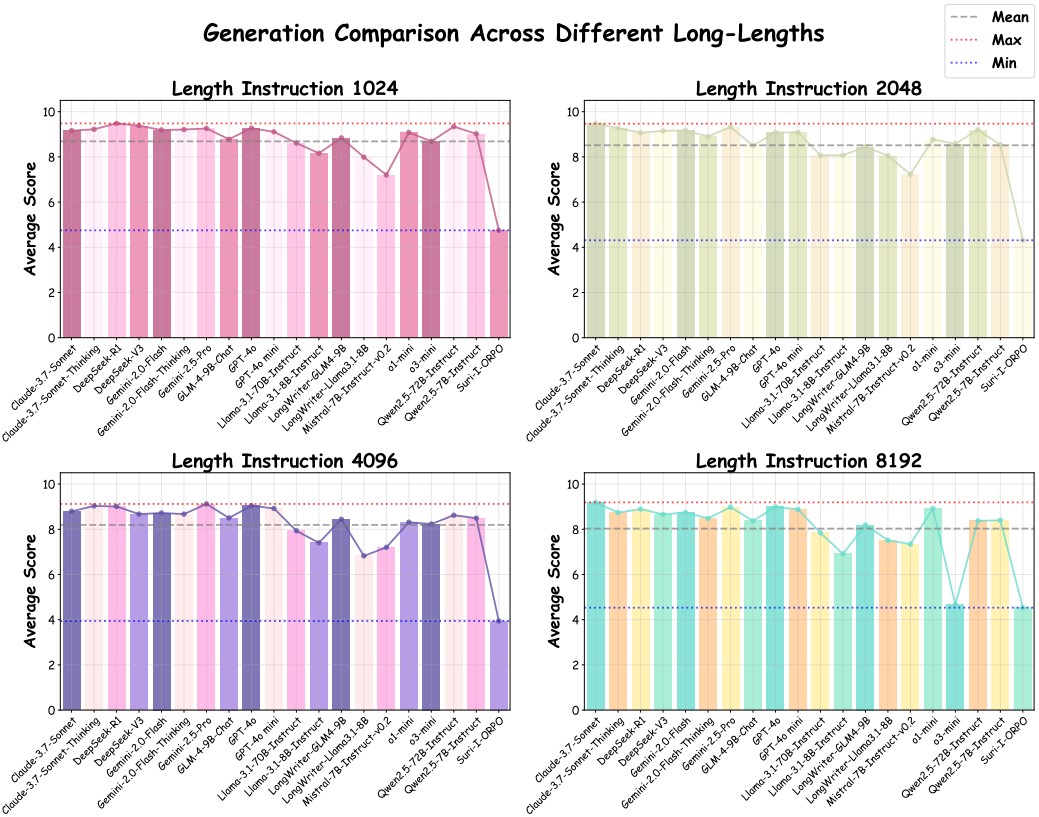

Figure 29: Under the four length constraints of [1024, 2048, 4096, 8192], different models generate average histograms of text quality. We marked the Mean, Max and Min reference lines of the bar in each sub-graph.

The results are presented in Figure 29. We observe that, with the exception of the special case `o3-mini`, models such as `Claude-3.7-Sonnet`, `Claude-3.7-Sonnet-Thinking`, and `Gemini-2.5-Pro` consistently achieve higher generation quality under long-length constraints. These findings highlight the relative strength of these models in maintaining output quality while adhering to demanding length requirements. The full score table is as follows:

| Model | 1024 | | | | | 2048 | | | | | 4096 | | | | | 8192 | | | | |
|---|---|---|---|---|---|---|---|---|---|---|---|---|---|---|---|---|---|---|---|---|
| | Re | Ac | Co | Cl | BD | Re | Ac | Co | Cl | BD | Re | Ac | Co | Cl | BD | Re | Ac | Co | Cl | BD |
| Claude-3.7-Sonnet | 9.50 | 9.35 | 9.15 | 9.20 | 8.60 | 9.75 | 9.75 | 9.40 | 9.35 | 9.10 | 9.05 | 9.00 | 8.55 | 8.55 | 8.80 | 9.25 | 9.80 | 8.95 | 8.85 | 9.10 |
| Claude-3.7-Sonnet-Thinking | 9.70 | 9.75 | 9.05 | 9.15 | 8.45 | 9.65 | 9.65 | 9.10 | 9.00 | 8.95 | 8.95 | 9.42 | 9.00 | 9.00 | 8.79 | 9.11 | 8.84 | 8.47 | 8.26 | 9.00 |
| DeepSeek-R1 | 9.85 | 9.50 | 9.45 | 9.30 | 9.30 | 9.50 | 9.10 | 9.15 | 8.95 | 8.65 | 9.40 | 9.20 | 8.85 | 8.70 | 8.90 | 9.45 | 9.20 | 8.90 | 8.55 | 8.35 |
| DeepSeek-V3 | 9.65 | 9.80 | 9.40 | 9.40 | 8.65 | 9.65 | 9.40 | 9.05 | 9.25 | 8.40 | 8.90 | 9.20 | 8.60 | 8.70 | 7.95 | 9.05 | 9.20 | 8.70 | 8.80 | 7.55 |
| Gemini-2.0-Flash | 9.50 | 9.70 | 9.15 | 9.20 | 8.40 | 9.50 | 9.55 | 9.15 | 9.15 | 8.50 | 8.75 | 9.45 | 8.55 | 8.45 | 8.40 | 9.00 | 9.25 | 8.75 | 8.50 | 8.20 |
| Gemini-2.0-Flash-Thinking | 9.60 | 9.60 | 9.25 | 9.05 | 8.55 | 9.15 | 9.50 | 8.75 | 8.50 | 8.60 | 9.10 | 9.35 | 8.30 | 8.05 | 8.55 | 8.80 | 9.05 | 8.05 | 7.70 | 8.80 |
| Gemini-2.5-Pro | 9.55 | 9.65 | 9.10 | 9.05 | 8.95 | 9.70 | 9.50 | 9.25 | 8.85 | 9.30 | 9.60 | 9.55 | 8.80 | 8.60 | 9.05 | 9.25 | 9.50 | 8.70 | 8.50 | 8.95 |
| GLM-4-9B-Chat | 9.05 | 9.30 | 8.75 | 8.80 | 8.00 | 8.90 | 9.10 | 8.50 | 8.60 | 7.40 | 8.80 | 9.00 | 8.35 | 8.60 | 7.75 | 8.75 | 9.15 | 8.20 | 8.20 | 7.55 |
| GPT-4o | 9.50 | 9.70 | 9.35 | 9.30 | 8.50 | 9.45 | 9.55 | 9.05 | 9.00 | 8.35 | 9.30 | 9.45 | 9.00 | 9.00 | 8.45 | 9.25 | 9.70 | 9.05 | 9.20 | 7.75 |
| GPT-4o mini | 9.60 | 9.55 | 9.15 | 8.95 | 8.30 | 9.30 | 9.55 | 9.05 | 9.10 | 8.45 | 9.40 | 9.45 | 8.95 | 8.80 | 8.00 | 9.30 | 9.70 | 8.65 | 8.75 | 8.00 |
| Llama-3.1-70B-Instruct | 9.45 | 9.60 | 8.35 | 8.45 | 7.20 | 8.65 | 9.10 | 7.55 | 8.10 | 6.95 | 8.07 | 8.95 | 7.55 | 7.80 | 6.85 | 8.25 | 9.00 | 7.35 | 7.75 | 6.85 |
| Llama-3.1-8B-Instruct | 8.65 | 8.90 | 8.20 | 8.05 | 7.00 | 8.80 | 8.65 | 7.80 | 8.10 | 7.00 | 7.75 | 8.20 | 7.20 | 7.40 | 6.45 | 7.55 | 8.35 | 6.25 | 6.40 | 6.00 |
| LongWriter-GLM4-9B | 9.20 | 9.50 | 8.95 | 8.85 | 7.70 | 8.60 | 9.30 | 8.20 | 8.60 | 7.55 | 8.50 | 9.30 | 8.30 | 8.30 | 7.80 | 8.05 | 8.90 | 7.95 | 7.90 | 8.15 |
| LongWriter-Llama3.1-8B | 8.30 | 9.25 | 7.85 | 8.10 | 6.45 | 8.60 | 9.10 | 7.75 | 7.90 | 6.90 | 6.85 | 8.30 | 6.25 | 6.75 | 5.95 | 7.45 | 8.50 | 7.10 | 7.05 | 7.45 |
| Mistral-7B-Instruct-v0.2 | 7.70 | 7.20 | 7.35 | 7.20 | 6.55 | 7.50 | 7.35 | 7.25 | 7.55 | 6.50 | 7.20 | 7.50 | 7.40 | 7.45 | 6.45 | 7.60 | 7.30 | 7.50 | 7.60 | 6.70 |
| o1-mini | 9.40 | 9.40 | 9.10 | 9.05 | 8.45 | 9.00 | 9.60 | 8.55 | 8.65 | 8.05 | 8.35 | 9.30 | 8.05 | 8.20 | 7.65 | 9.10 | 9.50 | 9.05 | 8.95 | 8.00 |
| o3-mini | 9.22 | 9.40 | 8.50 | 8.33 | 8.11 | 9.16 | 9.32 | 8.11 | 7.74 | 8.58 | 8.63 | 9.42 | 7.58 | 7.53 | 8.00 | 3.89 | 5.53 | 4.79 | 5.84 | 3.26 |
| Qwen2.5-72B-Instruct | 9.60 | 9.50 | 9.50 | 9.55 | 8.55 | 9.60 | 9.60 | 9.25 | 9.20 | 8.35 | 8.75 | 8.75 | 8.50 | 8.65 | 8.45 | 8.55 | 8.95 | 8.25 | 8.45 | 7.65 |
| Qwen2.5-7B-Instruct | 9.25 | 9.45 | 9.20 | 9.25 | 7.95 | 8.95 | 9.05 | 8.50 | 8.30 | 7.85 | 8.60 | 9.25 | 8.40 | 8.30 | 7.90 | 8.55 | 9.05 | 8.45 | 8.40 | 7.55 |
| Suri-I-ORPO | 4.75 | 6.45 | 4.30 | 4.45 | 3.80 | 4.90 | 5.90 | 3.30 | 4.05 | 3.40 | 4.25 | 5.60 | 3.20 | 3.85 | 2.80 | 4.75 | 5.80 | 4.00 | 4.40 | 3.70 |

Table 17: Model comparison across four length constraints. **Metrics:** Re means *Relevance*, Ac means *Accuracy*, Co means *Coherence*, Cl means *Clarity*, BD means *Breadth & Depth*.

## L  Comparison with Existing Leaderboard

Chatbot Arena is an open platform for crowdsourced AI benchmarking [18]. With over one million user votes, the platform ranks leading LLMs and AI chatbots using the Bradley-Terry model, producing continuously updated leaderboards.

We compare the *Length Score* of models evaluated on LIFEBENCH with the *Arena Scores* from the Chatbot Arena LLM Leaderboard. For models with multiple available versions, we select the version closest to our evaluated model (detailed version mappings are listed in Table 18). We compute both Pearson and Spearman correlation coefficients between *Length Score* and the *Arena Scores*. The strongest correlation is observed for "Hard Prompts w/SC", with Pearson and Spearman coefficients of 0.78 and 0.71, respectively, indicating a strong positive correlation. "Longer Query" exhibits the next highest correlation, with Pearson and Spearman coefficients of 0.75 and 0.67, also reflecting a strong positive correlation. In contrast, the lowest correlations are observed for "Creative Writing", with Pearson and Spearman coefficients of 0.64 and 0.51, and for "Multi-Turn", with Pearson and Spearman coefficients of 0.67 and 0.50, which suggest a moderate positive correlation.

As shown in Figure 30, several models exhibit notable deviations between their *Length Score* and *Arena Scores*, especially among those with higher *Arena Scores*. In particular, `o3-mini` achieves a considerably higher *Length Score* (75.4) relative to its Arena Score (1305), while both `Gemini-2.0-Flash` and `DeepSeek-R1` demonstrate notably lower *Length Score* values (48.4 and 47.7, respectively) despite high Arena Scores (1354 and 1358, respectively).

These results suggest that even among models with generally strong overall performance, there can be substantial differences in their ability to follow length instructions. This highlights that length instruction following is a distinct aspect of model capability that is not fully captured by aggregate leaderboard scores. Consequently, explicitly evaluating and aligning models on length-specific behaviors is essential for comprehensive benchmarking and systematic improvement of overall model capabilities.

Table 18: Correspondence between models evaluated in LIFEBENCH and their respective versions and overall *Arena Scores* on the Chatbot Arena LLM Leaderboard.

| Model | Chatbot Arena Model Version | Overall Arena Score[1] |
|---|---|---|
| GPT-4o mini | gpt-4o-mini-2024-07-18 | 1272 |
| GPT-4o | gpt-4o-2024-08-06 | 1265 |
| o1-mini | o1-mini | 1303 |
| o3-mini | o3-mini | 1305 |
| Claude-3.7-Sonnet | claude-3-7-sonnet-20250219 | 1290 |
| Claude-3.7-Sonnet-Thinking | claude-3-7-sonnet-20250219-thinking-32k | 1301 |
| Gemini-2.0-Flash | gemini-2.0-flash-001 | 1354 |
| Gemini-2.5-Pro | gemini-2.5-pro-preview-05-06 | 1447 |
| DeepSeek-R1 | deepseek-r1 | 1358 |
| DeepSeek-V3 | deepseek-v3 | 1318 |
| Llama-3.1-8B-Instruct | llama-3.1-8b-instruct | 1175 |
| Llama-3.1-70B-Instruct | llama-3.1-70b-instruct | 1247 |
| Qwen2.5-72B-Instruct | qwen2.5-72b-instruct | 1257 |
| Qwen3-235B-A22B | qwen3-235b-a22b | 1341 |
| Mistral-7B-Instruct-v0.2 | mistral-7b-instruct-v0.2 | 1072 |

[1] The data was collected on May 12, 2025, from the official leaderboard at `https://lmarena.ai/?leaderboard`.

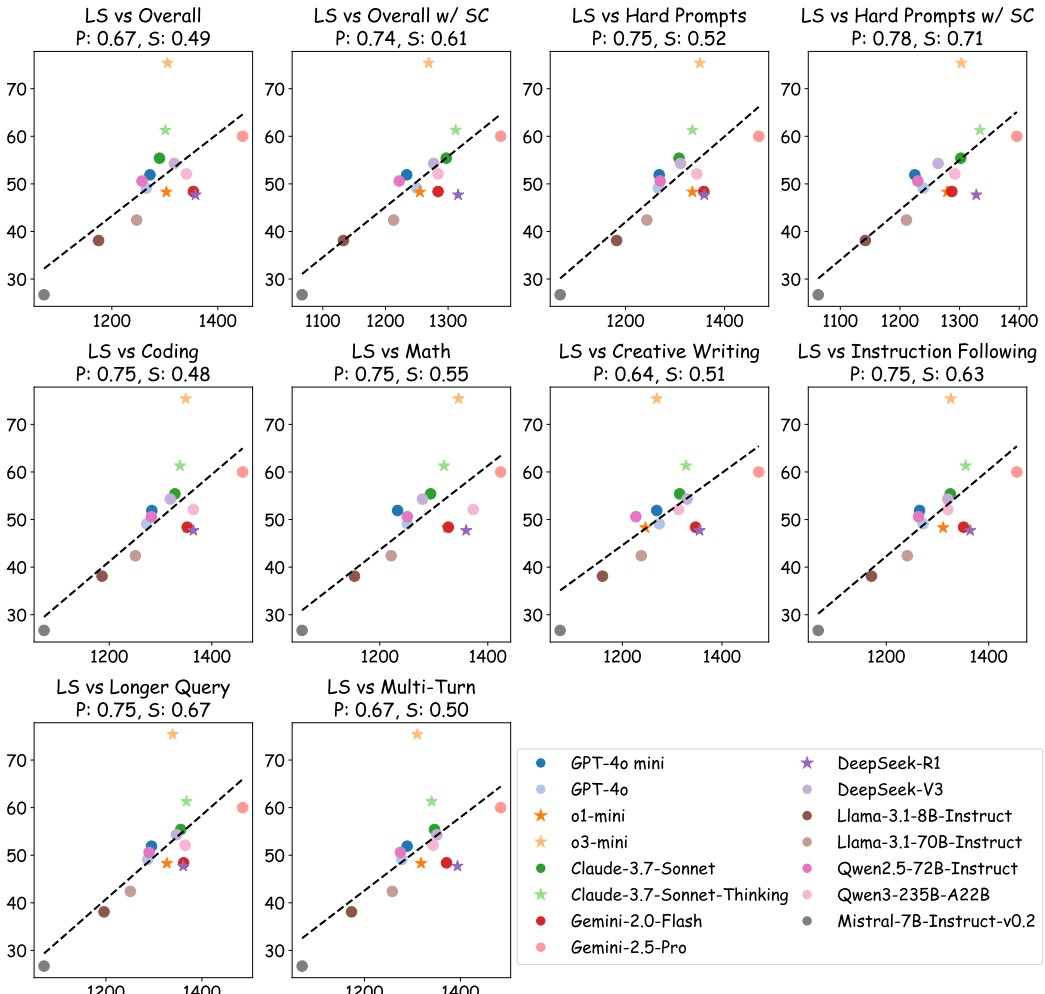

Figure 30: Comparison of *Length Score* with the *Arena Scores*. **P** denotes the Pearson correlation coefficient, and **S** denotes the Spearman correlation coefficient. Reasoning models are indicated by a star marker, while non-reasoning models use a circle marker. "w/SC" stands for "without style control". The black dashed line represents the fitted regression result.

# M  Future Directions and Potential Solutions

In this section, we outline future directions and potential solutions for improving length instruction following in LLMs. Section M.1 investigates the model's internal understanding of input prompts under varying length constraints, examining the correlation between actual output length and the logit/probability of the end-of-sequence (EoS) token. In Section M.2, we assess the ability of unaligned models to generate long text and follow explicit length instructions, offering insight into the role of pre-training in shaping these capabilities. Finally, Section M.3 introduces the *Pre-Planning* method, which significantly enhances LLM's adherence to length instructions and improves its ability to extrapolate toward the upper bounds of its generation capacity.

## M.1  Interpretability of Length Instruction Following

Understanding how LLMs interpret and respond to varying length constraints is crucial for diagnosing why these models fail to consistently adhere to such instructions. However, interpretability analyses in this area remain unexplored. In this section, we provide preliminary insights through a straightforward examination of model behavior, highlighting avenues for future interpretability research.

Specifically, we analyze the behavior of the end-of-sequence (EoS) token, a special token that signals LLMs to terminate generation. When an LLM deems the current sequence incomplete, the probability or logit assigned to the EoS token for the subsequent prediction should remain low. Intuitively, by varying only the specified length constraint (e.g., comparing prompts such as *"Please generate a summary of 256 words"* versus *"Please generate a summary of 8192 words"*), we can assess how internal predictions of the EoS token shift in response, offering insights into the model's intrinsic representation of output completeness.

Our experiments preliminarily confirm this intuition, as illustrated in Figure 31. We observe that LLMs generally possess a good capability to differentiate among distinct length constraints. Indeed, a clear inverse relationship emerges between the models' actual generated length and their corresponding EoS logits or probabilities—especially at shorter constraints, where models perform relatively well. For instance, models such as `Llama-3.1-70B-Instruct` and `Mistral-7B-Instruct-v0.2` demonstrate a consistent pattern: under constraints of up to 2048 words, lower EoS logits or probabilities correlate strongly with longer generations. However, some anomalies appear in other models: `Llama-3.1-8B-Instruct` shows this inverse relationship clearly only in probability, while `Qwen2.5-7B-Instruct` primarily demonstrates it in logits. Additionally, at very long constraints, all models exhibit abnormal behavior, with EoS logits and probabilities often reversing their earlier trends.

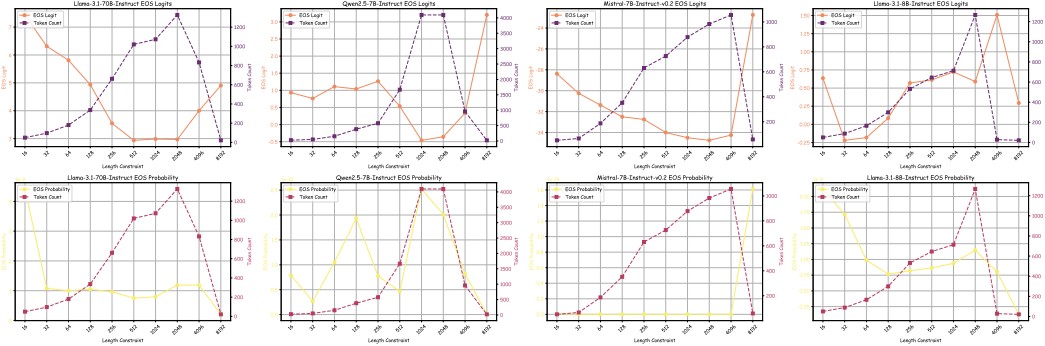

Figure 31: This figure illustrates the relationship between the model's actual output length and the generation dynamics of the end-of-sequence (EoS) token when only the length constraint in the instruction is varied. (**First Row**) The left Y-axis indicates the logit value of the EoS token, while the right Y-axis shows the actual number of tokens generated. (**Second Row**) The left Y-axis presents the probability of the EoS token, and the right Y-axis again reflects the actual number of tokens generated.

Although we provide initial insights, these analyses remain exploratory, as they do not fully explain the common underlying reasons for length-instruction failures. Nevertheless, we believe these

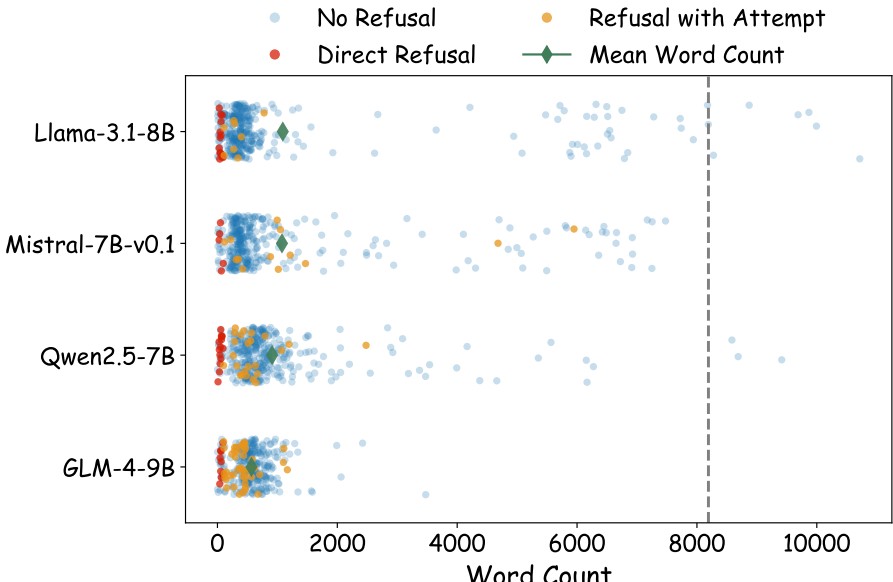

Figure 32: Output length distributions of base models. The gray dashed line denotes the length constraint.

preliminary findings point toward an interpretable solution for addressing insufficient adherence to length instructions—an avenue we intend to further investigate in future work.

## M.2 Improving Length Instruction Following During Pre-training

In Section H, we identified behaviors such as premature termination and explicit refusals as notable issues in length instruction following. While our analysis indicated that safety alignment might partly explain rejection behaviors, it remains unclear whether post-training processes adversely affect the underlying long-text generation capabilities, leading specifically to premature termination. Understanding whether these limitations are inherent to the pre-trained models or introduced during post-training is crucial. If deficiencies in long-text generation and length adherence emerge primarily from post-training, targeted improvements to post-training strategies would suffice. Conversely, if these limitations originate during pre-training, a revised pre-training approach will be necessary.

To investigate this, we evaluated the base pre-trained models' length instruction following capabilities using in-context learning [64], thereby excluding post-training influences and assessing inherent pre-trained capabilities directly. We conducted experiments on 360 fundamental data units under an 8192-word constraint with the *Equal To* control method. Four representative open-source base models were tested: `Llama-3.1-8B`, `GLM-4-9B`, `Qwen2.5-7B`, and `Mistral-7B-v0.1`. The experimental results are summarized as follows:

- `Llama-3.1-8B`: Average length: 1090.29; Direct Refusal: 3.95%; Refusal with Attempt: 2.41%.
- `GLM-4-9B`: Average length: 566.97; Direct Refusal: 2.91%; Refusal with Attempt: 18.11%.
- `Qwen2.5-7B`: Average length: 912.54; Direct Refusal: 3.91%; Refusal with Attempt: 7.50%.
- `Mistral-7B-v0.1`: Average length: 1076.74; Direct Refusal: 1.47%; Refusal with Attempt: 4.02%.

We further illustrate these findings in Figure 32, which shows the distribution of actual output lengths relative to the imposed constraint. Notably, even these base models exhibit lazy generation behaviors, such as premature termination and explicit refusal. Although extreme rejection behaviors (e.g., those observed with `o3-mini`) might be amplified through safety-oriented post-training, our results indicate that inherent deficiencies already exist within the pre-trained models. Moreover, these base models

consistently perform worse than their instruction-aligned counterparts in tasks requiring outputs of 8192 words, highlighting intrinsic limitations in their long-text generation capability.

In conclusion, these findings suggest that the problems associated with long-text generation and excessive rejection behaviors primarily originate during the pre-training stage. While post-training alignment may exacerbate these issues, it does not constitute their root cause. We hypothesize that inadequate representation of long-form texts in pre-training datasets might contribute significantly to these observed deficiencies. Thus, integrating more extensive long-text data during pre-training may be a promising avenue for improving length instruction following capabilities in future model development.

### M.3 Improving Length Instruction Following During Post-training

Although Section M.2 highlights that deficiencies in length instruction following, particularly for long-form generation, originate primarily during the pre-training stage, we conjecture that these shortcomings can also be substantially mitigated through targeted post-training methods. Analogously, just as models initially incapable of reasoning or even answering simple questions (e.g., *"How many 'r's are there in 'strawberry'?"*) can acquire these capabilities via suitable post-training alignment, length instruction adherence may similarly benefit from specialized alignment strategies.

Inspired by Chain-of-Thought [102], wherein reasoning-based LLMs are encouraged to perform structured pre-planning before generating responses, we explored a similar pre-planning strategy tailored specifically for long-text generation. Concretely, we manually composed structured content outlines resembling book catalogs, then allowed LLMs to refine the prompt based on these improved outlines before generating extensive texts. We conducted experiments using four proprietary models—GPT-4o mini, Gemini-2.0-Flash, Doubao-1.5-Pro, and DeepSeek-V3—all evaluated on creative generation tasks from LIFEBENCH, with an *Equal To* control method and a fixed length constraint of 8192 words.

The results presented in Table 19 clearly indicate that this pre-planning approach significantly improves the performance of models that initially struggled with length instruction following. Among the four models evaluated, we observe substantial improvements in *Length Score* scores, with increases consistently exceeding 57 points and the highest achieving 76.4—substantially outperforming the top-performing model, Gemini-2.5-Pro (39.4, as reported in Table 11). Correspondingly, significant reduc-

Table 19: *Length Score* and *Length Deviation* values for each model after pre-planning, with improvements relative to the baseline highlighted in green.

| Model | LS | LD |
|---|---|---|
| GPT-4o mini | 71.9 (↑ 70.2) | 16% (↓ 69%) |
| Gemini-2.0-Flash | 66.6 (↑ 57.4) | 9% (↓ 48%) |
| Doubao-1.5-Pro | 72.0 (↑ 59.7) | 11% (↓ 41%) |
| DeepSeek-V3 | 76.4 (↑ 70.3) | 9% (↓ 59%) |

tions in *Length Deviation* indicate notably better adherence to the specified length constraints. These findings strongly suggest that enabling LLMs to explicitly plan and structure their outputs in advance can markedly enhance their long-text generation capabilities, presenting a promising direction for improving length instruction adherence through targeted post-training strategies.

# N Length Instruction Following under Output Format Constraints

To further investigate how length instruction following is affected when models are required to follow multiple-objective instructions, we extend LIFEBENCH by introducing an additional output format constraint. Specifically, we examine the impact of requiring generated text to not only follow a specified word count but also produce outputs conforming to specified structural formats. This approach allows us to assess how format-related complexities influence models' capabilities to follow precisely length instructions.

We define three distinct output formats, arranged by increasing structural complexity:

- **Markdown:** Requires only basic structural elements such as headings and lists, and employs lightweight syntax, making it the simplest format. The prompt specifies: *"Your output must be in Markdown format. Use ## for headings, - for lists, and standard Markdown syntax for formatting."*

- **HTML:** Involves a broader set of structural elements and nested tags, representing moderate formatting complexity. The prompt specifies: *"Your output must be in HTML format. Use `<h2>` for headings, `<ul><li>` for lists, and standard HTML tags for formatting."*

- **LaTeX:** Demands strict command-level syntax and structural conformity, posing the greatest challenge among the three formats. The prompt specifies: *"Your output must be in LaTeX format. Use \section{} for headings, \itemize{} for lists, and standard LaTeX commands for formatting. Do not include preamble or document class."*

We evaluate model performance across five length constraints: 128, 256, 512, 1024, and 2048 words, covering a representative range from short to moderate generation lengths. These constraints were carefully selected to ensure added formatting syntax does not exceed the maximum generation length supported by the models. Our evaluation includes leading proprietary models from OpenAI (GPT-4o, GPT-4o-mini, o1-mini, o3-mini) and Google (Gemini-2.0-Flash, Gemini-2.0-Flash-Thinking, Gemini-2.5-Pro).

To accurately measure the semantic word count of generated outputs, we employ a rule-based post-processing pipeline tailored to each output format. Specifically, HTML tags are removed by stripping all content enclosed within angle brackets (`<...>`). For Markdown and LaTeX, lines or tokens corresponding explicitly to formatting commands or syntax elements—such as headers (#), list markers (-), or command sequences (\)—are removed. This preprocessing ensures accurate assessment of semantic content length, unaffected by formatting-related artifacts.

Table 20: Effect of **Markdown** format constraints on *Length Score*. Green indicates improved scores, while red denotes decreased scores.

| Model | Length Constrainats | | | | |
|---|---|---|---|---|---|
| | 128 | 256 | 512 | 1024 | 2048 |
| GPT-4o mini | 50.5 (↓ 12.3) | 62.6 (↓ 7.2) | 70.5 (↑ 3.7) | 71.6 (↓ 3.1) | 26.5 (↑ 1.0) |
| GPT-4o | 56.6 (↓ 10.4) | 69.8 (↑ 0.2) | 68.4 (↑ 1.9) | 61.0 (↓ 8.0) | 22.6 (↑ 6.1) |
| o1-mini | 53.4 (↓ 13.2) | 52.6 (↓ 9.9) | 44.7 (↓ 6.5) | 38.5 (↓ 1.8) | 20.2 (↓ 14.0) |
| Gemini-2.0-Flash | 44.2 (↓ 15.6) | 51.6 (↓ 8.1) | 34.0 (↓ 4.2) | 51.2 (↑ 3.5) | 52.2 (↓ 0.5) |
| Gemini-2.0-Flash-Thinking | 51.0 (↓ 2.4) | 52.1 (↑ 3.1) | 56.6 (↑ 26.6) | 47.0 (↑ 14.9) | 28.6 (↓ 13.9) |
| Gemini-2.5-Pro | 67.1 (↓ 5.7) | 63.7 (↓ 3.7) | 54.2 (↑ 4.7) | 35.4 (↓ 1.6) | 30.6 (↓ 17.4) |

Tables 20, Table 21, and Table 22 summarize the model performances measured by *Length Score* under Markdown, HTML, and LaTeX format constraints, respectively. Overall, imposing additional format constraints generally leads to lower *Length Score* scores across most models, especially at extreme length targets (128 and 2048 words). For instance, o1-mini and Gemini-2.0-Flash consistently show reduced performance across nearly all scenarios, suggesting that the complexity of structural formatting can substantially impair the ability of models to precisely follow length instructions.

Table 21: Effect of **HTML** format constraints on *Length Score*. Green indicates improved scores, while red denotes decreased scores.

| Model | Length Constrainats | | | | |
|---|---|---|---|---|---|
| | 128 | 256 | 512 | 1024 | 2048 |
| GPT-4o mini | 51.0 (↓ 11.8) | 64.6 (↓ 5.2) | 77.8 (↑ 11.0) | 55.1 (↓ 19.6) | 19.0 (↓ 6.5) |
| GPT-4o | 59.6 (↓ 7.4) | 71.9 (↑ 2.3) | 68.9 (↑ 2.4) | 59.8 (↓ 9.2) | 12.2 (↓ 4.3) |
| o1-mini | 55.8 (↓ 10.8) | 51.9 (↓ 10.6) | 47.3 (↓ 3.9) | 31.9 (↓ 8.4) | 14.0 (↓ 20.2) |
| Gemini-2.0-Flash | 44.1 (↓ 15.7) | 51.3 (↓ 8.4) | 35.6 (↓ 2.6) | 39.8 (↓ 7.9) | 53.3 (↑ 0.6) |
| Gemini-2.0-Flash-Thinking | 56.3 (↑ 2.9) | 54.9 (↑ 5.9) | 55.3 (↑ 25.3) | 38.5 (↑ 6.4) | 18.0 (↓ 24.5) |
| Gemini-2.5-Pro | 64.0 (↓ 8.8) | 58.2 (↓ 9.2) | 50.9 (↑ 1.4) | 39.8 (↑ 2.8) | 30.5 (↓ 17.5) |

Table 22: Effect of **LaTex** format constraints on *Length Score*. Green indicates improved scores, while red denotes decreased scores.

| Model | Length Constrainats | | | | |
|---|---|---|---|---|---|
| | 128 | 256 | 512 | 1024 | 2048 |
| GPT-4o mini | 48.2 (↓ 14.6) | 63.2 (↓ 6.6) | 71.8 (↑ 5.0) | 65.6 (↓ 9.1) | 24.9 (↓ 0.6) |
| GPT-4o | 54.3 (↓ 12.7) | 67.3 (↓ 2.3) | 66.4 (↓ 0.1) | 61.4 (↓ 7.6) | 14.1 (↓ 2.4) |
| o1-mini | 60.2 (↓ 6.4) | 52.3 (↓ 10.2) | 44.2 (↓ 7.0) | 40.3 ( - ) | 17.5 (↓ 16.7) |
| Gemini-2.0-Flash | 52.9 (↓ 6.9) | 51.0 (↓ 8.7) | 37.3 (↓ 0.9) | 54.7 (↑ 7.0) | 59.5 (↑ 6.8) |
| Gemini-2.0-Flash-Thinking | 58.0 (↑ 4.6) | 55.3 (↑ 6.3) | 48.7 (↑ 18.7) | 34.9 (↑ 2.8) | 25.1 (↓ 17.4) |
| Gemini-2.5-Pro | 62.9 (↓ 9.9) | 60.8 (↓ 6.6) | 56.7 (↑ 7.2) | 37.8 (↑ 0.8) | 33.7 (↓ 14.3) |

However, the `Gemini-2.0-Flash-Thinking` model displays a notable deviation from this general trend, achieving improved *Length Score* scores at intermediate length constraints (256, 512, and 1024 words) across all three output formats. This behavior suggests that, under certain circumstances, structured output requirements may enhance rather than impede length compliance, possibly by encouraging the model to employ more deliberate, organized generation strategies.

In summary, although output format constraints typically introduce significant additional challenges to length instruction adherence, the exceptional performance of `Gemini-2.0-Flash-Thinking` indicates promising directions for future model design and training. Specifically, incorporating structured formatting constraints into training may not only mitigate performance degradation but could potentially facilitate more effective and precise length instruction following.

