# OpenReview forum: "LIFEBENCH: Evaluating Length Instruction Following in Large Language Models"
_NeurIPS.cc/2025/Datasets_and_Benchmarks_Track — NeurIPS 2025 Datasets and Benchmarks Track poster_

### Official Review · Reviewer_Ltyk · 2025-06-08

**Rating:** 5
**Confidence:** 4

**Summary:**

LIFEBENCH introduces a new benchmark that reveals an unexpected weakness in today's LLMs: they struggle to follow simple length instructions even though they can handle complex reasoning tasks. Testing 26 models across over 10,000 instances, the study shows that generating text of a specific length—something that seems straightforward—actually poses serious challenges for current systems. The research uncovers a troubling gap between what companies claim their models can produce and what they actually deliver, with many models resorting to "lazy strategies" like cutting responses short or simply refusing to generate longer content. Perhaps more concerning, models consistently get their own word counts wrong, showing they lack basic awareness of how much they've written. The benchmark also finds notable differences in performance across languages and reveals that longer input texts make the problem worse. These discoveries call into question some basic assumptions about what LLMs can do well, suggesting that length control is a more important and overlooked skill than previously recognized. The open-source benchmark gives researchers a tool to better understand and hopefully fix these limitations in future models.

**Additional Feedback:**

Please refer to the Limitations Weaknesses.

**Dataset Code Accessibility:**

Yes

**Dataset Code Comments:**

I have taken a closed look at the github repo, project page, and huggingface dataset provided by the authors. I think the dataset and code are avalable in a usable format and well documented. The proposed benchmark dataset is fully open-sourced and will be easily followed.

**Ethical Comments:**

I have taken a closed look at the provided data and code, and find no ethical concerns.

**Ethical Considerations:**

No, there are no or only very minor ethics concerns

**Final Justification:**

The new experimental results, which have addressed all my previous concerns. I have raised my score from 4 to 5.

**Limitations Weaknesses:**

• How can we ensure the tasks are reasonable? Some tasks might not be suitable for short or long outputs, and there could be potential conflicts between task requirements and length constraints that make certain combinations fundamentally problematic or artificial.

• Could the authors provide more justification or intuitive reasoning for the Length Score hyperparameters (k1=5, k2=2), as these settings might lead to over-penalization and it's unclear what the rationale behind these specific values is?

• Why wasn't there any evaluation metric designed for generation quality? It seems like a significant oversight since models might achieve good length compliance while producing low-quality content, and without quality assessment, we can't tell if length control comes at the expense of output usefulness or coherence.

• Language analysis could be more thorough—while Chinese-English comparisons are valuable, the consistent finding of longer Chinese outputs warrants deeper investigation of tokenization effects, and token-based comparisons might reveal additional insights about cross-linguistic differences.

• Results for reasoning models don't clarify whether thinking processes are included in output measurements, and it would be valuable to explore how thinking length relates to output constraints or whether similar patterns emerge when limiting reasoning traces.

• Outstanding performance by o3-mini deserves deeper investigation to understand the underlying factors that enable such superior results, which could inform improvements in other model families.

**Strengths Contributions:**

• The paper tackles a fundamental yet surprisingly neglected aspect of LLM evaluation—while length control is essential for practical applications, this basic capability has received little systematic attention despite its widespread importance.

• Testing scope is impressive, covering diverse tasks across English and Chinese with length constraints from 16 to 8,192 words, effectively capturing real-world requirements while maintaining methodological rigor through careful paradigm design and statistical analysis.

• The experimental work is comprehensive with substantial effort invested, making the results highly credible and trustworthy.

• Experimental work goes far beyond surface-level performance measurement through comprehensive diagnostic studies including length awareness testing, input characteristic analysis, and reasoning trace experiments that provide deep insights into underlying failure mechanisms.

• Findings reveal counterintuitive patterns that advance LLM understanding, such as reasoning models outperforming specialized long-text models, significant gaps between vendor claims and actual capabilities, and interesting behaviors in how different model types approach length planning.

• Authors provide meaningful theoretical grounding by discussing potential causes and solutions from interpretability, pre-training, and post-training perspectives, offering valuable research directions even though these explorations remain preliminary.

• Open-source release creates immediate practical value, giving researchers and practitioners a standardized tool for consistent evaluation and progress tracking in length-controlled generation tasks.

---

> ### Author Rebuttal · Authors · 2025-07-28
>
> Thank you for your thoughtful review and helpful feedback. We have carefully considered your comments and provide our responses to each point below. Your suggestions have been very valuable in improving our work.
>
> ---
>
>
>
> ## For Weakness 1
>
> We fully recognize that conflicts can arise between task requirements and length constraints, and we have taken multiple steps to ensure the reasonableness and validity of our tasks:
>
> - **Task Selection Based on Paradigm Suitability:** As detailed in *Appendix B.1: Mapping of NLP Tasks to Length Paradigms*, we systematically categorized NLP tasks into four Length Paradigms. Only tasks from the Flexible-Length Paradigm, where output length can be meaningfully adjusted, were included in the main LIFEBENCH dataset. This ensures that imposing length constraints is both appropriate and meaningful. **Tasks unsuited to arbitrary length constraints were excluded by design.**
> - **Rigorous Data Filtering and Manual Review:** As described in *Appendix B.2: Data Collection*, we established strict filtering rules and conducted manual review for all tasks. For instance, QA questions were required to be open-ended and context-rich, allowing for both concise and detailed answers. We excluded factoid, binary, multiple-choice, duplicate, or ill-formed questions, as these do not support the intended range of outputs. **All data passing these rules underwent an additional human review to ensure compatibility with diverse length constraints.**
> - **Task-Specific Templates:** In *Appendix B.3: Refinement Process*, we explain how separate templates were designed for each task subtype to ensure clear instructions and proper length constraints. Original instructions from source data were removed when necessary to prevent confusion or contradictions.
> - **Additional Datasets for Other Paradigms:** For Fixed-Length and Adaptive-Length Paradigm tasks—which can still be meaningfully length-constraints—we introduced two supplementary datasets(see *Appendix C*). These datasets use tailored constraints and templates, and their results are reported under all relevant length settings in *Appendix J*.
>
> ## For Weakness 2
>
> Thank you for your question regarding the Length Score hyperparameters. As explained in *Appendix D.2.2: Length Deviation and Length Score*, our choice of hyperparameters ($k1=5$, $k2=2$) is empirical and based on practical considerations. We observed that over-generation (LD ≥ 0) is generally more tolerable than under-generation, since excess content can be trimmed after generation, while missing content cannot be easily supplemented by users. Therefore, our scoring function penalizes under-generation more heavily and over-generation more lightly, increasing sensitivity to insufficient length compliance.
>
> ## For Weakness 3
>
> Thank you for raising this important point. **We have carefully discussed whether to include generation quality evaluation in our metrics, especially considering that strict length following could potentially impact content quality.** However, we chose not to incorporate quality as an evaluation metric for several reasons:
>
> - **Benchmark Focus:** LIFEBENCH is specifically designed to evaluate length instruction following. Including a quality metric could shift the benchmark’s focus, as there are already multiple established benchmarks for long-form generation quality.
> - **Length Bias in Quality Judging:** Prior work has shown that LLM-as-Judge may introduce length bias, with models tending to assign higher scores to longer outputs. Since output length is tightly coupled with the task in LIFEBENCH, including a quality metric would risk unfair comparisons across different constraints and potentially incentivize over-generation.
> - **Empirical Quality Analysis:** Despite this, we still provide an empirical analysis of generation quality under different length constraints in *Appendix K*. While we observe performance differences among models (see Figure 27) and a correlation between output length and quality score (see Figure 28), these findings further support our decision not to use quality as a primary evaluation metric for length compliance.
>
> ## For Weakness 4
>
> Since we used word-level constraints when designing the benchmark, as these are more consistent with human usage and expectations. All our datasets are thus evaluated using “words” rather than “tokens.”  We recognize that this approach does not directly address potential issues arising from the model’s ability to distinguish between words and tokens.**To investigate this further, we conducted additional experiments on our lightweight dataset, LIFEBENCH-LITE, using token-based constraints.** The experimental details are as follows:
>
> > Due to the small sample size in these experiments, Length Deviation is more sensitive to outliers (for example, outputs that repeat to reach the maximum window size). Therefore, we capped the maximum Length Deviation at 100%, which differs from the settings in the main paper. The length constraints tested were 16-8192 tokens, with the control method set to “equal to.” The reported LD and LS represent the averages across all results.
>
> **Table 1: Length Deviation and Length Score for all token-based length constraints (Due to the space limitations of this rebuttal, only a subset of the results is shown.)**
>
> | Models| LD(words) | LS(words) | LD(tokens) | LS(tokens) |
> | ----------------------------- | :-------: | :-------: | :--------: | :--------: |
> | GPT-4o|31%|49.1|53%|30.8|
> |o3-mini|13%| **75.4**|51%|38.9|
> |Claude-3.7-Sonnet|30%|55.4|54%|35.7|
> |Claude-3.7-Sonnet-Thinking|33%|61.3|55%|36.1|
> |Gemini-2.0-Flash|36%|48.4|59%|29.6|
> |Gemini-2.0-Flash-Thinking |53%|44.0|50%|  **40.0**  |
> |Doubao-1.5-Pro| 23%|48.7|60%|31.8|
> |Doubao-1.5-Thinking-Pro|29%|50.6|64%|27.1|
> |DeepSeek-V3| 27%|54.3|61%| 25.7|
> |DeepSeek-R1 |    36%    |   47.7    |    52%     |    35.3    |
> |Llama-3.1-70B-Instruct|    61%    |   42.4    |    66%     |    20.4    |
> |Qwen2.5-72B-Instruct|    28%    |   50.6    |    57%     |    27.4    |
> |Qwen3-235B-A22B|    22%    |   52.1    |    44%     |    29.8    |
> |Qwen3-235B-A22B-Thinking|    23%    |   57.7    |    48%     |    34.1    |
> |GLM-4-9B-Chat|40%|   35.2    |    66%     |    15.5    |
> |Mistral-7B-Instruct-v0.2|    84%    |   26.7    |    70%     |    15.3    |
>
> Based on our experimental results, all models exhibit even poorer length instruction following performance when evaluated with "tokens" as the unit, compared to using "words". **This finding further indicates that the observed language differences across models cannot be attributed to tokenization effects, as their performance is even lower at the token level.**
>
> ## For Weakness 5
>
> We would like to clarify that **all the experimental results we report exclude the output of the model’s reasoning/thinking process, and no length constraints are imposed on such intermediate outputs**. Our rationale is as follows:
>
> - **In practical applications, there is rarely a need to specify the strict length of the reasoning process.** Currently, model reasoning traces are primarily used to assist with tasks such as step-by-step reasoning or mathematical problem solving, serving as intermediate steps rather than final outputs. Imposing strict length constraints on these intermediate steps would be unrealistic and may not benefit the quality of the final answer.
> - **Limitations from model providers prevent a comprehensive analysis of reasoning traces.** Many proprietary models (e.g., *o1-mini*, *o3-mini*, *Gemini-2.5-Pro*) do not support access to the reasoning process, and some open-source models (e.g., *DeepSeek-R1*, *Qwen3-32B-Thinking*, *Qwen3-235B-A22B-Thinking*) have restrictions on the output length of intermediate traces. As a result, it is not possible to systematically evaluate or compare reasoning outputs across all models.
> - **LIFEBENCH is primarily focused on evaluating the final outputs rather than intermediate reasoning steps.** We regard the reasoning process as an auxiliary step to enhance output quality, but not the main focus of benchmarking. Imposing constraints on reasoning traces may interfere with the model’s actual output and would not meaningfully reflect its capabilities.
>
> ## For Weakness 6
>
> Thank you for your insightful suggestion. We also found the exceptional performance of *o3-mini* to be surprising and agree that it warrants deeper investigation. However, as *o3-mini* is a  proprietary model, there is currently limited public information available—such as training data, methods, and deployment details. Furthermore, the model’s internal reasoning process is not accessible via the API, making a comprehensive technical analysis difficult. Nevertheless, we conducted some preliminary case studies to explore possible reasons for its performance:
>
> - We observed that o3-mini frequently employs “lazy strategies.” Interestingly, when the model refuses to answer, its output latency is often much longer than for normal responses. **We hypothesize that o3-mini may be performing extensive internal reasoning or repeated attempts before ultimately refusing to answer**—a behavior that might be similar to what we observed in *Claude-3.7-Sonnet* (see *Appendix I*)
> - Our token-level evaluation results (Table 1 in this rebuttal) show a notable drop in o3-mini's performance when length constraints are imposed at the token level rather than the word level. This suggests that *o3-mini* may have been more extensively trained on word-based data, while token-based constraints are less common in practice and therefore less well learned. **This implies that its strong performance may be more attributable to tailored training data rather than unique model architecture.**
>
> ------
>
> We sincerely appreciate your thoughtful feedback, which has greatly contributed to improving the depth and quality of our work. **If our revisions have addressed your concerns, we would sincerely appreciate your consideration of a higher score.**

---

> > ### Comment · Reviewer_Ltyk · 2025-08-01
> >
> > Thank you for your thorough response and the new experimental results, which have addressed all my previous concerns. I have raised my score from 4 to 5.

---

### Official Review · Reviewer_iR3t · 2025-06-16

**Rating:** 5
**Confidence:** 4

**Summary:**

This paper introduces LIFEBENCH, a comprehensive benchmark designed to evaluate the ability of Large Language Models (LLMs) to follow explicit length instructions. The authors highlight that while LLMs excel at complex tasks, they often struggle with seemingly simpler requests like generating text of a specific word count, frequently producing outputs that are too short or refusing requests for long generations. LIFEBENCH addresses this gap by providing 10,800 instances across four task categories in both English and Chinese. It covers a wide spectrum of length constraints and three control methods. The paper evaluates 26 prominent LLMs, revealing that most models perform reasonably on short-length instructions but their performance sharply deteriorates for longer lengths, often failing to reach vendor-claimed maximum output capacities. Key findings also indicate that input characteristics like task type, language, and input length significantly impact performance. The benchmark and findings aim to uncover fundamental limitations in current LLMs and guide future development. The benchmark dataset and evaluation code are fully open-sourced, providing a standardized and extensible resource for the research community.

**Dataset Code Accessibility:**

Yes

**Ethical Considerations:**

No, there are no or only very minor ethics concerns

**Limitations Weaknesses:**

**Weaknesses:**

• Task design might have some gaps, particularly for long-text scenarios where it doesn't cover other common tasks like machine translation and code generation that are quite prevalent in real-world applications.

• The paper mentions "Discrepancies Between Claimed and Actual Maximum Output Length," but it's unclear whether Maximum Output Length refers to the context window, and whether the context window provided by models tends to be more about input capacity rather than specifically advertised output capabilities.

• "Lazy Strategies" represents an interesting phenomenon, but the paper lacks a rigorous definition and systematic investigation of this behavior. It also seems to miss deeper discussions, such as whether there are correlations with different control methods or length constraints.

• I only see a case study of Claude-3.7-Sonnet-Thinking in Appendix I, and it's questionable whether a single model case study can adequately support claims about this being a common characteristic of reasoning models in general.

• The task evaluation might face issues where models simply follow length instructions while output quality severely deteriorates—for example, generating meaningless or repetitive content—since the benchmark metrics don't assess instruction quality, and the data filtering process doesn't seem to consider removing such problematic outputs.

• While the paper identifies key phenomena, it provides limited mechanistic insights into what's actually causing these issues. The diagnostic findings could really benefit from deeper quantitative analysis or interpretability studies to go beyond just observing patterns.

**Limitations:**

The authors appropriately acknowledge in Section 6 that the work focuses on benchmarking rather than providing solutions to underlying causes.

**Strengths Contributions:**

• LIFEBENCH presents a quite interesting piece of work that takes a different approach from typical instruction-following benchmarks by concentrating specifically on length control--by building this targeted task and running comprehensive experiments, the authors uncover systematic weaknesses in what should be basic length following capabilities while offering insightful explanations that really help advance understanding of LLM limitations.

• This benchmark actually fills a real gap by establishing the first comprehensive evaluation framework specifically for length instruction following, and the methodological design is quite solid with four task types, bilingual support, granular length constraints (16-8192 words), and three control methods that reflect how people actually use these models.

•Evaluation methodology shows good experimental rigor throughout, with well-documented processes and thoughtfully designed metrics. Length Score does a nice job addressing limitations of simple deviation measures while staying interpretable.

• Experimental scope is pretty impressive and up to date, testing 26 diverse LLMs including models like o3-mini, Gemini-2.5-Pro, Claude-3.7-Sonnet, and Qwen3 series, which means findings actually reflect what's happening with current capabilities.

• Diagnostic analyses go well beyond just reporting performance numbers, identifying specific failure modes like length awareness deficits, lazy generation strategies, and intermediate reasoning limitations that genuinely help us understand how these models behave.

• Overall, this work is well-executed and clearly written throughout, and making it publicly available on HuggingFace is a solid contribution to the community that should help future research efforts.

---

> ### Author Rebuttal · Authors · 2025-07-28
>
> Thank you very much for your thorough review and insightful suggestions. We provide detailed responses and clarifications below to address the specific issues you raised.
>
> ---
>
> ## For Weakness 1
>
> Thank you very much for your comments. We believe there may be some misunderstandings regarding the design and scope of our task types. In fact, **our benchmark covers a wide range of NLP tasks, and we have also provided supplementary datasets to include additional tasks.** We address your concerns from two perspectives:
>
> - **Regarding task design**, the selection and categorization of tasks are detailed in *Appendix B: Details of LIFEBENCH*. We categorize NLP tasks into four Length Paradigms. The main LIFEBENCH dataset focuses exclusively on tasks within the Flexible-Length Paradigm. **Tasks such as machine translation and code generation do not fit this paradigm, since imposing arbitrary length restrictions on these tasks can compromise task completion—such as producing incomplete translations or non-compilable code. Therefore, these tasks are not included in the main dataset by design.**
> - **Regarding the supplementary datasets**, as mentioned in *Appendix B.1*, we acknowledge that Fixed-Length and Adaptive-Length tasks can also be subject to length control. To enhance the diversity of our benchmark, we have introduced two additional datasets in *Appendix C*, which specifically cover Fixed-Length and Adaptive-Length Paradigms, respectively. For example, **the *LIFEBENCH-REFACTOR* dataset contains 13 samples of machine translation, directly addressing your suggestion.** As code generation typically operates at the line (rather than word) level, it is not included in our datasets.
>
> ## For Weakness 2
>
> Thank you for pointing out this issue. There may be some confusion regarding the terminology. Specifically, "Maximum Output Length" does not refer to the model's context window. We clarify the distinction below:
>
> - **Maximum output length** refers to the maximum number of tokens the model can generate in one output. For example, Qwen2.5 models (*e.g., Qwen2.5-7B-Instruct, Qwen2.5-72B-Instruct*) explicitly specify their maximum output length in their documentation. As stated in the *Qwen2.5-72B-Instruct Model Card*: " **can generate up to 8K tokens**." For closed-source models, this value is usually specified by the API provider, For example, the *GPT-4o-mini Model Card* specifies "**16,384 max output tokens.**"
> - **Context window** refers to the maximum total number of tokens (input and output) the model can process. For example, the *Qwen2.5-72B-Instruct Model Card* states: "**Long-context Support up to 128K tokens**" and the *GPT-4o-mini Model Card* specifies a "**128,000 context window.**"
>
> Further details can be found in *Appendix D.1: Model Setup*, which lists the API or code reference, context window, and maximum output length for each model.
>
> ## For Weakness 3
>
> Thank you very much for your comments regarding our experiments and your suggestions for improvement. We agree that "lazy strategies" represent a particularly interesting phenomenon. **Due to space limitations in the main text, we were unable to provide a comprehensive discussion there; however, we have conducted an in-depth analysis in several appendices**:
>
> - In *Appendix H*, **we provide a clear definition and categorization of "lazy strategies," describe our detection methodology, and present both illustrative examples and distributional analysis under varying length constraints.**
> - In *Appendix M.2*, we further investigate the underlying mechanisms of lazy strategies and explore at which stage of training this behavior may emerge. By experimenting with four alignment-free models, we found that "lazy strategies" still occur frequently, suggesting that such behaviors can be introduced during pre-training. **Based on these findings, we propose possible optimizations at the pre-training stage, such as improving data quality and incorporating more long-text samples.**
>
> While we have provided a detailed analysis of this phenomenon, we highly appreciate your comments, which inspired us to conduct further experiments on the relationship between "lazy strategies" and different control methods:
>
> > We calculated the average proportion of "refusal with attempt" across all models. As shown in *Appendix H*, lazy strategies are almost absent under low length constraints, so we focused on cases with length constraints $\geq$ 1024 words to provide more meaningful results.
>
> **Table 1: Average proportion of lazy strategies across all models under different control methods.  (Due to the space limitations of this rebuttal, only a subset of the results is shown.)**
>
> | Models| equal to  | at most | at least |
> | -------------------------- | :-------: | :-----: | :------: |
> | GPT-4o | **5.3%**  | 0.8% |  3.1% |
> | o3-mini  | **16.3%** |  0.2%  |  2.6% |
> | Claude-3.7-Sonnet |   4.7%   | 3.1% | **6.9%** |
> | Claude-3.7-Sonnet-Thinking | 4.0%|  2.8%|   6.6%   |
> | Gemini-2.0-Flash | **4.7%**  |1.9%|   2.5%|
> | Gemini-2.0-Flash-Thinking  |3.1%|  2.3% | **3.5%** |
> | DeepSeek-V3 |1.5% |0.7%| **2.4%**|
> | DeepSeek-R1  | **8.3%** |  1.9% | 7.4% |
> | Llama-3.1-70B-Instruct | **2.0%**  |0.7% |1.4%|
> | Qwen2.5-72B-Instruct| **2.3%** |  0.6% |2.1%|
> | Qwen3-235B-A22B |0.7%|  2.0% |**2.2%**|
> | Qwen3-235B-A22B-Thinking |2.1%|1.5%| **2.4%** |
>
> Our results indicate that control methods indeed influence the occurrence of lazy strategies. When using the "at most" control method, the overall proportion is lowest. In contrast, under the "equal to" control condition, the average proportion is highest. **These findings demonstrate that models are more likely to employ lazy strategies when required to generate outputs of more precise or challenging length constraints.**
>
> ## For Weakness 4
>
> Thank you for your careful observation. We agree that presenting a case study with only one model could lead to some misunderstanding. We would like to clarify this point. Although reasoning models possess the ability to dynamically adjust output length, this capability does not fully resolve the issue. Furthermore, as closed-source models such as *o1-mini*, *o3-mini*, and *Gemini-2.5-Pro* do not provide access to their intermediate reasoning processes, it is not possible to perform similar case studies on these models. Therefore, we focus our analysis on the *Claude-3.7-Sonnet-Thinking* model in *Appendix I*.
>
> It is important to emphasize that the main purpose of this analysis is to illustrate that stepwise reasoning can sometimes enhance a model’s length following, but this does not imply that all reasoning models consistently adopt this approach, nor that all responses from a given model will follow this pattern. **Thus, we have not claimed that this is a “common characteristic of reasoning models in general.”** In fact, our observations indicate that this behavior is not universal. Typical failure patterns we observed in reasoning models include:
>
> - Generating only an outline of content rather than dynamically repeating or elaborating
> - Errors in word count at shorter lengths, where the model may still under- or over-count even with stepwise counting, leading to deviation from the length instruction
> - Inability to perform multi-step dynamic generation at longer lengths, resulting in outputs that do not meet the required length
> - Resorting to “lazy strategies” and refusing to answer when the model determines the specified length cannot be met
>
> ## For Weakness 5
>
> Thank you for raising this important and meaningful question. We have discussed this issue extensively within our author team. **It is true that strict length compliance can sometimes result in a decline in output quality, yet our evaluation protocol does not currently include a direct measure of quality.** The reasons are as follows:
>
> - **LIFEBENCH is specifically designed to evaluate length instruction following. Its primary goal is to provide a dataset that measures a model’s ability to adhere to specified length constraints. Incorporating a quality metric could shift the focus away from this core purpose.** Furthermore, there already exist multiple established benchmarks for evaluating long-form generation quality, so we believe that quality evaluation is outside the primary scope of our benchmark.
> - Previous research has shown that **using LLM-as-Judge methods introduces length bias**: models may assign higher scores to longer outputs. Since outputs in LIFEBENCH are highly correlated with length, introducing a quality metric would inevitably bring length bias, making fair comparison across different length constraints difficult. This was a key factor in our decision not to include automated quality metrics.
> - Nevertheless, we do empirically analyze the relationship between output quality and length constraint in *Appendix K*,  we observe that different models indeed show varying quality under different length constraints (see Figure 27). **Additionally, we observe a clear correlation between output length and quality score (see Figure 28), which further supports our decision not to use quality as an official metric for length compliance.**
>
> ## For Weakness 6
>
> Thank you for highlighting this important aspect and suggesting potential directions for improvement. While the main contribution of our paper is the proposed benchmark and the identification of key issues, **we have also discussed possible solutions in detail**. Specifically, in *Appendix M: Future Directions and Potential Solutions*, we address this issue from three perspectives: *Interpretability of Length Instruction Following*, *Improving Length Instruction Following During Pre-training* and *Improving Length Instruction Following During Post-training.*
>
> ------
>
> We are grateful for your suggestions, which have helped us to further enhance the rigor and clarity of our work. **If our revisions have addressed your concerns, we would sincerely appreciate your consideration of a higher score.**

---

### Official Review · Reviewer_Lxuy · 2025-07-01

**Rating:** 5
**Confidence:** 4

**Summary:**

The paper propose LIFEBench, a benchmark for length instruction following evluation of current Large Language Models (LLMs). The benchmark dataset, openly released on Hugging Face, contains data of diverse tasks, various lengths and langauges. The evaluation of length instruction following on both proprietary and open-sourced LLMs shows that current LLMs are still not reliable in generating the outputs of particular length.

**Additional Feedback:**

Please consider to improve the paper by address the weaknesses above.

**Dataset Code Accessibility:**

Yes

**Dataset Code Comments:**

The benchmark is released on Huggingface and provide sufficient details for reproducibility.

**Ethical Comments:**

The paper studies the length following capacities of LLMs and does not involve sensitive data.

**Ethical Considerations:**

No, there are no or only very minor ethics concerns

**Final Justification:**

I have read the authors' responses which have addressed my concerns.

**Limitations Weaknesses:**

**Paper writing needs to be improved**: Although the paper is generally well written, I think it can be enhanced by providing details and motivations of critical designs. For example, when presenting the metrics and the dataset design choices, it is important to clarify the reasons and motivations. In the current form there are missed description fo this point.

** Missed discussion**: It is also important to elaborate potential improvement for the length instruction following, as an inpiration for future research. The paper conducts various analysis but there lacks a summarization of potential solutions.

Additionally, I think there are the following minor issues:
- Section 5.2 does not provide statistics on the output length of the reasoning process in reasoning models. As far as I know, reasoning models can typically produce longer outputs through test-time scaling during the reasoning stage; however, there may still be issues with output length in the non-reasoning parts. Therefore, it is necessary to clarify further whether the current output length limitation is due to the model's inherent capabilities or insufficient application of test-time scaling in the non-reasoning parts.
- The "length awareness" experiment in Section 5.3 mainly demonstrates that the model often reports the output length inaccurately, but does not prove whether it has a genuine internal awareness of length. I believe this method cannot fully reflect whether the model has an internal understanding of output or input length, and only shows that its self-assessment is inaccurate. If the authors could supplement this with tests on awareness of input length or analyze the model's internal state, the evaluation of length perception would be more convincing and comprehensive.
- Although Section 5.3 states that "by analyzing intermediate reasoning traces, we observe that reasoning models can calibrate their output length during the reasoning process," and notes that "this ability to dynamically adjust output length only partially alleviates the problem for short length constraints," the results in Table 1 do not fully support this conclusion. According to the experimental results, many reasoning models perform even worse than general models under short length constraints. Moreover, the case study in Appendix I does not provide specific examples of reasoning model failures for short lengths. The authors should offer a more detailed explanation and analysis of these inconsistencies to clarify the actual performance of reasoning models in complying with length instructions.

**Strengths Contributions:**

**Significance**: The paper designs a new benchmark across languages and tasks, allowing future LLM designer to well consider the length instruction following ability of the LLM. Imposing length is useful and necessary in many cases including daily writing tasks and downstream LLM application where the output should be too short or too long.

**Novelty**: The paper studies an interesting research problem, and proposes novel evaluation metrics to quantify the length instruction following ability. The study is the first study on the length ability while considering various tasks.

** Impact and Relevance**: I think the paper is relevant to the main topics of dataset and benchmark track and can be useful for the community to design better length instruction following LLM.

---

> ### Author Rebuttal · Authors · 2025-07-28
>
> Thank you very much for your thoughtful review and constructive comments. We have carefully considered your feedback and have addressed your concerns as follows.
>
> ---
>
> ## For Weakness 1
>
> Thank you for your suggestions regarding our writing. **Due to the length limitations of the main text**, we were unable to fully elaborate on the motivations and reasoning behind our selection of metrics and dataset design, which may have led to some confusion. **However, we have provided detailed explanations of the benchmark and metrics design, along with their underlying motivations, in *Appendix B: Details of LIFEBENCH* and *Appendix D.2: Evaluation Metrics*. Please find a summary below:**
>
> - **Appendix B**: In Section B.1, we introduce four types of Length Paradigms—Length-Independent Paradigm, Fixed-Length Paradigm, Adaptive-Length Paradigm, and Flexible-Length Paradigm. We detail their representative NLP tasks, explain our rationale for choosing certain tasks and data for the Flexible-Length Paradigm, and discuss the motivation for designing these four types of tasks in LIFEBENCH. To further enrich our dataset, we also provide two supplementary datasets, *LIFEBENCH-LABEL* and *LIFEBENCH-REFACTOR*, in *Appendix C: Details of LIFEBENCH-SUPPLEMENTARY*. Section B.2 further describes the data collection processes, sources, and selection criteria for all four tasks, including screening rules and manual verification. **We believe this supplementary information in Appendix B addresses your concerns regarding the dataset.**
> - **Appendix D.2**: Here we explain the design and motivation behind our two evaluation metrics, *Length Deviation* and *Length Score*. We analyze the limitations of using only one metric and provide concrete examples to demonstrate the complementary benefits of both. We also describe the statistical methods used for different control strategies and visualize the metrics in Figure 8. Additionally, we discuss our choice of hyperparameters: $k_1$ is set larger ($k_1 = 5$) and $k_2$ smaller ($k_2 = 2$), since over-generation ($LD ≥ 0$) is generally more acceptable than under-generation; excess content can be trimmed, whereas missing content cannot be easily supplemented. We further illustrate the impact of these parameters with specific examples. **We believe that the details in Appendix D.2 address your concerns about our evaluation metrics.**
>
> ## For Weakness 2
>
> Thank you for your suggestions regarding possible improvements to our paper. While the main contribution of our work is the proposed benchmark, accompanying analysis, and multiple insights, **we have also provided a comprehensive discussion of potential solutions**. Specifically, these are addressed in three parts in *Appendix M: Future Directions and Potential Solutions*:
>
> - **Interpretability of Length Instruction Following**: We analyze the relationship between EOS logits, EOS probability, and length instructions for four models. Through experiments, we observe the diverse behaviors of different models and suggest possible solutions based on mechanism analysis.
>
> - **Improving Length Instruction Following During Pre-training**: We investigate the "lazy strategy" phenomenon, exploring at which training stage it may arise. Experiments with four unaligned models reveal the prevalence of this phenomenon, leading us to recommend improvements for the pre-training stage, such as enhancing data quality and incorporating more long-text data.
>
> - **Improving Length Instruction Following During Post-training**: **To address limitations in long-text generation, we propose a pre-planning strategy.** This involves manually drafting structured content outlines (similar to book tables of contents) and using these to guide LLMs in generating longer, more coherent texts. **Our experiments demonstrate significant improvements across multiple models using this method.(see Table 19)**
>
> ## For Weakness 3
>
> Thank you for your valuable comments on potential improvements to our article. We address your points as follows:
>
> - Your observation is very insightful. While many reasoning models can now generate longer step-by-step thought processes through post-training, **in real-world applications, it is uncommon for the reasoning process to be required to follow a strict length constraint**. Typically, the reasoning process serves as an intermediate step to facilitate problem-solving in tasks such as reasoning and mathematics, and enforcing an exact output length is often neither reasonable nor beneficial. Therefore, we did not evaluate whether the model’s reasoning process can produce content of a specified length.
>   **Nevertheless, we provide a detailed analysis of how reasoning models respond to length instructions in *Appendix I: Case Study: How Reasoning Models Handle Length Constraints via Stepwise Thinking.*** In this appendix, we analyze model performance under three different length requirements and observe frequent over-reasoning and redundant steps, indicating that current reasoning models often employ inefficient strategies in attempts to satisfy length constraints.
>   **Additionally, limitations imposed by many model providers further restricted our experiments.** For example, closed-source models such as *o1-mini*, *o3-mini*, and *Gemini-2.5-Pro* do not support outputting the model’s reasoning process. Similarly, some open-source models (*DeepSeek-R1*, *Qwen3-32B-Thinking*, and *Qwen3-235B-A22B-Thinking*) impose restrictions on output length, preventing us from conducting comprehensive quantitative analyses. As a result, we selected *Claude-3.7-Sonnet-Thinking* as a representative case study.
>
> - Regarding the "length awareness" experiment, **we have already provided a detailed analysis in *Appendix F: Length Awareness: Do LLMs Know How Long Their Generations Are?***. We not only measured the models’ output length, but also conducted a “Marker-Based Length Monitoring” experiment, where models were prompted to output a special token at every specified interval to assess their length awareness ability. The results show that, even with such markers, performance degrades as length constraints increase, revealing inherent deficiencies in current models' length awareness.
>   **Furthermore, we analyzed the internal mechanism of EOS generation from an interpretability perspective in *Appendix M.1: Interpretability of Length Instruction Following*, which further explains the internal mechanism of length awareness.**
>   We very much appreciate your suggestion to also consider input awareness. To address this, we conducted additional experiments on our lightweight dataset, *LIFEBENCH-LITE*, as follows:
>
>   > For these experiments, we modified *LIFEBENCH-LITE* by removing the [Instruction] and [Requirement] fields (see *Appendix B.3: Refinement Process*), and added a new prompt: *[Requirement] Count the number of words in the text above and output the word count directly.*
>   > We compared the outputted word count to the actual input using the statistical method described in *Appendix D.2.1 Output Word Count Computation*
>
>   **Table 1:  Input Length Awareness Results. (Due to the space limitations of this rebuttal, only a subset of the results is shown.)**
>
>   | Models                     | Average  Word  Count Difference | Average Percentage  Error (%) |
>   | -------------------------- | :-----------------------------: | :---------------------------: |
>   | GPT-4o                     |             183.50              |            213.90%            |
>   | o3-mini                    |             166.55              |            205.46%            |
>   | Claude-3.7-Sonnet          |             183.05              |            221.30%            |
>   | Claude-3.7-Sonnet-Thinking |             170.15              |            198.12%            |
>   | Gemini-2.0-Flash           |             167.10              |            184.02%            |
>   | Gemini-2.0-Flash-Thinking  |             173.20              |            233.20%            |
>   | DeepSeek-V3                |             187.25              |            264.53%            |
>   | DeepSeek-R1                |             169.20              |            201.07%            |
>   | Llama-3.1-70B-Instruct     |             199.30              |            252.48%            |
>   | Qwen2.5-72B-Instruct       |             181.60              |            230.88%            |
>   | Qwen3-235B-A22B            |             182.65              |            235.92%            |
>   | Qwen3-235B-A22B-Thinking   |             167.70              |            214.52%            |
>
>   According to the experimental results,**the models still have notable deficiencies in input length awareness, consistent with our overall findings.**
>
> - Your careful observation regarding confusing behaviors in reasoning models is much appreciated. To clarify, in *Section 5.3: Diagnosis of Length Instruction Following Failures*, we note: “this ability to dynamically adjust output length only partially alleviates the problem for short length constraints and still fails under longer constraints.” This indicates that, **while reasoning models can adjust output length to some extent, this capability is limited and does not guarantee success, especially for longer constraints.**
>   Due to the restrictions of closed-source models, our case studies in *Appendix I* focus on *Claude-3.7-Sonnet-Thinking*. The main purpose of this analysis is to demonstrate that reasoning models can sometimes improve length-following through stepwise generation, but **this is not a universal solution and may not apply to all models or all cases.** Therefore, the length-following performance is not strictly determined by whether a model is a reasoning model.
>
> ------
>
> Thank you once again for your detailed review and valuable feedback. **If our revisions have addressed your concerns, we would sincerely appreciate your consideration of a higher score.**

---

### Official Review · Reviewer_eaEw · 2025-07-03

**Rating:** 5
**Confidence:** 4

**Summary:**

The authors highlight that while current language models possess long-context reasoning abilities, they struggle with adhering to length constraints. Existing benchmarks only check the generation quality and do not check such a length constraint. The authors intend to fill this gap with their benchmark, LIFEBENCH, that asks models to abide by length constraints across a variety of tasks and length ranges in English and Chinese. Via this benchmark, they test present-day models and find that they are good at following short-length instructions but fail for longer-length instructions. Authors formalize the problem, provide metrics, and a taxonomy of tasks grouped under different length paradigms. Through a host of experiments, authors show limitations of current models in following length instructions across NLP tasks.

**Additional Feedback:**

Some corrections:
- The reference to Figure 1 in the last paragraph of the introduction should be "top right" instead of "top left".

**Dataset Code Accessibility:**

Yes

**Dataset Code Comments:**

I checked the benchmark on Hugging Face, and looked at the code in the repository. The code appears to be well-documented and usable, thereby supporting reproducibility.

**Ethical Considerations:**

No, there are no or only very minor ethics concerns

**Final Justification:**

I will keep my score at 5. I think the benchmark's objective is still confounded in testing counting abilities. I see it as a benchmark that measures whether the model can follow the provided instruction (for which many benchmarks already exist) and constrain its response to be within the mentioned limit (which indeed requires counting its output tokens). Unless the models can reliably count their generated tokens (and additionally know the tokenizer's merge rules in case the constrain is in terms of words), they will not perform well on this benchmark. Hence, the results here are not surprising. I would be surprised if a model that could otherwise track its response length (i.e., tells you the response length correctly) fails on this benchmark. Authors argue that counting is trivial (which I do not agree with; I think it is a very challenging task) and that their benchmark measures much more than that. This is not coming out well and remains an important limitation. Despite the fact, I still think this benchmark is ready to be revealed to the community so further research can take place collectively.

**Limitations Weaknesses:**

- You mention under RQ1 (in introduction) that the length instruction following failures are due to LLMs being unable to track their generated word count. So, is LIFEBENCH measuring LLMs' counting abilities? Also, is this inability due to LLM confusing between words and tokens? Have you tried instructing the model to abide by length constraints in terms of tokens? Figure 4 has results for the number of tokens, but I am not sure if the model was instructed as such.
- I fail to understand the difference between RQ1 and RQ2. The first part of RQ2 is already covered under RQ1. Only the (in)ability to follow vendor-claimed generation length is new. I suggest rephrasing it.
- **Metrics.** Even though the unit of measurement is introduced later, I suggest labeling the y-axis of Figure 1 (top right). Also, *Length Score* is not defined before being used in the introduction section for RQ1 results. It is hard to make sense of those results without defining the metric.
- **Writing could be improved.** Concepts such as metrics and length control methods are used before being introduced or defined. A careful refinement will aid readability. Moreover, the paper does not follow the NeurIPS review template with line numbers.
- Was it simply an intuitive choice, or is there some science to the selection of $k_1=5$ and $k_2=2$ in *Length Score* definition?

**Strengths Contributions:**

- The paper highlights a problem (failure to abide by length instructions) and proposes a setup to quantify it. I found it quite interesting that LMs fail to reach vendor-claimed generation lengths. A careful examination of why such biases toward shorter lengths emerge might inform future model development.
- The experiments in the paper present interesting findings, such as tasks that are hardest in terms of length instruction following, and language-specific biases (overextension in Chinese compared to English).
- I appreciated how the authors have carefully developed this setup to evaluate the said limitation in LLMs. The metrics and the concept of length paradigms with grouping of NLP tasks are very well done.

---

> ### Author Rebuttal · Authors · 2025-07-28
>
> We sincerely thank you for your valuable comments. We will address the weaknesses in order to alleviate your concerns about our paper.
>
> ---
>
> ## For Weakness 1
>
> Thank you for your valuable and insightful comments. We address your questions point by point below:
>
> - **Regarding “Is LIFEBENCH measuring LLMs' counting abilities?”**
>   We acknowledge that RQ1 in the introduction may not have clearly conveyed our intended meaning, potentially leading to misunderstandings. We would like to clarify that **LIFEBENCH is designed to evaluate the model's ability to follow length instructions**. The phrase “measuring LLMs' counting abilities” does not accurately capture the scope of our benchmark. In fact, evaluating length instruction following is a broader and more complex task than simply measuring counting abilities. The key distinctions are as follows:
>
>   - **Task Objective:** *Length Instruction Following* assesses whether a model can comprehend and adhere to length instructions during text generation, whereas *counting abilities* focus primarily on the model’s capacity to identify and quantify specific elements, such as the number of words.
>   - **Task Difficulty:** *Length Instruction Following* requires the model to understand both the semantic content and the length constraints, generating text that meets the specified requirements. In contrast, *counting abilities* involve merely counting words, making the former a significantly more challenging task.
>   - **Practical Relevance:** *Length Instruction Following* is essential for a wide range of NLP tasks, such as open-ended question answering (see *Appendix B.1: Mapping of NLP Tasks to Length Paradigms*). *Counting abilities*, on the other hand, represent a relatively simple skill that can often be addressed by basic word-counting tools, and thus offer limited practical value for comprehensive evaluation.
>
>   In summary, counting ability can be viewed as a subset of length instruction following. The statement in RQ1 regarding “LLMs being unable to track their generated word count” is intended to highlight current model limitations in following length instructions, rather than to define the primary evaluation objective of LIFEBENCH. Our further analyses, such as those in *Appendix F (“Length Awareness: Do LLMs Know How Long Their Generations Are?”)*, are meant to provide additional insights into the challenges of length instruction following, not as the main purpose of the benchmark.
>
> - **Regarding the question: “Is this inability due to LLMs confusing between words and tokens?”**
>   In designing the benchmark, we chose to use word-level constraints as these are more consistent with human usage and expectations. All our datasets are thus evaluated using “words” rather than “tokens.” For the results presented in Figure 4, the unit of measurement is also “words”; we only use “tokens” for supplementary statistical analysis. We recognize that this approach does not directly address potential issues arising from the model’s ability to distinguish between words and tokens.
>
>   To investigate this further, we conducted additional experiments on our lightweight dataset, LIFEBENCH-LITE, using token-based constraints. The experimental details are as follows:
>
>   > Due to the small sample size in these experiments, Length Deviation is more sensitive to outliers (for example, outputs that repeat to reach the maximum window size). Therefore, we capped the maximum Length Deviation at 100%, which differs from the settings in the main paper. The length constraints tested were 16-8192 tokens, with the control method set to “equal to.” The reported LD and LS represent the averages across all results.
>
>   **Table 1: Length Deviation and Length Score for all token-based length constraints (Due to the space limitations of this rebuttal, only a subset of the results is shown.)**
>
>   | Models                        | LD(words) | LS(words) | LD(tokens) | LS(tokens) |
>   | ----------------------------- | :-------: | :-------: | :--------: | :--------: |
>   | GPT-4o                        |    31%    |   49.1    |    53%     |    30.8    |
>   | o3-mini                       |  13%  | **75.4**  |    51%     |    38.9    |
>   | Claude-3.7-Sonnet             |    30%    |   55.4    |    54%     |    35.7    |
>   | Claude-3.7-Sonnet-Thinking    |    33%    |   61.3    |    55%     |    36.1    |
>   | Gemini-2.0-Flash              |    36%    |   48.4    |    59%     |    29.6    |
>   | Gemini-2.0-Flash-Thinking |    53%    |   44.0    |    50%     |  **40.0**  |
>   | Doubao-1.5-Pro                |    23%    |   48.7    |    60%     |    31.8    |
>   | Doubao-1.5-Thinking-Pro       |    29%    |   50.6    |    64%     |    27.1    |
>   | DeepSeek-V3                   |    27%    |   54.3    |    61%     |    25.7    |
>   | DeepSeek-R1                   |    36%    |   47.7    |    52%     |    35.3    |
>   | Llama-3.1-70B-Instruct        |    61%    |   42.4    |    66%     |    20.4    |
>   | Qwen2.5-72B-Instruct          |    28%    |   50.6    |    57%     |    27.4    |
>   | Qwen3-235B-A22B               |    22%    |   52.1    |    44%     |    29.8    |
>   | Qwen3-235B-A22B-Thinking      |    23%    |   57.7    |    48%     |    34.1    |
>   | GLM-4-9B-Chat                 |    40%    |   35.2    |    66%     |    15.5    |
>   | Mistral-7B-Instruct-v0.2      |    84%    |   26.7    |    70%     |    15.3    |
>
>   Based on our experimental results, **all models exhibit even poorer length instruction following performance when evaluated with "tokens" as the unit, compared to using "words"**. We believe this may be because training data typically includes "words," while explicit "token"-level constraints are rare. This finding further demonstrates that the model's difficulty in following length instructions is not primarily due to confusion between "words" and "tokens," but rather stems from an inherent limitation in the model's length instruction following capability.
>
> ## For Weakness 2
>
> Thank you very much for your valuable suggestions, especially your observation that part of the first step in RQ2 overlaps with RQ1. We would like to clarify that the first part of RQ2 is indeed distinct from RQ1. Below, we elaborate on the differences between RQ1 and RQ2 from the perspectives of related work and research focus, in the hope that this explanation clarifies their distinctions:
>
> - **From the perspective of related work:** There is a clear difference between research on length instruction compliance and research on long text generation. This distinction is discussed in detail in our *Appendix A: Comparison of LIFEBENCH with Existing Benchmarks*. For example, benchmarks focused on length instruction compliance, such as *IFEval* and *AlpacaEval-LI*, typically target shorter lengths (generally less than 1,000 words), while long text generation research, such as *LongBench-Write*, often overlooks compliance for shorter lengths (less than 100 words). LIFEBENCH encompasses both ends of the spectrum. **To facilitate comparison with these two lines of related work, we present the evaluation results in two separate RQs.**
> - **From the perspective of research focus:** RQ1 investigates whether a model can precisely generate responses within a specified length constraint, while RQ2 explores the model's ability to generate long-form text. For example, in our experiments, models such as *LongWriter-Llama3.1-8B* and *LongWriter-GLM4-9B-Chat* have been specifically optimized for long text generation and are capable of producing significantly longer outputs than most other models. Nevertheless, as shown in Table 1 and Table 11, while these models excel in generating long texts (RQ2), they still struggle to accurately conform to specific length constraints, especially for shorter lengths (less than 100 words, RQ1). Thus, **it is possible for a model to perform well on RQ2 but not on RQ1, indicating that RQ2 cannot be fully subsumed under RQ1.**
>
> We have, in fact, carefully considered and discussed this issue multiple times during the writing process, especially with respect to how we articulate RQ1 and RQ2. The current statement reflects several rounds of refinement. Nevertheless, we are eager to continue discussing with you how we might further revise and improve our paper. Your guidance is extremely valuable to us, and we sincerely appreciate your suggestions.
>
> ## For Weakness 3 and Weakness 4
>
> Thank you very much for your kind suggestions regarding writing and illustration improvements. **We will define our metrics and introduce the relevant metrics and length control methods in the Introduction, as well as add corresponding labels to Figure 1. Additionally, as you suggested, we will include line numbers to further enhance the readability of the paper.**
>
> ## For Weakness 5
>
> Regarding your question, our reasoning is as follows: **This is an empirical choice,** as explained in *Appendix D.2.2: Length Deviation and Length Score (Hyperparameter Settings)*. This approach reflects the observation that over-generation (LD ≥ 0) is generally more acceptable than under-generation, since excess content can be more easily trimmed post hoc, whereas missing content may require users to supplement the output. Consequently, our scoring function penalizes under-generation more severely and over-generation more leniently, thereby increasing the sensitivity to insufficient length adherence. We hope this explanation addresses your concern.
>
> ## For Additional Feedback
>
> We greatly appreciate your careful attention to detail. **Your correction is absolutely accurate, and we will make the necessary amendment.**
>
> ------
>
> Once again, thank you very much for your thorough review and constructive feedback. If you have any further questions or suggestions, please feel free to raise any further concerns. **If our revisions have successfully addressed your concerns, we would greatly appreciate your consideration in raising your score.**

---

> > ### Comment · Reviewer_eaEw · 2025-08-02
> > **Thanks for the rebuttal**
> >
> > I thank the authors for their rebuttal. My evaluation of this work continues to be positive and I choose to maintain my rating of 5.

---

### Decision · Program_Chairs · 2025-09-18

**Decision:**

Accept (poster)

**Comment:**

This paper presents LIFEBENCH, a benchmark for evaluating length instruction following in LLMs. The reviewers found the work timely and significant, addressing a fundamental yet underexplored capability with clear practical importance. Strengths include the well-designed benchmark (four task types, bilingual support, fine-grained constraints), the Length Score metric, and the comprehensive evaluation of 26 diverse LLMs. The diagnostic analyses offer useful insights into failure modes such as lack of length awareness and lazy generation. I recommend acceptance.